# Linear Convergence of Natural Policy Gradient Methods with Log-Linear Policies

**Rui Yuan**
FAIR, Meta AI
LTCI, Télécom Paris and Institut Polytechnique de Paris
yy42606r@gmail.com

**Simon S. Du**
University of Washington
ssdu@cs.washington.edu

**Robert M. Gower**
CCM, Flatiron Institute
gowerrobert@gmail.com

**Alessandro Lazaric**
FAIR, Meta AI
lazaric@meta.com

**Lin Xiao**
FAIR, Meta AI
linx@meta.com

## Abstract

We consider infinite-horizon discounted Markov decision processes and study the convergence rates of the natural policy gradient (NPG) and the Q-NPG methods with the log-linear policy class. Using the compatible function approximation framework, both methods with log-linear policies can be written as inexact versions of the policy mirror descent (PMD) method. We show that both methods attain linear convergence rates and $\tilde{\mathcal{O}}(1/\epsilon^2)$ sample complexities using a simple, non-adaptive geometrically increasing step size, without resorting to entropy or other strongly convex regularization. Lastly, as a byproduct, we obtain sublinear convergence rates for both methods with arbitrary constant step size.

## 1 Introduction

Policy gradient (PG) methods have emerged as a popular class of algorithms for reinforcement learning. Unlike classical methods based on (approximate) dynamic programming (e.g., Puterman, 1994; Sutton & Barto, 2018), PG methods update directly the policy and its parametrization along the gradient direction of the value function (e.g., Williams, 1992; Sutton et al., 2000; Baxter & Bartlett, 2001). An important variant of PG is the natural policy gradient (NPG) method (Kakade, 2001). NPG uses the Fisher information matrix of the policy distribution as a preconditioner to improve the policy gradient direction, similar to quasi-Newton methods in classical optimization. Variants of NPG with policy parametrization through deep neural networks were shown to have impressive empirical successes (Schulman et al., 2015; Lillicrap et al., 2016; Mnih et al., 2016; Schulman et al., 2017).

Motivated by the success of NPG in practice, there is now a concerted effort to develop convergence theories for the NPG method. Neu et al. (2017) provide the first interpretation of NPG as a mirror descent (MD) method (Nemirovski & Yudin, 1983; Beck & Teboulle, 2003). By leveraging different techniques for analyzing MD, it has been established that NPG converges to the global optimum in the tabular case (Agarwal et al., 2021; Khodadadian et al., 2021b; Xiao, 2022) and some more general settings (Shani et al., 2020; Vaswani et al., 2022; Grudzien et al., 2022). In order to get a fast linear convergence rate for NPG, several recent works consider the regularized NPG methods, such as the entropy-regularized NPG (Cen et al., 2021) and other convex regularized NPG methods (Lan, 2022; Zhan et al., 2021). By designing appropriate step sizes, Khodadadian et al. (2021b) and Xiao (2022) obtain linear convergence of NPG without regularization. However, all these linear convergence results are limited in the tabular setting with a direct parametrization. It remains unclear whether this same linear convergence rate can be established in function approximation settings.

In this paper we provide an affirmative answer to this question for the log-linear policy class. Our approach is based on the framework of *compatible function approximation* (Sutton et al., 2000; Kakade, 2001), which was extensively developed by Agarwal et al. (2021). Using this framework, variants of NPG with log-linear policies can be written as policy mirror descent (PMD) methods with inexact evaluations of the advantage function or Q-function (giving rise to NPG or Q-NPG respectively). Then by extending a recent analysis of PMD (Xiao, 2022), we obtain a non-asymptotic linear

convergence of both NPG and Q-NPG with log-linear policies. A distinctive feature of this approach is the use of a simple, non-adaptive geometrically increasing step size, without resorting to entropy or other (strongly) convex regularization. The extensions are highly nontrivial and require quite different techniques. This linear convergence leads to the $\tilde{\mathcal{O}}(1/\epsilon^2)$ sample complexities for both methods. In particular, our sample complexity analysis also fixes errors of previous work. Lastly, as a byproduct, we obtain sublinear convergence rates for both methods with arbitrary constant step size. See Appendix A for a thorough review. In particular, Table 1 provides a complete overview of our results.

## 2 PRELIMINARIES ON MARKOV DECISION PROCESSES

We consider an MDP denoted as $\mathcal{M} = \{\mathcal{S}, \mathcal{A}, \mathcal{P}, c, \gamma\}$, where $\mathcal{S}$ is a finite state space, $\mathcal{A}$ is a finite action space, $\mathcal{P} : \mathcal{S} \times \mathcal{A} \to \mathcal{S}$ is a Markovian transition model with $\mathcal{P}(s' \mid s, a)$ being the transition probability from state $s$ to $s'$ under action $a$, $c$ is a cost function with $c(s, a) \in [0, 1]$ for all $(s, a) \in \mathcal{S} \times \mathcal{A}$, and $\gamma \in [0, 1)$ is a discounted factor. Here we use cost instead of reward to better align with the minimization convention in the optimization literature.

The agent's behavior is modeled as a stochastic policy $\pi \in \Delta(\mathcal{A})^{|\mathcal{S}|}$, where $\pi_s \in \Delta(\mathcal{A})$ is the probability distribution over actions $\mathcal{A}$ in state $s \in \mathcal{S}$. At each time $t$, the agent takes an action $a_t \in \mathcal{A}$ given the current state $s_t \in \mathcal{S}$, following the policy $\pi$, i.e., $a_t \sim \pi_{s_t}$. Then the MDP transitions into the next state $s_{t+1}$ with probability $\mathcal{P}(s_{t+1} \mid s_t, a_t)$ and the agent encounters the cost $c_t = c(s_t, a_t)$. Thus, a policy induces a distribution over trajectories $\{s_t, a_t, c_t\}_{t \geq 0}$. In the infinite-horizon discounted setting, the cost function of $\pi$ with an initial state $s$ is defined as

$$V_s(\pi) \overset{\text{def}}{=} \mathbb{E}_{a_t \sim \pi_{s_t}, s_{t+1} \sim \mathcal{P}(\cdot \mid s_t, a_t)} \Big[ \sum\nolimits_{t=0}^{\infty} \gamma^t c(s_t, a_t) \mid s_0 = s \Big]. \tag{1}$$

Given an initial state distribution $\rho \in \Delta(\mathcal{S})$, the goal of the agent is to find a policy $\pi$ that minimizes

$$V_\rho(\pi) \overset{\text{def}}{=} \mathbb{E}_{s \sim \rho} [V_s(\pi)] = \sum\nolimits_{s \in \mathcal{S}} \rho_s V_s(\pi) = \langle V(\pi), \rho \rangle.$$

A more granular characterization of the performance of a policy is the state-action cost function (Q-function). For any pair $(s, a) \in \mathcal{S} \times \mathcal{A}$, it is defined as

$$Q_{s,a}(\pi) \overset{\text{def}}{=} \mathbb{E}_{a_t \sim \pi_{s_t}, s_{t+1} \sim \mathcal{P}(\cdot \mid s_t, a_t)} \Big[ \sum\nolimits_{t=0}^{\infty} \gamma^t c(s_t, a_t) \mid s_0 = s, a_0 = a \Big]. \tag{2}$$

Let $Q_s \in \mathbb{R}^{|\mathcal{A}|}$ denote the vector $[Q_{s,a}]_{a \in \mathcal{A}}$. Then we have $V_s(\pi) = \mathbb{E}_{a \sim \pi_s}[Q_{s,a}(\pi)] = \langle \pi_s, Q_s(\pi) \rangle$. The advantage function[1] is a centered version of the Q-function: $A_{s,a}(\pi) \overset{\text{def}}{=} Q_{s,a}(\pi) - V_s(\pi)$, which satisfies $\mathbb{E}_{a \sim \pi_s}[A_{s,a}(\pi)] = 0$ for all $s \in \mathcal{S}$.

**Visitation probabilities.** Given a starting state distribution $\rho \in \Delta(\mathcal{S})$, we define the *state visitation distribution* $d^\pi(\rho) \in \Delta(\mathcal{S})$, induced by a policy $\pi$, as

$$d_s^\pi(\rho) \overset{\text{def}}{=} (1 - \gamma) \mathbb{E}_{s_0 \sim \rho} \Big[ \sum\nolimits_{t=0}^{\infty} \gamma^t \Pr^\pi(s_t = s \mid s_0) \Big],$$

where $\Pr^\pi(s_t = s \mid s_0)$ is the probability that $s_t$ is equal to $s$ by following the trajectory generated by $\pi$ starting from $s_0$. Intuitively, the state visitation distribution measures the probability of being at state $s$ across the entire trajectory. We define the *state-action visitation distribution* $\bar{d}^\pi(\rho) \in \Delta(\mathcal{S} \times \mathcal{A})$ as

$$\bar{d}_{s,a}^\pi(\rho) \overset{\text{def}}{=} d_s^\pi(\rho) \pi_{s,a} = (1 - \gamma) \mathbb{E}_{s_0 \sim \rho} \Big[ \sum\nolimits_{t=0}^{\infty} \gamma^t \Pr^\pi(s_t = s, a_t = a \mid s_0) \Big]. \tag{3}$$

In addition, we extend the definition of $\bar{d}^\pi(\rho)$ by specifying the initial state-action distribution $\nu$, i.e.,

$$\tilde{d}_{s,a}^\pi(\nu) \overset{\text{def}}{=} (1 - \gamma) \mathbb{E}_{(s_0, a_0) \sim \nu} \Big[ \sum\nolimits_{t=0}^{\infty} \gamma^t \Pr^\pi(s_t = s, a_t = a \mid s_0, a_0) \Big]. \tag{4}$$

The difference in the last two definitions is that for the former, the initial action $a_0$ is sampled directly from $\pi$, whereas for the latter, it is prescribed by the initial state-action distribution $\nu$. We use $\tilde{d}$ compared to $\bar{d}$ to better distinguish the cases with $\nu$ and $\rho$. Without specification, we even omit the argument $\nu$ or $\rho$ throughout the paper to simplify the presentation as they are self-evident. From these definitions, we have for all $(s, a) \in \mathcal{S} \times \mathcal{A}$,

$$d_s^\pi \geq (1 - \gamma)\rho_s, \qquad \bar{d}_{s,a}^\pi \geq (1 - \gamma)\rho_s \pi_{s,a}, \qquad \tilde{d}_{s,a}^\pi \geq (1 - \gamma)\nu_{s,a}. \tag{5}$$

---

[1]An advantage function should measure how much better is $a$ compared to $\pi$, while here $A$ is positive when $a$ is worse than $\pi$. We keep calling $A$ advantage function to better align with the convention in the RL literature.

**Policy parametrization.** In practice, both the state and action spaces $\mathcal{S}$ and $\mathcal{A}$ can be large and some form of function approximation is needed to reduce the dimensions and make the computation feasible. In particular, the policy $\pi$ is often parametrized as $\pi(\theta)$ with $\theta \in \mathbb{R}^m$, where $m$ is much smaller than $|\mathcal{S}|$ and $|\mathcal{A}|$. In this paper, we focus on the log-linear policy class. Indeed, we assume that for each state-action pair $(s, a)$, there is a feature mapping $\phi_{s,a} \in \mathbb{R}^m$ and the policy takes the form

$$\pi_{s,a}(\theta) \;=\; \frac{\exp(\phi_{s,a}^\top \theta)}{\sum_{a' \in \mathcal{A}} \exp(\phi_{s,a'}^\top \theta)}. \tag{6}$$

This setting is important since it is the simplest instantiation of the widely-used neural policy parametrization. To simplify notations in the rest , we use the shorthand $V_\rho(\theta)$ for $V_\rho(\pi(\theta))$ and similarly $Q_{s,a}(\theta)$ for $Q_{s,a}(\pi(\theta))$, $A_{s,a}(\theta)$ for $A_{s,a}(\pi(\theta))$, $d_s^\theta$ for $d_s^{\pi(\theta)}$, $\bar{d}_{s,a}^\theta$ for $\bar{d}_{s,a}^{\pi(\theta)}$, and $\tilde{d}_{s,a}^\theta$ for $\tilde{d}_{s,a}^{\pi(\theta)}$.

**Natural Policy Gradient (NPG) Method.** Using the notations defined above, the parametrized policy optimization problem is to minimize the function $V_\rho(\theta)$ over $\theta \in \mathbb{R}^m$. The policy gradient is given by (see, e.g., Williams, 1992; Sutton et al., 2000)

$$\nabla_\theta V_\rho(\theta) = \frac{1}{1-\gamma} \mathbb{E}_{s \sim d^\theta,\, a \sim \pi_s(\theta)} \left[ Q_{s,a}(\theta) \, \nabla_\theta \log \pi_{s,a}(\theta) \right]. \tag{7}$$

For parametrizations that are differentiable and satisfy $\sum_{a \in \mathcal{A}} \pi_{s,a}(\theta) = 1$, including the log-linear class defined in (6), we can replace $Q_{s,a}(\theta)$ by $A_{s,a}(\theta)$ in the above expression (Agarwal et al., 2021). The NPG method (Kakade, 2001) takes the form

$$\theta^{(k+1)} \;=\; \theta^{(k)} - \eta_k F_\rho\big(\theta^{(k)}\big)^\dagger \nabla_\theta V_\rho\big(\theta^{(k)}\big), \tag{8}$$

where $\eta_k > 0$ is a scalar step size, $F_\rho(\theta)$ is the Fisher information matrix

$$F_\rho(\theta) \;\stackrel{\text{def}}{=}\; \mathbb{E}_{s \sim d^\theta,\, a \sim \pi_s(\theta)} \big[ \nabla_\theta \log \pi_{s,a}(\theta) \big( \nabla_\theta \log \pi_{s,a}(\theta) \big)^\top \big], \tag{9}$$

and $F_\rho(\theta)^\dagger$ denotes the Moore-Penrose pseudoinverse of $F_\rho(\theta)$.

## 3 NPG WITH COMPATIBLE FUNCTION APPROXIMATION

The parametrized value function $V_\rho(\theta)$ is non-convex in general (see, e.g., Agarwal et al., 2021). Despite being a non-convex optimization problem, there is still additional structure we can leverage to ensure convergence. Following Agarwal et al. (2021), we adopt the framework of *compatible function approximation* (Sutton et al., 2000; Kakade, 2001), which exploits the MDP structure and leads to tight convergence rate analysis.

For any $w, \theta \in \mathbb{R}^m$ and state-action distribution $\zeta$, define *compatible function approximation error* as

$$L_A(w, \theta, \zeta) \;\stackrel{\text{def}}{=}\; \mathbb{E}_{(s,a) \sim \zeta} \big[ \big( w^\top \nabla_\theta \log \pi_{s,a}(\theta) - A_{s,a}(\theta) \big)^2 \big]. \tag{10}$$

Kakade (2001) showed that the NPG update (8) is equivalent to (up to a constant scaling of $\eta_k$)

$$\theta^{(k+1)} \;=\; \theta^{(k)} - \eta_k w_\star^{(k)}, \qquad w_\star^{(k)} \in \operatorname{argmin}_{w \in \mathbb{R}^m} L_A\big(w, \theta^{(k)}, \bar{d}^{(k)}\big), \tag{11}$$

where $\bar{d}^{(k)}$ is a shorthand for the state-action visitation distribution $\bar{d}^{\pi(\theta^{(k)})}(\rho)$ defined in (3). A derivation of (11) is provided in Appendix B (Lemma 1) for completeness. In other words, $w_\star^{(k)}$ is the solution to a regression problem that tries to approximate $A_{s,a}(\theta^{(k)})$ using $\nabla_\theta \log \pi_{s,a}(\theta^{(k)})$ as features. This is where the term "compatible function approximation error" comes from. For the log-linear policy class defined in (6), we have

$$\nabla_\theta \log \pi_{s,a}(\theta) \;=\; \bar{\phi}_{s,a}(\theta) \;\stackrel{\text{def}}{=}\; \phi_{s,a} - \sum_{a' \in \mathcal{A}} \pi_{s,a'}(\theta) \phi_{s,a'} \;=\; \phi_{s,a} - \mathbb{E}_{a' \sim \pi_s(\theta)} [\phi_{s,a'}], \tag{12}$$

where $\bar{\phi}_{s,a}(\theta)$ are called *centered features vectors*.

In practice, we cannot minimize $L_A$ exactly; instead, a sample-based regression problem is solved to obtain an approximate solution $w^{(k)}$. This leads to the following inexact NPG update rule:

$$\theta^{(k+1)} \;=\; \theta^{(k)} - \eta_k w^{(k)}, \qquad w^{(k)} \approx \operatorname{argmin}_w L_A\big(w, \theta^{(k)}, \bar{d}^{(k)}\big). \tag{13}$$

The inexact NPG updates require samples of unbiased estimates of $A_{s,a}(\theta)$, the corresponding sampling procedure is given in Algorithm 4, and a sample-based regression solver to minimize $L_A$ is given in Algorithm 5 in the Appendix.

Alternatively, as proposed by Agarwal et al. (2021), we can define the compatible function approximation error as

$$L_Q(w, \theta, \zeta) \stackrel{\text{def}}{=} \mathbb{E}_{(s,a)\sim\zeta}\left[\left(w^\top \phi_{s,a} - Q_{s,a}(\theta)\right)^2\right] \tag{14}$$

and use it to derive a variant of the inexact NPG update called *Q-NPG*:

$$\theta^{(k+1)} = \theta^{(k)} - \eta_k w^{(k)}, \qquad w^{(k)} \approx \arg\min_w L_Q\left(w, \theta^{(k)}, \bar{d}^{(k)}\right). \tag{15}$$

For Q-NPG, the sampling procedure for estimating $Q_{s,a}(\theta)$ is given in Algorithm 3 and a sample-based regression solver for $w^{(k)}$ is proposed in Algorithm 6 in the Appendix.

The sampling procedure and the regression solver of NPG are less efficient than those of Q-NPG. Indeed, the sampling procedure for $A_{s,a}(\theta)$ in Algorithm 4 not only estimates $Q_{s,a}(\theta)$, but also requires an additional estimation of $V_s(\theta)$, and thus doubles the amount of samples as compared to Algorithm 3. Furthermore, the stochastic gradient estimator of $L_Q$ in Algorithm 6 only computes on a single action of the feature map $\phi_{s,a}$. Whereas the one of $L_A$ in Algorithm 5 computes on the centered feature map $\bar{\phi}_{s,a}(\theta)$ defined in (12), which needs to go through the entire action space, thus is $|\mathcal{A}|$ times more expensive to run. See Appendix C for more details.

Following Agarwal et al. (2021), we consider slightly different variants of NPG and Q-NPG, where $\bar{d}^{(k)}$ in (13) and (15) is replaced by a more general state-action visitation distribution $\tilde{d}^{(k)} = \tilde{d}^{\pi(\theta^{(k)})}(\nu)$ defined in (4) with $\nu \in \Delta(\mathcal{S} \times \mathcal{A})$. The advantage of using $\tilde{d}^{(k)}$ is that it allows better exploration than $\bar{d}^{(k)}$ as $\nu$ can be chosen to be independent to the policy $\pi(\theta^{(k)})$. For example, it can be seen from (5) that the lower bound of $\tilde{d}^\pi$ is independent to $\pi$, which is not the case for $\bar{d}^\pi$. This property is crucial in the forthcoming convergence analysis.

## 3.1 Formulation as Inexact Policy Mirror Descent

Given an approximate solution $w^{(k)}$ for minimizing $L_Q\left(w, \theta^{(k)}, \tilde{d}^{(k)}\right)$, the Q-NPG update rule $\theta^{(k+1)} = \theta^{(k)} - \eta_k w^{(k)}$, when plugged in the log-linear parametrization (6), results in a new policy

$$\pi_{s,a}^{(k+1)} = \pi_{s,a}^{(k)} \exp\left(-\eta_k \phi_{s,a}^T w^{(k)}\right)/Z_s^{(k)}, \qquad \forall (s,a) \in \mathcal{S} \times \mathcal{A},$$

where $\pi^{(k)}$ is a shorthand for $\pi_{s,a}(\theta^{(k)})$ and $Z_s^{(k)}$ is a normalization factor to ensure $\sum_{a\in\mathcal{A}} \pi_{s,a}^{(k+1)} = 1$, for each $s \in \mathcal{S}$. We note that the above $\pi^{(k+1)}$ can also be obtained by a mirror descent update:

$$\pi_s^{(k+1)} = \arg\min_{p\in\Delta(\mathcal{A})} \left\{\eta_k \left\langle \Phi_s w^{(k)}, p\right\rangle + D(p, \pi_s^{(k)})\right\}, \quad \forall s \in \mathcal{S}, \tag{16}$$

where $\Phi_s \in \mathbb{R}^{|\mathcal{A}|\times m}$ is a matrix with rows $(\phi_{s,a})^\top \in \mathbb{R}^{1\times m}$ for $a \in \mathcal{A}$, and $D(p,q)$ denotes the Kullback-Leibler (KL) divergence between two distributions $p, q \in \Delta(\mathcal{A})$, i.e., $D(p,q) \stackrel{\text{def}}{=} \sum_{a\in\mathcal{A}} p_a \log\left(\frac{p_a}{q_a}\right)$. A derivation of (16) is provided in Appendix B (Lemma 2) for completeness.

If we replace $\Phi_s w^{(k)}$ in (16) by the vector $\left[Q_{s,a}(\pi^{(k)})\right]_{a\in\mathcal{A}} \in \mathbb{R}^{|\mathcal{A}|}$, then it becomes the *policy mirror descent* (PMD) method in the tabular setting studied by, for example, Shani et al. (2020), Lan (2022) and Xiao (2022). In fact, the update rule (16) can be viewed as an inexact PMD method where $Q_s(\pi^{(k)})$ is linearly approximated by $\Phi_s w^{(k)}$ through compatible function approximation (14). Similarly, we can write the inexact NPG update rule as

$$\pi_s^{(k+1)} = \arg\min_{p\in\Delta(\mathcal{A})} \left\{\eta_k \left\langle \bar{\Phi}_s^{(k)} w^{(k)}, p\right\rangle + D(p, \pi_s^{(k)})\right\}, \quad \forall s \in \mathcal{S}, \tag{17}$$

where $w^{(k)}$ is an inexact solution for minimizing $L_A\left(w, \theta^{(k)}, \tilde{d}^{(k)}\right)$ defined in (10), and $\bar{\Phi}_s^{(k)} \in \mathbb{R}^{|\mathcal{A}|\times m}$ is a matrix whose rows consist of the centered feature maps $\left(\bar{\phi}_{s,a}(\theta^{(k)})\right)^\top$, as defined in (12).

Reformulating Q-NPG and NPG into the mirror descent forms (16) and (17), respectively, allows us to adapt the analysis of PMD method developed in Xiao (2022) to obtain sharp convergence rates. In particular, we show that with an increasing step size $\eta_k \propto \gamma^k$, both NPG and Q-NPG with log-linear policy parametrization converge linearly up to an error floor determined by the quality of the compatible function approximation.

## 4 ANALYSIS OF Q-NPG WITH LOG-LINEAR POLICIES

In this section, we provide the convergence analysis of the following inexact Q-NPG method

$$\theta^{(k+1)} = \theta^{(k)} - \eta_k w^{(k)}, \qquad w^{(k)} \approx \operatorname{argmin}_w L_Q\big(w, \theta^{(k)}, \tilde{d}^{(k)}\big), \tag{18}$$

where $\tilde{d}^{(k)}$ is shorthand for $\tilde{d}^{\pi(\theta^{(k)})}(\nu)$ and $\nu \in \Delta(\mathcal{S} \times \mathcal{A})$ is an arbitrary state-action distribution that does not depend on $\rho$. The exact minimizer is denoted as $w_\star^{(k)} \in \operatorname{argmin}_w L_Q\big(w, \theta^{(k)}, \tilde{d}^{(k)}\big)$. Following Agarwal et al. (2021), the compatible function approximation error can be decomposed as

$$L_Q\big(w^{(k)}, \theta^{(k)}, \tilde{d}^{(k)}\big) = \underbrace{L_Q\big(w^{(k)}, \theta^{(k)}, \tilde{d}^{(k)}\big) - L_Q\big(w_\star^{(k)}, \theta^{(k)}, \tilde{d}^{(k)}\big)}_{\text{Statistical error (excess risk)}} + \underbrace{L_Q\big(w_\star^{(k)}, \theta^{(k)}, \tilde{d}^{(k)}\big)}_{\text{Approximation error}}.$$

The statistical error measures how accurate is our solution to the regression problem, i.e., how good $w^{(k)}$ is compared with $w_\star^{(k)}$. The approximation error measures the best possible solution for approximating $Q_{s,a}(\theta^{(k)})$ using $\phi_{s,a}$ as features in the regression problem (modeling error). One way to proceed with the analysis is to assume that both the statistical error and the approximation error are bounded for all iterations, which is the approach we take in Section 4.2 and is also the approach we take later in Section 5 for the analysis of the NPG method.

However, in Section 4.1, we first take an alternative approach proposed by Agarwal et al. (2021), where the assumption of bounded approximation error is replaced by a bounded *transfer error*. The transfer error refers to $L_Q\big(w_\star^{(k)}, \theta^{(k)}, \tilde{d}^*\big)$, where the iteration-dependent visitation distribution $\tilde{d}^{(k)}$ is shifted to a fixed one $\tilde{d}^*$ (defined in Section 4.1).

These two approaches require different additional assumptions and result in slightly different convergence rates. Here we first state the common assumption on the bounded statistical error.

**Assumption 1** (Bounded statistical error, Assumption 6.1.1 in Agarwal et al. (2021)). *There exists $\epsilon_{\text{stat}} > 0$ such that for all iterations $k \geq 0$ of the Q-NPG method (18), we have*

$$\mathbb{E}\left[L_Q\big(w^{(k)}, \theta^{(k)}, \tilde{d}^{(k)}\big) - L_Q\big(w_\star^{(k)}, \theta^{(k)}, \tilde{d}^{(k)}\big)\right] \leq \epsilon_{\text{stat}}. \tag{19}$$

By solving the regression problem with sampling based approaches, we can expect $\epsilon_{\text{stat}} = \mathcal{O}(1/\sqrt{T})$ (Agarwal et al., 2021) or $\epsilon_{\text{stat}} = \mathcal{O}(1/T)$ (see Corollary 1) where $T$ is the number of iterations used to find the inexact solution $w^{(k)}$.

### 4.1 ANALYSIS WITH BOUNDED TRANSFER ERROR

Here we introduce some additional notations. For any state distributions $p, q \in \Delta(\mathcal{S})$, we define the *distribution mismatch coefficient* of $p$ relative to $q$ as $\|p/q\|_\infty \overset{\text{def}}{=} \max_{s \in \mathcal{S}} p_s/q_s$. Let $\pi^*$ be an arbitrary *comparator policy*, which is not necessarily an optimal policy and does not need to belong to the log-linear policy class. Fix a state distribution $\rho \in \Delta(\mathcal{S})$. We denote $d^{\pi^*}(\rho)$ as $d^*$ and $d^{\pi(\theta^{(k)})}(\rho)$ as $d^{(k)}$, and define the following distribution mismatch coefficients:

$$\vartheta_k \overset{\text{def}}{=} \left\|\frac{d^*}{d^{(k)}}\right\|_\infty \overset{(5)}{\leq} \frac{1}{1-\gamma}\left\|\frac{d^*}{\rho}\right\|_\infty \qquad \text{and} \qquad \vartheta_\rho \overset{\text{def}}{=} \frac{1}{1-\gamma}\left\|\frac{d^*}{\rho}\right\|_\infty \geq \frac{1}{1-\gamma}. \tag{20}$$

Thus, for all $k \geq 0$, we have $\vartheta_k \leq \vartheta_\rho$. We assume that $\vartheta_\rho < \infty$, which is the case, for example, if $\rho_s > 0$ for all $s \in \mathcal{S}$. This is commonly used in the literature on policy gradient methods (e.g., Zhang et al., 2020; Wang et al., 2020) and the NPG convergence analysis (e.g., Cayci et al., 2021; Xiao, 2022). We further relax this condition in Appendix H.

Given a state distribution $\rho$ and a comparator policy $\pi^*$, we define a state-action measure $\tilde{d}^*$ as

$$\tilde{d}_{s,a}^* \overset{\text{def}}{=} d_s^* \cdot \text{Unif}_{\mathcal{A}}(a) \overset{\text{def}}{=} d_s^*/|\mathcal{A}|, \tag{21}$$

and use it to express the transfer error as $L_Q\big(w_\star^{(k)}, \theta^{(k)}, \tilde{d}^*\big)$.

**Assumption 2** (Bounded transfer error, Assumption 6.1.2 in Agarwal et al. (2021))**.** *There exists* $\epsilon_{\text{bias}} > 0$ *such that for all iterations* $k \geq 0$ *of the Q-NPG method* (18)*, we have*

$$\mathbb{E}\left[L_Q\left(w_\star^{(k)}, \theta^{(k)}, \tilde{d}^*\right)\right] \leq \epsilon_{\text{bias}}. \tag{22}$$

The $\epsilon_{\text{bias}}$ is often referred to as the transfer error, since it is the error due to replacing the relevant distribution $\tilde{d}^{(k)}$ by $\tilde{d}^*$. This transfer error bound characterizes how well the Q-values can be linearly approximated by the feature maps $\phi_{s,a}$. It can be shown that $\epsilon_{\text{bias}} = 0$ when $\pi^{(k)}$ is the softmax tabular policy (Agarwal et al., 2021) or the MDP has a certain low-rank structure (Jiang et al., 2017; Yang & Wang, 2019; Jin et al., 2020). For rich neural parametrizations, $\epsilon_{\text{bias}}$ can be made small (Wang et al., 2020).

The next assumption concerns the relative condition number between two covariance matrices of $\phi_{s,a}$ defined under different state-action distributions.

**Assumption 3** (Bounded relative condition number, Assumption 6.2 in Agarwal et al. (2021))**.** *Fix a state distribution* $\rho$, *a state-action distribution* $\nu$ *and a comparator policy* $\pi^*$. *Let*

$$\Sigma_{\tilde{d}^*} \overset{def}{=} \mathbb{E}_{(s,a)\sim\tilde{d}^*}\left[\phi_{s,a}\phi_{s,a}^\top\right], \qquad and \qquad \Sigma_\nu \overset{def}{=} \mathbb{E}_{(s,a)\sim\nu}\left[\phi_{s,a}\phi_{s,a}^\top\right], \tag{23}$$

*where* $\tilde{d}^*$ *is specified in* (21)*. We define the relative condition number between* $\Sigma_{\tilde{d}^*}$ *and* $\Sigma_\nu$ *as*

$$\kappa_\nu \overset{def}{=} \max_{w \in \mathbb{R}^m} \frac{w^\top \Sigma_{\tilde{d}^*} w}{w^\top \Sigma_\nu w}, \tag{24}$$

*and assume that* $\kappa_\nu$ *is finite.*

The $\kappa_\nu$ is referred to as the relative condition number, since the ratio is between two different matrix induced norm. Assumption 3 benefits from the use of $\nu$. In fact, it is shown in Agarwal et al. (2021, Remark 22 and Lemma 23) that $\kappa_\nu$ can be reasonably small (e.g., $\kappa_\nu \leq m$ is always possible) and independent to the size of the state space by controlling $\nu$.

Our analysis also needs the following assumption, which does not appear in Agarwal et al. (2021).

**Assumption 4** (Concentrability coefficient for state visitation)**.** *There exists a finite* $C_\rho > 0$ *such that for all iterations* $k \geq 0$ *of the Q-NPG method* (18)*, it holds that*

$$\mathbb{E}_{s\sim d^*}\left[\left(\frac{d_s^{(k)}}{d_s^*}\right)^2\right] \leq C_\rho. \tag{25}$$

The concentrability coefficient is studied in the analysis of approximate dynamic programming algorithms (Munos, 2003; 2005; Munos & Szepesvári, 2008). It measures how much $\rho$ can get amplified in $k$ steps as compared to the reference distribution $d_s^*$. Let $\rho_{\min} = \min_{s\in\mathcal{S}} \rho_s$. A sufficient condition for Assumption 4 to hold is that $\rho_{\min} > 0$. Indeed,

$$\sqrt{\mathbb{E}_{s\sim d^*}\left[\left(\frac{d_s^{(k)}}{d_s^*}\right)^2\right]} \leq \left\|\frac{d^{(k)}}{d^*}\right\|_\infty \overset{(5)}{\leq} \frac{1}{1-\gamma}\left\|\frac{d^{(k)}}{\rho}\right\|_\infty \leq \frac{1}{(1-\gamma)\rho_{\min}}. \tag{26}$$

In reality, $\sqrt{C_\rho}$ can be much smaller than the pessimistic bound shown above. This is especially the case if we choose $\pi^*$ to be the optimal policy and $d^{(k)} \to d^*$. We further replace $C_\rho$ by $C_\nu$ defined in Section 4.2 that is independent to $\rho$ and thus is more easily satisfied.

Now we present our first main result.

**Theorem 1.** *Fix a state distribution* $\rho$, *an state-action distribution* $\nu$ *and a comparator policy* $\pi^*$. *We consider the Q-NPG method* (18) *with the uniform initial policy* [2] *and the step sizes satisfying* $\eta_0 \geq \frac{1-\gamma}{\gamma}\log|\mathcal{A}|$ *and* $\eta_{k+1} \geq \frac{1}{\gamma}\eta_k$. *Suppose that Assumptions 1, 2, 3 and 4 all hold. Then we have for all* $k \geq 0$,

$$\mathbb{E}\left[V_\rho(\pi^{(k)})\right] - V_\rho(\pi^*) \leq \left(1 - \frac{1}{\vartheta_\rho}\right)^k \frac{2}{1-\gamma} + \frac{2\sqrt{|\mathcal{A}|}(\vartheta_\rho\sqrt{C_\rho}+1)}{1-\gamma}\left(\sqrt{\frac{\kappa_\nu}{1-\gamma}\epsilon_{\text{stat}}} + \sqrt{\epsilon_{\text{bias}}}\right).$$

---

[2]For simplicity, we present all our results with the uniform initial policy in the main paper. See Theorem 5, 7 and 8 in the Appendix for more general results with any initial policy.

The main differences between our Theorem 1 and Theorem 20 of Agarwal et al. (2021), which is their corresponding result on the inexact Q-NPG method, are summarized as follows.

- The convergence rate of Agarwal et al. (2021, Theorem 20) is $\mathcal{O}(1/\sqrt{k})$ up to an error floor determined by $\epsilon_{\text{stat}}$ and $\epsilon_{\text{bias}}$. We have linear convergence up to an error floor that also depends on $\epsilon_{\text{stat}}$ and $\epsilon_{\text{bias}}$. However, the magnitude of our error floor is worse (larger) by a factor of $\vartheta_\rho \sqrt{C_\rho}$, due to the concentrability and the distribution mismatch coefficients used in our proof. A very pessimistic bound on this factor is as large as $|\mathcal{S}|^2/(1-\gamma)^2$.

- In terms of required conditions, both results use Assumptions 1, 2 and 3. Agarwal et al. (2021, Theorem 20) further assume that the norms of the feature maps $\phi_{s,a}$ are uniformly bounded and $w^{(k)}$ has a bounded norm (e.g., obtained by a projected stochastic gradient descent). Due to different analysis techniques referred next, we avoid such boundedness assumptions but rely on the concentrability coefficient $C_\rho$ defined in Assumption 4.

- Agarwal et al. (2021, Theorem 20) uses a diminishing step size $\eta \propto 1/\sqrt{k}$ where $k$ is the total number of iterations, but we use a geometrically increasing step size $\eta_k \propto \gamma^k$ for all $k \geq 0$. This discrepancy reflects the different analysis techniques adopted. The key analysis tool in Agarwal et al. (2021) is a *NPG Regret Lemma* (their Lemma 34) which relies on the smoothness of the functions $\log \pi_{s,a}(\theta)$ (thus the boundedness of $\|\phi_{s,a}\|$) and the boundedness of $\|w^{(k)}\|$, and thus the classical $\mathcal{O}(1/\sqrt{k})$ diminishing step size in the optimization literature. Our analysis exploits the three-point descent lemma (Chen & Teboulle, 1993) and the performance difference lemma (Kakade & Langford, 2002), without reliance on smoothness parameters. As a consequence, we can take advantage of exponentially growing step sizes and avoid assuming the boundedness of $\|\phi_{s,a}\|$ or $\|w^{(k)}\|$.

The reason of using increasing step size can be interpreted as approximate policy iteration. See Appendix H the connection with policy iteration for more details.

As a by product, we also obtain a sublinear $\mathcal{O}(1/k)$ convergence result while using arbitrary constant step size.

**Theorem 2.** *Fix a state distribution $\rho$, an state-action distribution $\nu$ and an optimal policy $\pi^*$. We consider the Q-NPG method (18) with the uniform initial policy and any constant step size $\eta_k = \eta > 0$. Suppose that Assumptions 1, 2, 3 and 4 all hold. Then we have for all $k \geq 0$,*

$$\frac{1}{k}\sum_{t=0}^{k-1} \mathbb{E}[V_\rho(\pi^{(t)})] - V_\rho(\pi^*) \leq \frac{1}{(1-\gamma)k}\left(\frac{\log|\mathcal{A}|}{\eta} + 2\vartheta_\rho\right) + \frac{2\sqrt{|\mathcal{A}|}\left(\vartheta_\rho\sqrt{C_\rho}+1\right)}{1-\gamma}\left(\sqrt{\frac{\kappa_\nu}{1-\gamma}\epsilon_{\text{stat}}} + \sqrt{\epsilon_{\text{bias}}}\right).$$

### 4.2 ANALYSIS WITH BOUNDED APPROXIMATION ERROR

In this section, instead of assuming bounded transfer error, we provide a convergence analysis based on the usual notion of approximation error and a weaker concentrability coefficient.

**Assumption 5** (Bounded approximation error). *There exists $\epsilon_{\text{approx}} > 0$ such that for all iterations $k \geq 0$ of the Q-NPG method (18), it holds that*

$$\mathbb{E}\left[L_Q\left(w_\star^{(k)}, \theta^{(k)}, \tilde{d}^{(k)}\right)\right] \leq \epsilon_{\text{approx}}. \tag{27}$$

As mentioned in Agarwal et al. (2021), Assumption 5 is stronger than Assumption 2 (bounded transfer error). Indeed,

$$L_Q\left(w_\star^{(k)}, \theta^{(k)}, \tilde{d}^*\right) \leq \left\|\frac{\tilde{d}^*}{\tilde{d}^{(k)}}\right\|_\infty L_Q\left(w_\star^{(k)}, \theta^{(k)}, \tilde{d}^{(k)}\right) \overset{(5)}{\leq} \frac{1}{1-\gamma}\left\|\frac{\tilde{d}^*}{\nu}\right\|_\infty L_Q\left(w_\star^{(k)}, \theta^{(k)}, \tilde{d}^{(k)}\right).$$

**Assumption 6** (Concentrability coefficient for state-action visitation). *There exists $C_\nu < \infty$ such that for all iterations of the Q-NPG method (18), we have*

$$\mathbb{E}_{(s,a)\sim\tilde{d}^{(k)}}\left[\left(\frac{h_{s,a}^{(k)}}{\tilde{d}_{s,a}^{(k)}}\right)^2\right] \leq C_\nu, \tag{28}$$

*where $h_{s,a}^{(k)}$ represents all of the following quantities:*

$$d_s^{(k+1)}\pi_{s,a}^{(k+1)}, \qquad d_s^{(k+1)}\pi_{s,a}^{(k)}, \qquad d_s^*\pi_{s,a}^{(k)}, \qquad \text{and} \quad d_s^*\pi_{s,a}^*. \tag{29}$$

Since we are free to choose $\nu$ independently to $\rho$, we can choose $\nu_{s,a} > 0$ for all $(s,a) \in \mathcal{S} \times \mathcal{A}$ for Assumption 6 to hold. Indeed, with $\nu_{\min}$ denoting $\min_{(s,a) \in \mathcal{S} \times \mathcal{A}} \nu_{s,a}$, we have

$$\sqrt{\mathbb{E}_{(s,a)\sim\tilde{d}^{(k)}}\left[\left(\frac{h_{s,a}^{(k)}}{\tilde{d}_{s,a}^{(k)}}\right)^2\right]} \leq \max_{(s,a)\in\mathcal{S}\times\mathcal{A}} \frac{h_{s,a}^{(k)}}{\tilde{d}_{s,a}^{(k)}} \overset{(5)}{\leq} \frac{1}{(1-\gamma)\nu_{\min}}, \tag{30}$$

where the upper bound can be smaller than that in (26) if $\rho_{\min}$ is smaller than $\nu_{\min}$.

**Theorem 3.** *Fix a state distribution $\rho$, an state-action distribution $\nu$ and a comparator policy $\pi^*$. We consider Q-NPG method (18) with uniform initial policy and step sizes satisfying $\eta_0 \geq \frac{1-\gamma}{\gamma} \log|\mathcal{A}|$ and $\eta_{k+1} \geq \frac{1}{\gamma}\eta_k$. Suppose that Assumptions 1, 5 and 6 hold. Then we have for all $k \geq 0$,*

$$\mathbb{E}[V_\rho(\pi^{(k)})] - V_\rho(\pi^*) \leq \left(1 - \frac{1}{\vartheta_\rho}\right)^k \frac{2}{1-\gamma} + \frac{2\sqrt{C_\nu}(\vartheta_\rho+1)}{1-\gamma}\left(\sqrt{\epsilon_{\mathrm{stat}}} + \sqrt{\epsilon_{\mathrm{approx}}}\right).$$

Compared to Theorem 1, while the approximation error assumption is stronger than the transfer error assumption, we do not require the assumption on relative condition number $\kappa_\nu$ and the error floor does not depends on $\kappa_\nu$ nor explicitly on $|\mathcal{A}|$. Besides, we can always choose $\nu$ so that $C_\nu$ is finite even if $C_\rho$ is unbounded. However, it is not clear if Theorem 3 is better than Theorem 1.

**Remark 1.** *Note that Theorems 1, 2 and 3 benefit from using the visitation distribution $\tilde{d}^{(k)}$ instead of $\bar{d}^{(k)}$ (i.e., benefit from using $\nu$ instead of $\rho$). In particular, from (5), $\tilde{d}^{(k)}$ has a lower bound that is independent to the policy $\pi^{(k)}$ or $\rho$. This property allows us to define a weak notion of relative condition number (Assumption 3) that is independent to the iterates, and also allows us to get a finite upper bound of $C_\nu$ (Assumption 6 and (30)) that is independent to $\rho$.*

### 4.3 SAMPLE COMPLEXITY OF Q-NPG

Here we establish the sample complexity results, i.e., total number of samples of single-step interaction with the environment, of a sample-based Q-NPG method (Algorithm 2 in Appendix C). Combined with a simple SGD solver, `Q-NPG-SGD` in Algorithm 6, the following corollary shows that Algorithm 2 converges globally by further assuming that the feature map is bounded and has non-singular covariance matrix (31). The explicit constants of the bound can be found in the appendix.

**Corollary 1.** *Consider the setting of Theorem 3. Suppose that the sample-based Q-NPG Algorithm 2 is run for $K$ iterations, with $T$ gradient steps of `Q-NPG-SGD` (Algorithm 6) per iteration. Furthermore, suppose that for all $(s,a) \in \mathcal{S} \times \mathcal{A}$, we have $\|\phi_{s,a}\| \leq B$ with $B > 0$, and we choose the step size $\alpha = \frac{1}{2B^2}$ and the initialization $w_0 = 0$ for `Q-NPG-SGD`. If for all $\theta \in \mathbb{R}^m$, the covariance matrix of the feature map followed by the initial state-action distribution $\nu$ satisfies*

$$\mathbb{E}_{(s,a)\sim\nu}\left[\phi_{s,a}\phi_{s,a}^\top\right] \overset{(23)}{=} \Sigma_\nu \geq \mu\mathbf{I}_m, \tag{31}$$

*where $\mathbf{I}_m \in \mathbb{R}^{m\times m}$ is the identity matrix and $\mu > 0$, then*

$$\mathbb{E}[V_\rho(\pi^{(K)})] - V_\rho(\pi^*) \leq \left(1 - \frac{1}{\vartheta_\rho}\right)^K \frac{2}{1-\gamma} + \mathcal{O}\left(\sqrt{\epsilon_{\mathrm{approx}}}/(1-\gamma)\right) + \mathcal{O}\left(\sqrt{m}/\left((1-\gamma)^3\sqrt{T}\right)\right).$$

In `Q-NPG-SGD`, each trajectory has the expected length $1/(1-\gamma)$ (see Lemma 4). Consequently, with $K = \mathcal{O}(\log(1/\epsilon)\log(1/(1-\gamma)))$ and $T = \mathcal{O}\left(\frac{1}{(1-\gamma)^6\epsilon^2}\right)$, Q-NPG requires $K * T/(1-\gamma) = \tilde{\mathcal{O}}\left(\frac{1}{(1-\gamma)^7\epsilon^2}\right)$ samples such that $\mathbb{E}\left[V_\rho(\pi^{(K)})\right] - V_\rho(\pi^*) \leq \mathcal{O}(\epsilon) + \mathcal{O}\left(\frac{\sqrt{\epsilon_{\mathrm{approx}}}}{1-\gamma}\right)$. The $\tilde{\mathcal{O}}(1/\epsilon^2)$ sample complexity matches with the one of value-based algorithms such as Q-learning (Li et al., 2020) .

Compared to Agarwal et al. (2021, Corollary 26) for the sampled based Q-NPG Algorithm 2, their sample complexity is $\mathcal{O}\left(\frac{1}{(1-\gamma)^{11}\epsilon^6}\right)$ with $K = \frac{1}{(1-\gamma)^2\epsilon^2}$ and $T = \frac{1}{(1-\gamma)^8\epsilon^4}$. Despite the improvement on the convergence rate for $K$, they use the optimization results of Shalev-Shwartz & Ben-David (2014, Theorem 14.8) to obtain $\epsilon_{\mathrm{stat}} = \mathcal{O}(1/\sqrt{T})$, while we use the one of Bach & Moulines (2013, Theorem 1) (see Theorem 12 as well) to establish faster $\epsilon_{\mathrm{stat}} = \mathcal{O}(1/T)$. Besides, they use the projected SGD method and require that the stochastic gradient is bounded which is incorrectly verified in their proof [3]. In contrast, to apply Theorem 12, the boundedness of the stochastic gradient is not necessary. Alternatively, we require a different condition (31). A proof sketch of our corollary and a discussion of the condition (31) are provided in Appendix D.5 for more details.

---

[3] Indeed, the stochastic gradient of $L_Q$ is unbounded, since the estimate $\widehat{Q}_{s,a}(\theta)$ of $Q_{s,a}(\theta)$ is unbounded. This is because each single sampled trajectory has unbounded length. See Appendix D.5 for more explanations.

## 5 ANALYSIS OF NPG WITH LOG-LINEAR POLICIES

We now return to the convergence analysis of the inexact NPG method, specifically,

$$\theta^{(k+1)} \ = \ \theta^{(k)} - \eta_k w^{(k)}, \qquad w^{(k)} \approx \operatorname{argmin}_w L_A\big(w, \theta^{(k)}, \tilde{d}^{(k)}\big). \tag{32}$$

Again, let $w_\star^{(k)} \in \operatorname{argmin}_w L_A\big(w, \theta^{(k)}, \tilde{d}^{(k)}\big)$ denote the minimizer. Our analysis of NPG is analogous to that of Q-NPG shown in the previous section. That is, we again exploit the inexact PMD formulation (17) and use techniques developed in Xiao (2022). The set of assumptions we use for NPG is analogous to the assumptions used in Section 4.2. In particular, we assume a bounded approximation error instead of transfer error (c.f., Assumption 2) in minimizing $L_A$ and do not need the assumption on relative condition number.

**Assumption 7** (Bounded statistical error, Assumption 6.5.1 in Agarwal et al. (2021)). *There exists* $\epsilon_{\text{stat}} > 0$ *such that for all iterations* $k \geq 0$ *of the NPG method* (32), *we have*

$$\mathbb{E}\big[L_A\big(w^{(k)}, \theta^{(k)}, \tilde{d}^{(k)}\big) - L_A\big(w_\star^{(k)}, \theta^{(k)}, \tilde{d}^{(k)}\big)\big] \ \leq \ \epsilon_{\text{stat}}. \tag{33}$$

**Assumption 8** (Bounded approximation error). *There exists* $\epsilon_{\text{approx}} > 0$ *such that for all iterations* $k \geq 0$ *of the NPG method* (32), *we have*

$$\mathbb{E}\big[L_A\big(w_\star^{(k)}, \theta^{(k)}, \tilde{d}^{(k)}\big)\big] \ \leq \ \epsilon_{\text{approx}}. \tag{34}$$

**Assumption 9** (Concentrability coefficient for state-action visitation). *There exists* $C_\nu < \infty$ *such that for all iterations* $k \geq 0$ *of the NPG method* (32), *we have*

$$\mathbb{E}_{(s,a)\sim\tilde{d}^{(k)}}\left[\left(\frac{\bar{d}_{s,a}^{(k+1)}}{\tilde{d}_{s,a}^{(k)}}\right)^2\right] \ \leq \ C_\nu \qquad and \qquad \mathbb{E}_{(s,a)\sim\tilde{d}^{(k)}}\left[\left(\frac{\bar{d}_{s,a}^{\pi^*}}{\tilde{d}_{s,a}^{(k)}}\right)^2\right] \ \leq \ C_\nu. \tag{35}$$

Under the above assumptions, we have the following result.

**Theorem 4.** *Fix a state distribution* $\rho$, *a state-action distribution* $\nu$, *and a comparator policy* $\pi^*$. *We consider the NPG method* (32) *with uniform initial policy and step sizes satisfying* $\eta_0 \geq \frac{1-\gamma}{\gamma}\log|\mathcal{A}|$ *and* $\eta_{k+1} \geq \frac{1}{\gamma}\eta_k$. *Suppose that Assumptions 7, 8 and 9 hold. Then we have for all* $k \geq 0$,

$$\mathbb{E}[V_\rho(\pi^{(k)})] - V_\rho(\pi^*) \leq \left(1 - \frac{1}{\vartheta_\rho}\right)^k \frac{2}{1-\gamma} + \frac{\sqrt{C_\nu}\,(\vartheta_\rho + 1)}{1-\gamma}\left(\sqrt{\epsilon_{\text{stat}}} + \sqrt{\epsilon_{\text{approx}}}\right).$$

Now we compare Theorem 4 with Theorem 29 in Agarwal et al. (2021) for the NPG analysis. The main differences are similar to those for Q-NPG as summarized right after Theorem 1: Their convergence rate is sublinear while ours is linear; they assume uniformly bounded $\phi_{s,a}$ and $w^{(k)}$ while we require bounded concentrability coefficient $C_\nu$ due to different proof techniques; they use diminishing step sizes and we use geometrically increasing ones. Moreover, Theorem 4 requires bounded approximation error, which is a stronger assumption than the bounded transfer error used by their Theorem 29, but we do not need the assumption on bounded relative condition number. In particular, such bounded relative condition number must hold for the covariance matrix of $\bar{\phi}_{s,a}(\theta^{(k)})$ for all $k \geq 0$ which depends on the iterates $\theta^{(k)}$. This is in contrast to our Assumption 3, where we use a single fixed covariance matrix for Q-NPG that is independent to the iterates, as defined in (23).

In addition, the inequalities in (35) only involve half of the state-action visitation distributions listed in (29), i.e., the first and the fourth terms. Thus, $C_\nu$ in (35) can share the same upper bound in (30) independent to the use of the algorithm Q-NPG or NPG. Consequently, our concentrability coefficient $C_\nu$ in (35) is weaker than Assumption 2 in Cayci et al. (2021) which studies the linear convergence of NPG with entropy regularization for the log-linear policy class. The reason is that the bound on $C_\nu$ in (30) does not depend on the policies throughout the iterations thanks to the use of $\tilde{d}^{(k)}$ instead of $\bar{d}^{(k)}$ (see Remark 1 as well). See Appendix H for a thorough discussion on the concentrability coefficient $C_\nu$.

As a by product, we also obtain a sublinear rate for NPG while using an unconstrained constant step size in Theorem 9 in Appendix F. By further assuming that the feature map is bounded and the Fisher information matrix (9) is non-singular, we also provide a $\tilde{\mathcal{O}}(1/\epsilon^2)$ sample complexity result of NPG as for Q-NPG and fix the errors occurred in the NPG sample complexity proofs of Agarwal et al. (2021) and Liu et al. (2020) in Appendix G.

ACKNOWLEDGMENTS

We gratefully acknowledge Daniel Russo who pointed out that we did not cite properly Bhandari & Russo (2021) in the literature review in the previous version.

We acknowledge the helpful discussion with Yanli Liu on the sample complexity analysis of both Q-NPG and NPG.

We would also like to thank the anonymous reviewers for their helpful comments.

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

# Appendix

## Table of Contents

Here we provide the related work discussion, the missing proofs from the main paper and some additional noteworthy observations made in the main paper.

## A  RELATED WORK

We provide an extended discussion for the context of our work, including a discussion comparing the technical novelty of the paper to the analysis of NPG with softmax tabular policies in Xiao (2022) and a comparison of the convergence theories of NPG in the literature. Furthermore, we discuss future work, such as extending our analysis to other RL settings.

### A.1  TECHNICAL CONTRIBUTION AND NOVELTY COMPARED TO XIAO (2022)

Our technical novelty compared to Xiao (2022) is summarized as follows.

- Our linear convergence results (i.e., Theorem 1, 3 and 4) are not direct applications of Theorem 10 in Xiao (2022). Indeed, Xiao (2022) establishes the connection between NPG and a specific form of policy mirror descent (PMD) with the use of the weighted Bregman divergence for the tabular setting, while we show that this connection can also be established for the function approximation setting via the compatible function approximation framework (10). We also modify the PMD framework of Xiao (2022) with the linear approximation of the advantage function in (17), inspired from the compatible function approximation framework. Thus, the approaches of deriving the PMD form update are different. Without this work of using the compatible function approximation framework to bridge NPG and PMD, it was not clear at all that the analysis of Xiao (2022) could be extended in the log-linear policy setting. So our work is the first step of showing that the proof techniques used in Xiao (2022) can be extended in function approximation regime. In fact, the extension is highly nontrivial and requires significant innovation (see details below). As for future work, one can extend our work to other function approximation setting through a similar compatible function approximation framework. See Appendix A.3 for more details about the future work.

- Besides, our linear convergence results only consider the inexact NPG update. Compared to Theorem 14 in Xiao (2022), which is their corresponding result on the inexact PMD method, we improve their analysis by making much weaker assumptions on the accuracy of the estimation $Q(\pi)$. Xiao (2022) requires an $L_\infty$ supremum norm bound on the estimation error of $Q$, i.e., $\|\widehat{Q}(\pi) - Q(\pi)\|_\infty \leq \epsilon_{\text{stat}}$, whereas our convergence guarantee depends on the expected $L_2$ error of the estimate, i.e., Assumption 1 and 7. For instance, Assumption 1 from equation (62) can be written as $\mathbb{E}\left[(\phi_{s,a}^\top w^{(k)} - \phi_{s,a}^\top w_\star^{(k)})^2\right] \leq \epsilon_{\text{stat}}$, which can be interpreted as $\mathbb{E}\left[(\widehat{Q}(\pi) - Q(\pi))^2\right] \leq \epsilon_{\text{stat}}$ under the linear approximation setting. The techniques for handling $L_\infty$ and $L_2$ errors are very different. Not only our assumption is weaker, it also benefits from the sample complexity analysis that we explain next.

- Consequently, when considering the sample complexity results we derived for sample-based (Q)-NPG in Corollary 1 and 3, the difference between our work and Theorem 16 in Xiao (2022), which corresponds to their sample complexity results, is even more significant. Corollary 1 with Algorithm `Q-NPG-SGD` (Algorithm 6) satisfies Assumption 1 with a number of samples that depends only on the feature dimension $m$ of $\phi$ and does not depend on the cardinality of state space $|\mathcal{S}|$ or action space $|\mathcal{A}|$. In contrast, the assumption $\|\widehat{Q}(\pi) - Q(\pi)\|_\infty \leq \epsilon_{\text{stat}}$ with the $L_\infty$ norm in Xiao (2022, Theorem 16) causes the sample complexity to depend on $|\mathcal{S}||\mathcal{A}|$.

  Furthermore, Xiao (2022) uses a Monte-Carlo approach with multiple independent rollouts per iteration, while our sample-based (Q)-NPG uses one single rollout (Algorithm 3 and 4) combined with regression solvers; Xiao (2022) derives a high probability sample complexity result, while we derive the convergence of the optimality gap $\mathbb{E}\left[V_\rho(\pi^{(K)})\right] - V_\rho(\pi^*)$ which can guarantee that the variance of $V_\rho(\pi^{(K)})$ converges to zero. Thus, our sample-based algorithms had not been considered in Xiao (2022) and our proofs of Corollary 1 and 3 require a different approach.

  In particular, our sample complexity analysis regarding to the policy evaluation is novel. Although our sample-based algorithms had been considered previously in Agarwal et al. (2021) and Liu et al. (2020), none of their analysis on the sample complexity was correct. Indeed, Agarwal et al. (2021) required the boundedness of the stochastic gradient estimator, which might not hold as we extensively discussed in Appendix D.5. We fixed this by showing that $\mathbb{E}\left[\widehat{Q}_{s,a}(\theta)^2\right]$ is bounded. See Appendix D.5 for all the subtleties, including a proof sketch of Corollary 1. Liu et al. (2020) also incorrectly used an inequality where the random variables are correlated. See the detailed explanation (Footnote 7) in Appendix G.1. We fixed this error with a careful conditional expectation argument. Please refer to Appendix G.1 for all the details, including a proof sketch of Corollary 3. These dimensions are where an important part of the technical work was done. Therefore, outside of the tabular setting, and considering NPG methods that make use of a regression solver, our complexity analysis is currently the only analysis that is entirely correct that we are aware of.

- Finally we not only extend the work of Xiao (2022) to NPG for log-linear policy, but also consider the Q-NPG method and establish its linear convergence analysis. This is a method that is unique to log-linear policy and again had not been considered in Xiao (2022).

## A.2 FINITE-TIME ANALYSIS OF THE NATURAL POLICY GRADIENT

**NPG for the softmax tabular policies.** For the softmax tabular policies, Shani et al. (2020) show that the unregularized NPG has a $\mathcal{O}(1/\sqrt{k})$ convergence rate and the regularized NPG has a faster $\mathcal{O}(1/k)$ convergence rate by using a decaying step size. Agarwal et al. (2021) improve the convergence rate of the unregularized NPG to $\mathcal{O}(1/k)$ with constant step sizes. Further, Khodadadian et al. (2021a) also achieves $\mathcal{O}(1/k)$ convergence rate for the off-policy natural actor-critic (NAC), and a slower sublinear result is established by Khodadadian et al. (2022a) for the two-time-scale NAC.

By using the entropy regularization, Cen et al. (2021) achieve a linear convergence rate for NPG. A similar linear convergence result has been obtained by rewriting the NPG update under the PMD framework with the Kullback–Leibler (KL) divergence (Lan, 2022) or with a more general convex regularizer (Zhan et al., 2021). Such approach is also applied in the averaged MDP setting to achieve linear convergence for NPG (Li et al., 2022a). However, adding regularization might induce bias

for the solution. Thus, Lan (2022) considers exponentially diminishing regularization to guarantee unbiased solution. Furthermore, by considering both the KL divergence and the diminishing entropy regularization, Li et al. (2022b) establish the linear convergence rate not only for the optimality gap but also for the policy. That is, the policy will converge to the fixed high entropy optimal policy. Consequently, Li et al. (2022b) show a local super-linear convergence of both the policy and optimality gap, as discussed in Xiao (2022, Section 4.3).

Recently, Bhandari & Russo (2021), Khodadadian et al. (2021b; 2022b) and Xiao (2022) show that regularization is unnecessary for obtaining linear convergence, and it suffices to use appropriate step sizes for NPG. In particular, Bhandari & Russo (2021) propose to use an exact line search for the step size (Theorem 1 (a)) or to choose an adaptive step size (Theorem 1 (c)). Similar adaptive step size is proposed by Khodadadian et al. (2021b; 2022b). Notice that such adaptive step size requires complete knowledge about the environmental model. Instead, a sufficiently large step size might be enough. In this paper, we extend the results of Xiao (2022) from the tabular setting to the log-linear policies, using *non-adaptive* geometrically increasing step size and obtaining a linear convergence rate for NPG without regularization.

**NPG with function approximation.** In the function approximation regime, there have been many works investigating the convergence rate of the NPG or NAC algorithms from different perspectives. Wang et al. (2020) establish the $\mathcal{O}(1/\sqrt{k})$ convergence rate for two-layer neural NAC with a projection step. The sublinear convergence results are also established by Zanette et al. (2021) and Hu et al. (2022) for the linear MDP (Jin et al., 2020). Agarwal et al. (2021) obtain the same $\mathcal{O}(1/\sqrt{k})$ convergence rate for both projected NPG with smooth policies and projected Q-NPG with log-linear policies. This was later improved to $\mathcal{O}(1/k)$ by Liu et al. (2020) by replacing the projection step with a strong regularity condition on the Fisher information matrix, and it was also improved to $\mathcal{O}(1/k)$ by Xu et al. (2020) with NAC under Markovian sampling. The same $\mathcal{O}(1/k)$ convergence rate is established for log-linear policies by Chen et al. (2022) when considering the off-policy NAC.

With entropy regularization and a projection step, Cayci et al. (2021) obtain a linear convergence for log-linear policies. Same entropy regularization and a projection step are applied by Cayci et al. (2022) for the neural NAC to improve the $\mathcal{O}(1/\sqrt{k})$ convergence rate of Wang et al. (2020) to $\mathcal{O}(1/k)$. In contrast, we show that by using a simple geometrically increasing step size, fast linear convergence can be achieved for log-linear policies without any additional regularization nor a projection step. We notice that Chen & Theja Maguluri (2022, Theorem 3.4)[4] also uses increasing step size and achieves linear convergence for log-linear policies without regularization. The main differences between our result and Theorem 3.4 in Chen & Theja Maguluri (2022) are fourfold. First, they rely on the contraction property of the generalized Bellman operator, while we consider the PMD analysis approach. So the proof techniques are completely different. Second, their parameter update results in the off-policy multi-step temporal difference learning, whereas we require to solve a linear regression problem to minimize the function approximation error. Third, their step size still depends on the iterates which is thus an adaptive step size and is proportional to the total number of iterations $K$, while ours is independent to the iterates nor to $K$. Finally, their assumption on the modeling error requires an $L_\infty$ supremum norm, i.e., $\|Q_s(\theta^{(k)}) - \Phi w_\star^{(k)}\|_\infty \leq \epsilon_{\text{bias}}$ for all states $s$ of the state space, our convergence guarantee depends on the expected error (e.g., Assumption 2, 5 or 8) which is a much weaker assumption. After our submission, we are aware of the concurrent work of Alfano & Rebeschini (2022). They only analyze the Q-NPG method and achieve similar linear convergence results as our Theorem 1. In particular, their result in Theorem 4.7 has a better concentrability coefficient compared to our Theorem 1. However, their Assumption 4.6 assumes that the relative condition number upper bounds a time-varying ratio which depends on the iterates, while our Assumption 3 is independent to the iterates, as defined in (24). Furthermore, they only consider the case when the initial state distribution is the same as the target state distribution, while our analysis generalizes with any target state distribution, which is extensively discussed on the distribution mismatch coefficients in Appendix H. See Table 1 a complete overview of NPG in the function approximation regime.

---

[4]This result appears after conference proceedings and is available on https://arxiv.org/pdf/2208.03247.pdf.

Table 1: Overview of different convergence results for NPG methods in the function approximation regime. The darker cells contain our new results. The light cells contain previously known results for NPG or Q-NPG with log-linear policies that we have a direct comparison to our new results. White cells contain existing results that do not have the same setting as ours, so that we could not make a direct comparison among them.

| Setting | Rate | Reg. | C.S. | I.S.[*] | Pros/cons compared to our work |
|---|---|---|---|---|---|
| Linear convergence | | | | | |
| Regularized NPG with log-linear (Cayci et al., 2021) | Linear | ✓ | ✓ | | Better concentrability coefficients $C_\nu$ |
| Off-policy NAC with log-linear (Chen & Theja Maguluri, 2022) | Linear | | | ✓ | Weaker assumptions on the approximation error with $L_2$ norm instead of $L_\infty$ norm; They use adaptive increasing stepsize, while we use non-adaptive increasing stepsize |
| Q-NPG with log-linear (Alfano & Rebeschini, 2022) | Linear | | | ✓ | Their relative condition number depends on $t$, while ours is independent to $t$ |
| Q-NPG/NPG with log-linear (this work) | Linear | | | ✓ | |
| Sublinear convergence | | | | | |
| PMD for linear MDP (Zanette et al., 2021; Hu et al., 2022) | $\mathcal{O}(\frac{1}{\sqrt{k}})$ | | ✓ | | |
| Two-layer neural NAC (Wang et al., 2020) | $\mathcal{O}(\frac{1}{\sqrt{k}})$ | | ✓ | | |
| Two-layer neural NAC (Cayci et al., 2022) | $\mathcal{O}(\frac{1}{k})$ | ✓ | ✓ | | |
| NPG with smooth policies (Agarwal et al., 2021) | $\mathcal{O}(\frac{1}{\sqrt{k}})$ | | ✓ | | |
| NAC under Markovian sampling with smooth policies (Xu et al., 2020) | $\mathcal{O}(\frac{1}{k})$ | | ✓ | | |
| NPG with smooth and Fisher-non-degenerate policies (Liu et al., 2020) | $\mathcal{O}(\frac{1}{k})$ | | ✓ | | |
| Q-NPG with log-linear (Agarwal et al., 2021) | $\mathcal{O}(\frac{1}{\sqrt{k}})$ | | ✓ | | They have better error floor than ours |
| Off-policy NAC with log-linear (Chen et al., 2022) | $\mathcal{O}(\frac{1}{k})$ | | | ✓ | Weaker assumptions on the approximation error with $L_2$ norm instead of $L_\infty$ norm; They use adaptive increasing stepsize, while we use non-adaptive increasing stepsize |
| Q-NPG/NPG with log-linear (this work) | $\mathcal{O}(\frac{1}{k})$ | | | ✓ | |

[*] **Reg.**: regularization; **C.S.**: constant stepsize; **I.S.**: increasing stepsize.

**Fast linear convergence of other policy gradient methods.** Different to the PMD analysis approach, by leveraging a gradient dominance property (Polyak, 1963; Łojasiewicz, 1963), fast linear convergence results have also been established for the PG methods under different settings, such as the linear quadratic control problems (Fazel et al., 2018) and the exact PG method with softmax tabular policy and entropy regularization (Mei et al., 2020; Yuan et al., 2022). Such gradient domination property is widely explored by Bhandari & Russo (2019) to identify more general structural MDP settings. Linear convergence of PG can also be obtained through exact line search (Bhandari & Russo, 2021, Theorem 1 (a)) or by exploiting non-uniform smoothness (Mei et al., 2021).

Alternatively, by considering a general strongly-concave utility function of the state-action occupancy measure and by exploiting the hidden convexity of the problem, Zhang et al. (2020) also achieve the linear convergence of a variational PG method. When the object is relaxed to a general concave utility function, Zhang et al. (2021) still achieve the linear convergence by leveraging the hidden convexity of the problem and by adding variance reduction to the PG method.

### A.3 FUTURE WORK

The main focus of this paper was the theoretical analysis of NPG method. The results we have obtained open up several experimental questions related to parameter settings for NPG and Q-NPG. We leave such questions as an important future work to further support our theoretical findings.

An interesting application from our work is to investigate the sample complexity of natural actor-critic with our PMD analysis. Indeed, our paper obtains $w^{(k)}$ by a regression solver. One can also use temporal difference (TD) learning (e.g., Cayci et al. (2021); Chen & Theja Maguluri (2022); Telgarsky (2022)) with Markovian sampling to achieve similar $O(1/\epsilon^2)$ sample complexity result. The performance analysis of TD learning will be expressed for $\epsilon_{\mathrm{stat}}$, which directly imply the total sample complexity results through our theorems.

One natural question is whether we can extend our analysis to the general policy classes. Here we provide one possible way. It can be extended by using a similar compatible function approximation framework. Concretely, consider the parameterized policy

$$\pi_{s,a}(\theta) = \frac{\exp(f_{s,a}(\theta))}{\sum_{a' \in \mathcal{A}} \exp(f_{s,a'}(\theta))},$$

where $f_{s,a}(\theta)$ is parameterized by $\theta \in \mathbb{R}^m$ and is differential. As Agarwal et al. (2021) mentioned, the gradient can be written as

$$\nabla_\theta \log \pi_{s,a}(\theta) = g_{s,a}(\theta) \quad \text{where} \quad g_{s,a}(\theta) = \nabla_\theta f_{s,a}(\theta) - \mathbb{E}_{a' \sim \pi_s(\theta)}\left[\nabla_\theta f_{s,a'}(\theta)\right].$$

The NPG update is equivalent to the following compatible function approximation framework

$$\theta^{(k+1)} = \theta^{(k)} - \eta_k w_\star^{(k)}, \qquad w_\star^{(k)} \in \arg\min_w \mathbb{E}_{(s,a) \sim \bar{d}^{(k)}}\left[\left(A_{s,a}(\theta^{(k)}) - w^\top g_{s,a}(\theta^{(k)})\right)^2\right].$$

As Alfano & Rebeschini (2022, Remark 4.8) mentioned, if we assume that for all $(s,a) \in \mathcal{S} \times \mathcal{A}$, function $f(\theta)$ satisfies

$$f_{s,a}(\theta^{(k+1)}) = f_{s,a}(\theta^{(k)}) - \eta_k (w_\star^{(k)})^\top g_{s,a}(\theta^{(k)}),$$

which is the case for the log-linear policies, then one can easily verify that the NPG update resulted in a new policy is also equivalent to the policy mirror descent update

$$\pi_s^{(k+1)} = \arg\min_{p \in \Delta(\mathcal{A})}\left\{\eta_k \left\langle G_s^{(k)} w^{(k)}, p\right\rangle + D(p, \pi_s^{(k)})\right\}, \quad \forall s \in \mathcal{S},$$

where $G_s^{(k)} \in \mathbb{R}^{|\mathcal{A}| \times m}$ is a matrix with rows $(g_{s,a}(\theta^{(k)}))^\top \in \mathbb{R}^{1 \times m}$ for $a \in \mathcal{A}$. Consequently, one can extend our work naturally in this general setting to derive linear convergence analysis for NPG.

Perhaps one can consider the *exponential tilting*, a generalization of Softmax to more general probability distributions. Another interesting venue of investigation is to consider the *generalized linear model* instead of linear function approximation for the $Q$ function and the advantage function.

One interesting open question is that is there a way to increase stepsize when the discount factor is unknown. So far the PMD proof techniques used in Lan (2022); Xiao (2022) and ours require that the discount factor is known. Perhaps the work of Li et al. (2022a) can help to find a way to increase stepsize when the discount factor is unknown. Indeed, Li et al. (2022a) consider the averaged MDP setting. So there is no discount factor. They achieve linear convergence for NPG by increasing the stepsize with some regularization parameters. It will be interesting to investigate if the way of increasing stepsize in Li et al. (2022a) can be applied in our setting.

## B STANDARD REINFORCEMENT LEARNING RESULTS

In this section, we prove the standard reinforcement learning results used in our main paper, including the NPG updates written through the compatible function approximation (11) and the NPG updates formalized as policy mirror descent ((16) and (17)). Then, we prove the performance difference lemma (Kakade & Langford, 2002), which is the first key ingredient for our PMD analysis. The three-point descent lemma (Lemma 11) is the second key ingredient for our PMD analysis.

**Lemma 1** (NPG updates via compatible function approximation, Theorem 1 in Kakade (2001))**.** *Consider the NPG updates* (8)

$$\theta^{(k+1)} \;=\; \theta^{(k)} - \eta_k F_\rho\big(\theta^{(k)}\big)^\dagger \nabla_\theta V_\rho\big(\theta^{(k)}\big),$$

*and the updates using the compatible function approximation* (11)

$$\theta^{(k+1)} \;=\; \theta^{(k)} - \eta_k w_\star^{(k)},$$

*where $w_\star^{(k)} \in \arg\min_{w\in\mathbb{R}^m} L_A\big(w, \theta^{(k)}, \bar{d}^{(k)}\big)$. If the parametrized policy is differentiable for all $\theta \in \mathbb{R}^m$, then the two updates are equivalent up to a constant scaling $(1-\gamma)$ of $\eta_k$.*

*Proof.* Indeed, using the policy gradient (7) and the fact that $\sum_{a\in\mathcal{A}} \nabla \pi_{s,a}(\theta) = 0$ for all $s \in \mathcal{S}$, as $\pi(\theta)$ is differentiable on $\theta$ and $\sum_{a\in\mathcal{A}} \pi_{s,a} = 1$, we have the policy gradient theorem (Sutton et al., 2000)

$$\nabla_\theta V_\rho(\theta) = \frac{1}{1-\gamma}\mathbb{E}_{s\sim d^\theta,\, a\sim\pi_s(\theta)}\left[ A_{s,a}(\theta)\, \nabla_\theta \log \pi_{s,a}(\theta) \right]. \tag{36}$$

Furthermore, consider the optima $w_\star^{(k)}$. By the first-order optimality condition, we have

$$\nabla_w L_A(w_\star^{(k)}, \theta^{(k)}, \bar{d}^{(k)}) = 0$$

$$\Longleftrightarrow \quad \mathbb{E}_{(s,a)\sim\bar{d}^{(k)}}\left[ \left( (w_\star^{(k)})^\top \nabla_\theta \log \pi_{s,a}^{(k)} - A_{s,a}(\theta^{(k)}) \right) \nabla_\theta \log \pi_{s,a}^{(k)} \right] = 0$$

$$\Longleftrightarrow \quad \mathbb{E}_{(s,a)\sim\bar{d}^{(k)}}\left[ \nabla_\theta \log \pi_{s,a}^{(k)} \left( \nabla_\theta \log \pi_{s,a}^{(k)} \right)^\top \right] w_\star^{(k)} = \mathbb{E}_{(s,a)\sim\bar{d}^{(k)}}\left[ A_{s,a}(\theta^{(k)}) \nabla_\theta \log \pi_{s,a}^{(k)} \right]$$

$$\overset{(8)+(36)}{\Longleftrightarrow} \quad F_\rho(\theta^{(k)}) w_\star^{(k)} = (1-\gamma)\nabla_\theta V_\rho(\theta^{(k)}).$$

Thus, we have

$$w_\star^{(k)} = (1-\gamma)F_\rho(\theta)^\dagger \nabla_\theta V_\rho(\theta^{(k)})$$

which yields the update (8) up to a constant scaling $(1-\gamma)$ of $\eta_k$. $\qquad\square$

**Lemma 2** (NPG updates as policy mirror descent)**.** *The closed form solution to* (16) *is given by*

$$\pi_s^{(k+1)} \;=\; \pi_s^{(k)} \odot \frac{\exp\big(-\eta_k \Phi_s w^{(k)}\big)}{\sum_{a\in\mathcal{A}} \pi_{s,a}^{(k)} \exp\big(-\eta_k \phi_{s,a}^\top w^{(k)}\big)} \tag{37}$$

$$= \pi_s^{(k)} \odot \frac{\exp\left(-\eta_k \bar{\Phi}_s^{(k)} w^{(k)}\right)}{\sum_{a\in\mathcal{A}} \pi_{s,a}^{(k)} \exp\left(-\eta_k \big(\bar{\phi}_{s,a}(\theta^{(k)})\big)^\top w^{(k)}\right)} \tag{38}$$

$$= \arg\min_{p\in\Delta(\mathcal{A})} \left\{ \eta_k \left\langle \bar{\Phi}_s^{(k)} w^{(k)}, p \right\rangle + D(p, \pi_s^{(k)}) \right\}, \quad \forall s \in \mathcal{S}, \tag{39}$$

*where $\odot$ is the element-wise product between vectors, and $\bar{\Phi}_s^{(k)} \in \mathbb{R}^{|\mathcal{A}|\times m}$ is defined in* (17)*, i.e.*

$$\left(\bar{\Phi}_{s,a}^{(k)}\right)^\top \overset{def}{=} \bar{\phi}_{s,a}(\theta^{(k)}) \overset{(12)}{=} \phi_{s,a} - \mathbb{E}_{a'\sim\pi_s^{(k)}}\left[ \phi_{s,a'} \right].$$

*Such policy update coincides the inexact NPG updates* (32) *of the log-linear policy, if $\theta^{(k+1)} = \theta^{(k)} - \eta_k w^{(k)}$ with $w^{(k)} \approx \arg\min_w L_A(w, \theta^{(k)}, \tilde{d}^{(k)})$; and coincides the inexact Q-NPG updates* (18) *of the log-linear policy, if $\theta^{(k+1)} = \theta^{(k)} - \eta_k w^{(k)}$ with $w^{(k)} \approx \arg\min_w L_Q(w, \theta^{(k)}, \tilde{d}^{(k)})$.*

*Proof.* For shorthand, let $g = \Phi_s w^{(k)}$. Thus, (16) fits the format of Lemma 10 in Appendix I where $q = \pi_s^{(k)}$. Consequently, the closed form solution is given by (99), that is

$$\pi_s^{(k+1)} \;=\; \frac{\pi_s^{(k)} \odot e^{-\eta_k g}}{\sum_{a\in\mathcal{A}} \pi_{s,a}^{(k)} e^{-\eta_k g_a}} \;=\; \frac{\pi_s^{(k)} \odot e^{-\eta_k \Phi_s w^{(k)}}}{\sum_{a\in\mathcal{A}} \pi_{s,a}^{(k)} e^{-\eta_k \phi_{s,a}^\top w^{(k)}}}$$

$$= \pi_s^{(k)} \odot \frac{\exp\big(-\eta_k \bar{\Phi}_s(\theta^{(k)}) w^{(k)}\big)}{\sum_{a\in\mathcal{A}} \pi_{s,a}^{(k)} \exp\left(-\eta_k \big(\bar{\phi}_{s,a}(\theta^{(k)})\big)^\top w^{(k)}\right)}, \tag{40}$$

where the last equality is obtained as

$$\bar{\phi}_{s,a}(\theta^{(k)}) = \phi_{s,a} - \mathbb{E}_{a' \sim \pi_s^{(k)}}[\phi_{s,a'}] = \phi_{s,a} - c_s,$$

with $c_s \in \mathbb{R}$ some constant independent to $a$.

Similarly, by applying Lemma 10 with $g = \bar{\Phi}_s^{(k)} w^{(k)}$, the closed form solution to (39) is (40).

As for the closed form updates of the policy for NPG (32) and Q-NPG (18) with the parameter updates $\theta^{(k+1)} = \theta^{(k)} - \eta_k w^{(k)}$, it is straightforward to verify that it coincides (37) and (38) given the specific structure of the log-linear policy (6), which concludes the proof. □

**Lemma 3** (Performance difference lemma (Kakade & Langford, 2002)). *For any policy $\pi, \pi' \in \Delta(\mathcal{A})^{\mathcal{S}}$ and $\rho \in \Delta(\mathcal{S})$,*

$$V_\rho(\pi) - V_\rho(\pi') = \frac{1}{1-\gamma}\mathbb{E}_{(s,a) \sim \bar{d}^\pi}[A_{s,a}(\pi')] \tag{41}$$

$$= \frac{1}{1-\gamma}\mathbb{E}_{s \sim d^\pi}[\langle Q_s(\pi'), \pi_s - \pi'_s \rangle], \tag{42}$$

*where $Q_s(\pi)$ is the shorthand for $[Q_{s,a}(\pi)]_{a \in \mathcal{A}} \in \mathbb{R}^{|\mathcal{A}|}$ for any policy $\pi$.*

*Proof.* From Lemma 2 in Agarwal et al. (2021), we have

$$V_\rho(\pi) - V_\rho(\pi') = \frac{1}{1-\gamma}\mathbb{E}_{(s,a) \sim \bar{d}^\pi}[A_{s,a}(\pi')] = \frac{1}{1-\gamma}\mathbb{E}_{s \sim d^\pi}[\langle A_s(\pi'), \pi_s \rangle],$$

where $A_s(\pi)$ is the shorthand for $[A_{s,a}(\pi)]_{a \in \mathcal{A}} \in \mathbb{R}^{|\mathcal{A}|}$ for any policy $\pi$. To show (42), it suffices to show

$$\langle A_s(\pi'), \pi_s \rangle = \langle Q_s(\pi'), \pi_s - \pi'_s \rangle, \quad \text{for all } s \in \mathcal{S} \text{ and } \pi, \pi' \in \Delta(\mathcal{A})^{\mathcal{S}}.$$

Let $\mathbf{1}_n$ denote a vector in $\mathbb{R}^n$ with coordinates equal to 1 element-wisely. Indeed, using the definition of the advantage function

$$A_{s,a}(\pi) = Q_{s,a}(\pi) - V_s(\pi),$$

we have

$$\begin{aligned}
\langle A_s(\pi'), \pi_s \rangle &= \langle Q_s(\pi') - V_s(\pi') \cdot \mathbf{1}_{|\mathcal{A}|}, \pi_s \rangle \\
&= \langle Q_s(\pi'), \pi_s \rangle - \langle V_s(\pi') \cdot \mathbf{1}_{|\mathcal{A}|}, \pi_s \rangle \\
&= \langle Q_s(\pi'), \pi_s \rangle - V_s(\pi') \\
&\overset{(1)}{=} \langle Q_s(\pi'), \pi_s - \pi'_s \rangle,
\end{aligned}$$

from which we conclude the proof. □

# C ALGORITHMS

## C.1 NPG AND Q-NPG ALGORITHM

Algorithm 1 combined with the sampling procedure (Algorithm 4) and the averaged SGD procedure, called `NPG-SGD` (Algorithm 5), provide the sample-based NPG methods.

Similarly, Algorithm 2 combined with the sampling procedure (Algorithm 3) and the averaged SGD procedure, called `Q-NPG-SGD` (Algorithm 6), provide the sample-based Q-NPG methods.

## C.2 SAMPLING PROCEDURES

In practice, we cannot compute the true minimizer $w_\star^{(k)}$ of the regression problem in either (32) or (18), since computing the expectation $L_A$ or $L_Q$ requires averaging over all state-action pairs $(s,a) \sim \tilde{d}^{(k)}$ and averaging over all trajectories $(s_0, a_0, c_0, s_1, \cdots)$ to compute the values of $Q_{s,a}^{(k)}$ and $A_{s,a}^{(k)}$. So instead, we provide a sampler which is able to obtain unbiased estimates of $Q_{s,a}(\theta)$ (or $A_{s,a}(\theta)$) with $(s,a) \sim \tilde{d}^\theta(\nu)$ for any $\pi(\theta)$.

---

**Algorithm 1:** Natural policy gradient

---

**Input:** Initial state-action distribution $\nu$, policy $\pi^{(0)}$, discounted factor $\gamma \in [0, 1)$, step size $\eta_0 > 0$ for NPG update, step size $\alpha > 0$ for `NPG-SGD` update, number of iterations $T$ for `NPG-SGD`

1 **for** $k = 0$ **to** $K - 1$ **do**
2    Compute $w^{(k)}$ of (32) by `NPG-SGD`, i.e., Algorithm 5 with inputs $(T, \nu, \pi^{(k)}, \gamma, \alpha)$
3    Update $\theta^{(k+1)} = \theta^{(k)} - \eta_k w^{(k)}$ and $\eta_k$

**Output:** $\pi^{(K)}$

---

**Algorithm 2:** Q-Natural policy gradient

---

**Input:** Initial state-action distribution $\nu$, policy $\pi^{(0)}$, discounted factor $\gamma \in [0, 1)$, step size $\eta_0 > 0$ for **Q-NPG** update, step size $\alpha > 0$ for `Q-NPG-SGD` update, number of iterations $T$ for `Q-NPG-SGD`

1 **for** $k = 0$ **to** $K - 1$ **do**
2    Compute $w^{(k)}$ of (18) by `Q-NPG-SGD`, i.e., Algorithm 6 with inputs $(T, \nu, \pi^{(k)}, \gamma, \alpha)$
3    Update $\theta^{(k+1)} = \theta^{(k)} - \eta_k w^{(k)}$ and $\eta_k$

**Output:** $\pi_{\theta^{(K)}}$

---

**Algorithm 3:** Sampler for: $(s, a) \sim \tilde{d}^\theta(\nu)$ and unbiased estimate $\widehat{Q}_{s,a}(\theta)$ of $Q_{s,a}(\theta)$

---

**Input:** Initial state-action distribution $\nu$, policy $\pi(\theta)$, discounted factor $\gamma \in [0, 1)$
1 Initialize $(s_0, a_0) \sim \nu$, the time step $h, t = 0$, the variable $X = 1$
2 **while** $X = 1$ **do**
3    **With probability** $\gamma$**:**
4       Sample $s_{h+1} \sim \mathcal{P}(\cdot \mid s_h, a_h)$
5       Sample $a_{h+1} \sim \pi_{s_{h+1}}(\theta)$
6       $h \leftarrow h + 1$
7    **Otherwise with probability** $(1 - \gamma)$**:**
8       $X = 0$                 ▷ `Accept` $(s_h, a_h)$
9 $X = 1$
10 Set the estimate $\widehat{Q}_{s_h, a_h}(\theta) = c(s_h, a_h)$      ▷ `Start to estimate` $\widehat{Q}_{s_h, a_h}(\theta)$
11 $t = h$
12 **while** $X = 1$ **do**
13    **With probability** $\gamma$**:**
14       Sample $s_{t+1} \sim \mathcal{P}(\cdot \mid s_t, a_t)$
15       Sample $a_{t+1} \sim \pi_{s_{t+1}}(\theta)$
16       $\widehat{Q}_{s_h, a_h}(\theta) \leftarrow \widehat{Q}_{s_h, a_h}(\theta) + c(s_{t+1}, a_{t+1})$
17       $t \leftarrow t + 1$
18    **Otherwise with probability** $(1 - \gamma)$**:**
19       $X = 0$            ▷ `Accept` $\widehat{Q}_{s_h, a_h}(\theta)$

**Output:** $(s_h, a_h)$ and $\widehat{Q}_{s_h, a_h}(\theta)$

---

To solve (18), we sample $(s,a) \sim \tilde{d}^{(k)}$ and $\widehat{Q}_{s,a}^{(k)}$ by a standard rollout, formalized in Algorithm 3. This sampling procedure is commonly used, for example in Agarwal et al. (2021, Algorithm 1).

It is straightforward to verify that $(s_h, a_h)$ and $\widehat{Q}_{s_h, a_h}(\theta)$ obtained in Algorithm 3 are unbiased for any $\pi(\theta)$. The expected length of the trajectory is $\frac{1}{1-\gamma}$. We provide its proof here for completeness.

**Lemma 4.** *Consider the output* $(s_h, a_h)$ *and* $\widehat{Q}_{s_h, a_h}(\theta)$ *of Algorithm 3. It follows that*

$$\mathbb{E}\left[h+1\right] = \frac{1}{1-\gamma},$$

$$\Pr(s_h = s, a_h = a) = \tilde{d}_{s,a}^{\theta}(\nu),$$

$$\mathbb{E}\left[\widehat{Q}_{s_h, a_h}(\theta) \mid s_h, a_h\right] = Q_{s_h, a_h}(\theta).$$

*Proof.* The expected length $(h+1)$ of sampling $(s,a)$ is

$$\mathbb{E}\left[h+1\right] = \sum_{k=0}^{\infty} \Pr(h = k)(k+1) = (1-\gamma)\sum_{k=0}^{\infty} \gamma^k (k+1) = \frac{1}{1-\gamma}.$$

The probability of the state-action pair $(s,a)$ being sampled by Algorithm 3 is

$$\Pr(s_h = s, a_h = a)$$

$$= \sum_{(s_0, a_0) \in \mathcal{S} \times \mathcal{A}} \nu_{s_0, a_0} \sum_{k=0}^{\infty} \Pr(h = k) \Pr^{\pi(\theta)}(s_h = s, a_h = a \mid h = k, s_0, a_0)$$

$$= \sum_{(s_0, a_0) \in \mathcal{S} \times \mathcal{A}} \nu_{s_0, a_0} (1-\gamma) \sum_{k=0}^{\infty} \gamma^k \Pr^{\pi(\theta)}(s_k = s, a_k = a \mid s_0, a_0) \stackrel{(4)}{=} \tilde{d}_{s,a}^{\theta}(\nu).$$

Now we verify that $\widehat{Q}_{s_h, a_h}(\theta)$ obtained from Algorithm 3 is an unbiased estimate of $Q_{s_h, a_h}(\theta)$. Indeed, from Algorithm 3, we have

$$\widehat{Q}_{s_h, a_h}(\theta) = \sum_{t=0}^{H} c(s_{t+h}, a_{t+h}), \tag{43}$$

where $(H+1)$ is the length of the horizon executed between lines 13 and 19 in Algorithm 3 for calculating $\widehat{Q}_{s_h, a_h}(\theta)$. To simplify notation, we consider the estimate of $\widehat{Q}_{s,a}$ for any $(s,a) \in \mathcal{S} \times \mathcal{A}$ following the same procedure starting from line 10 in Algorithm 3. Taking expectation, we have

$$\mathbb{E}\left[\widehat{Q}_{s,a}(\theta) \mid s, a\right] = \mathbb{E}\left[\sum_{t=0}^{H} c(s_t, a_t) \mid s_0 = s, a_0 = a\right]$$

$$= \sum_{k=0}^{\infty} \Pr(H = k)\mathbb{E}\left[\sum_{t=0}^{H} c(s_t, a_t) \mid s_0 = s, a_0 = a, H = k\right]$$

$$= \sum_{k=0}^{\infty} (1-\gamma)\gamma^k \mathbb{E}\left[\sum_{t=0}^{k} c(s_t, a_t) \mid s_0 = s, a_0 = a\right]$$

$$= (1-\gamma)\mathbb{E}\left[\sum_{t=0}^{\infty} c(s_t, a_t) \sum_{k=t}^{\infty} \gamma^k \mid s_0 = s, a_0 = a\right]$$

$$= \mathbb{E}\left[\sum_{t=0}^{\infty} \gamma^k c(s_t, a_t) \mid s_0 = s, a_0 = a\right] \stackrel{(2)}{=} Q_{s,a}(\theta).$$

The desired result is obtained by setting $s = s_h$ and $a = a_h$. $\qquad \square$

Similar to Algorithm 3, to solve (32), we sample $(s,a) \sim \tilde{d}^{(k)}$ by the same procedure and estimate $\widehat{A}_{s,a}^{(k)}$ with a slight modification, namely Algorithm 4 (also see Agarwal et al., 2021, Algorithm 3).

---

**Algorithm 4:** Sampler for: $(s, a) \sim \tilde{d}^{\theta}(\nu)$ and unbiased estimate $\widehat{A}_{s,a}(\theta)$ of $A_{s,a}(\theta)$

---

**Input:** Initial state-action distribution $\nu$, policy $\pi(\theta)$, discounted factor $\gamma \in [0, 1)$

1   Initialize $(s_0, a_0) \sim \nu$, the time step $h, t = 0$, the variable $X = 1$
2   **while** $X = 1$ **do**
3     **With probability** $\gamma$:
4       Sample $s_{h+1} \sim \mathcal{P}(\cdot \mid s_h, a_h)$
5       Sample $a_{h+1} \sim \pi_{s_{h+1}}(\theta)$
6       $h \leftarrow h + 1$
7     **Otherwise with probability** $(1 - \gamma)$:
8       $X = 0$                     $\triangleright$ Accept $(s_h, a_h)$
9   $X = 1$
10   Set the estimate $\widehat{Q}_{s_h, a_h}(\theta) = c(s_h, a_h)$       $\triangleright$ Start to estimate $\widehat{Q}_{s_h, a_h}(\theta)$
11   $t = h$
12   **while** $X = 1$ **do**
13     **With probability** $\gamma$:
14       Sample $s_{t+1} \sim \mathcal{P}(\cdot \mid s_t, a_t)$
15       Sample $a_{t+1} \sim \pi_{s_{t+1}}(\theta)$
16       $\widehat{Q}_{s_h, a_h}(\theta) \leftarrow \widehat{Q}_{s_h, a_h}(\theta) + c(s_{t+1}, a_{t+1})$
17       $t \leftarrow t + 1$
18     **Otherwise with probability** $(1 - \gamma)$:
19       $X = 0$                     $\triangleright$ Accept $\widehat{Q}_{s_h, a_h}(\theta)$
20   $X = 1$
21   Set the estimate $\widehat{V}_{s_h}(\theta) = 0$            $\triangleright$ Start to estimate $\widehat{V}_{s_h}(\theta)$
22   $t = h$
23   **while** $X = 1$ **do**
24     Sample $a_t \sim \pi_{s_t}(\theta)$
25     $\widehat{V}_{s_h}(\theta) \leftarrow \widehat{V}_{s_h}(\theta) + c(s_t, a_t)$
26     **With probability** $\gamma$:
27       Sample $s_{t+1} \sim \mathcal{P}(\cdot \mid s_t, a_t)$
28       $t \leftarrow t + 1$
29     **Otherwise with probability** $(1 - \gamma)$:
30       $X = 0$                     $\triangleright$ Accept $\widehat{V}_{s_h}(\theta)$

**Output:** $(s_h, a_h)$ and $\widehat{A}_{s_h, a_h}(\theta) = \widehat{Q}_{s_h, a_h}(\theta) - \widehat{V}_{s_h}(\theta)$

---

Notice that the sampling procedure for estimating $Q_{s,a}(\theta)$ in Algorithm 3 is simpler than that for estimating $A_{s,a}(\theta)$ in Algorithm 4, since Algorithm 4 requires an additional estimation of $V_s(\theta)$ and thus doubles the number of samples to estimate $A_{s,a}(\theta)$. As in Lemma 4, we verify in the following lemma that the output $(s_h, a_h)$ is sampled from the distribution $\tilde{d}^\theta$ and $\widehat{A}_{s_h,a_h}(\theta)$ in Algorithm 4 is an unbiased estimator of $A_{s_h,a_h}(\theta)$ for all policy $\pi(\theta)$.

**Lemma 5.** *Consider the output $(s_h, a_h)$ and $\widehat{A}_{s_h,a_h}(\theta)$ of Algorithm 4. It follows that*

$$\mathbb{E}\left[h+1\right] = \frac{1}{1-\gamma},$$
$$\Pr(s_h = s, a_h = a) = \tilde{d}^\theta_{s,a}(\nu),$$
$$\mathbb{E}\left[\widehat{A}_{s_h,a_h}(\theta) \mid s_h, a_h\right] = A_{s_h,a_h}(\theta).$$

*Proof.* Since the procedure of sampling $(s_h, a_h)$ in Algorithm 4 is identical to the one in Algorithm 3, from Lemma 4, the first two results are verified. It remains to show that $\widehat{A}_{s_h,a_h}(\theta)$ is unbiased.

The estimation of $\widehat{A}_{s_h,a_h}(\theta)$ is decomposed into the estimations of $\widehat{Q}_{s_h,a_h}(\theta)$ and $\widehat{V}_{s_h}(\theta)$. The procedure of estimating $\widehat{Q}_{s_h,a_h}(\theta)$ is also identical to the one in Algorithm 3. Thus, from Lemma 4, we have

$$\mathbb{E}\left[\widehat{Q}_{s_h,a_h}(\theta) \mid s_h, a_h\right] = Q_{s_h,a_h}(\theta).$$

By following the similar arguments of Lemma 4, one can verify that

$$\mathbb{E}\left[\widehat{V}_{s_h}(\theta) \mid s_h, a_h\right] = V_{s_h}(\theta).$$

Combine the above two equalities and obtain that

$$\mathbb{E}\left[\widehat{A}_{s_h,a_h}(\theta) \mid s_h, a_h\right] = \mathbb{E}\left[\widehat{Q}_{s_h,a_h}(\theta) - \widehat{V}_{s_h}(\theta) \mid s_h, a_h\right] = Q_{s_h,a_h}(\theta) - V_{s_h}(\theta) = A_{s_h,a_h}(\theta).$$

$\square$

### C.3 SGD Procedures for Solving the Regression Problems of NPG and Q-NPG

Once we obtain the sampled $(s, a)$ and $\widehat{A}_{s,a}(\theta^{(k)})$ from Algorithm 4, we can apply the averaged SGD algorithm as in Bach & Moulines (2013) to solve the regression problem (32) of NPG for every iteration $k$.

Here we suppress the superscript $(k)$. For any parameter $\theta \in \mathbb{R}^m$, recall the compatible function approximation $L_A$ in (32)

$$L_A(w, \theta, \tilde{d}^\theta) = \mathbb{E}_{(s,a)\sim\tilde{d}^\theta}\left[\left(w^\top \bar{\phi}_{s,a}(\theta) - A_{s,a}(\theta)\right)^2\right].$$

With the output $(s, a) \sim \tilde{d}^\theta$ and $\widehat{A}_{s,a}(\theta)$ from Algorithm 4 (here we suppress the subscript $h$), we compute the stochastic gradient estimator of the function $L_A$ in (32) by

$$\widehat{\nabla}_w L_A(w, \theta, \tilde{d}^\theta) \overset{\text{def}}{=} 2\left(w^\top \bar{\phi}_{s,a}(\theta) - \widehat{A}_{s,a}(\theta)\right) \bar{\phi}_{s,a}(\theta). \tag{44}$$

Next, we show that (44) is an unbiased gradient estimator of the loss function $L_A$.

**Lemma 6.** *Consider the output $(s, a)$ and $\widehat{A}_{s,a}(\theta)$ of Algorithm 4 and the stochastic gradient (44). It follows that*

$$\mathbb{E}\left[\widehat{\nabla}_w L_A(w, \theta, \tilde{d}^\theta)\right] = \nabla_w L_A(w, \theta, \tilde{d}^\theta),$$

*where the expectation is with respect to the randomness in the sequence of the sampled $s_0, a_0, \cdots, s_t, a_t$ from Algorithm 4.*

*Proof.* The total expectation of the stochastic gradient is given by

$$\mathbb{E}\left[\widehat{\nabla}_w L_A(w, \theta, \tilde{d}^\theta)\right] \overset{(44)}{=} \mathbb{E}_{s, a, \widehat{A}_{s,a}(\theta)}\left[2\left(w^\top \bar{\phi}_{s,a}(\theta) - \widehat{A}_{s,a}(\theta)\right)\bar{\phi}_{s,a}(\theta)\right]$$

$$= \mathbb{E}_{(s,a)\sim \tilde{d}^\theta, \widehat{A}_{s,a}(\theta)}\left[2\left(w^\top \bar{\phi}_{s,a}(\theta) - \widehat{A}_{s,a}(\theta)\right)\bar{\phi}_{s,a}(\theta) \mid s, a\right], \quad (45)$$

where the second line is obtained by $(s, a) \sim \tilde{d}^\theta$ from Lemma 5.

From Lemma 5, we have

$$\mathbb{E}_{s_0, a_0, \cdots, s_t, a_t}\left[\widehat{A}_{s,a}(\theta) \mid s_0 = s, a_0 = a\right] = A_{s,a}(\theta). \quad (46)$$

Combining the above two equalities yield

$$\mathbb{E}\left[\widehat{\nabla}_w L_A(w, \theta, \tilde{d}^\theta)\right] \overset{(45)}{=} \mathbb{E}_{(s,a)\sim \tilde{d}^\theta}\left[2\left(w^\top \bar{\phi}_{s,a}(\theta) - \mathbb{E}\left[\widehat{A}_{s,a}(\theta) \mid s, a\right]\right)\bar{\phi}_{s,a}(\theta)\right]$$

$$\overset{(46)}{=} \mathbb{E}_{(s,a)\sim \tilde{d}^\theta}\left[2\left(w^\top \bar{\phi}_{s,a}(\theta) - A_{s,a}(\theta)\right)\bar{\phi}_{s,a}(\theta)\right]$$

$$= \nabla_w L_A(w, \theta, \tilde{d}^\theta),$$

as desired. □

Since (44) is unbiased shown in Lemma 6, we can use it for the averaged SGD algorithm to minimize $L_A$, called `NPG-SGD` in Algorithm 5 (also see Agarwal et al., 2021, Algorithm 4).

---

**Algorithm 5:** NPG-SGD

---

**Input:** Number of iterations $T$, step size $\alpha > 0$, initialization $w_0 \in \mathbb{R}^m$, initial state-action measure $\nu$, policy $\pi(\theta)$, discounted factor $\gamma \in [0, 1)$

1 **for** $t = 0$ **to** $T - 1$ **do**

2     Call Algorithm 4 with the inputs $(\nu, \pi(\theta), \gamma)$ to sample $(s, a) \sim \tilde{d}^\theta$ and $\widehat{A}_{s,a}(\theta)$

3     Update $w_{t+1} = w_t - \alpha\widehat{\nabla}_w L_A(w, \theta, \tilde{d}^\theta)$ by using (44)

**Output:** $w_{\text{out}} = \frac{1}{T}\sum_{t=1}^{T} w_t$

---

Similar to Algorithm 5, once we obtain the sampled $(s, a)$ and $\widehat{Q}_{s,a}(\theta)$ from Algorithm 3, we can apply the averaged SGD algorithm to solve (18) of Q-NPG.

Recall the compatible function approximation $L_Q$ in (18)

$$L_Q(w, \theta, \tilde{d}^\theta) = \mathbb{E}_{(s,a)\sim \tilde{d}^\theta}\left[\left(w^\top \phi_{s,a} - Q_{s,a}(\theta)\right)^2\right].$$

With the output $(s, a) \sim \tilde{d}^\theta$ and $\widehat{Q}_{s,a}(\theta)$ from Algorithm 3, we compute the stochastic gradient estimator of the function $L_Q$ in (18) by

$$\widehat{\nabla}_w L_Q(w, \theta, \tilde{d}^\theta) \overset{\text{def}}{=} 2\left(w^\top \phi_{s,a} - \widehat{Q}_{s,a}(\theta)\right)\phi_{s,a}, \quad (47)$$

and use it for the averaged SGD algorithm to minimize $L_Q$, called `Q-NPG-SGD` in Algorithm 6 (also see Agarwal et al., 2021, Algorithm 2). Compared to (44), the cost of computing (47) is $|\mathcal{A}|$ times cheaper than that of computing (47). Indeed, to compute (47), we only need one single action for $\phi_{s,a}$, while to compute (44), one needs to go through all the actions to compute $\bar{\phi}_{s,a}(\theta)$. Thus, the computational cost of `Q-NPG-SGD` is $|\mathcal{A}|$ times cheaper than that of `NPG-SGD`.

The estimator $\widehat{\nabla}_w L_Q(w, \theta, \tilde{d}^\theta)$ is also unbiased following the similar argument of the proof of Lemma 6. We formalize this in the following and omit the proof.

**Lemma 7.** *Consider the output $(s, a)$ and $\widehat{Q}_{s,a}(\theta)$ of Algorithm 3 and the stochastic gradient (47). It follows that*

$$\mathbb{E}\left[\widehat{\nabla}_w L_Q(w, \theta, \tilde{d}^\theta)\right] = \nabla_w L_Q(w, \theta, \tilde{d}^\theta),$$

*where the expectation is with respect to the randomness in the sequence of the sampled $s_0, a_0, \cdots, s_t, a_t$ from Algorithm 3.*

---

**Algorithm 6:** Q-NPG-SGD

---

**Input:** Number of iterations $T$, step size $\alpha > 0$, initialization $w_0 \in \mathbb{R}^m$, initial state-action measure $\nu$, policy $\pi(\theta)$, discounted factor $\gamma \in [0, 1)$

1 **for** $t = 0$ **to** $T - 1$ **do**

2     Call Algorithm 3 with the inputs $(\nu, \pi(\theta), \gamma)$ to sample $(s, a) \sim \tilde{d}^\theta$ and $\widehat{Q}_{s,a}(\theta)$

3     Update $w_{t+1} = w_t - \alpha \widehat{\nabla}_w L_Q(w, \theta, \tilde{d}^\theta)$ by using (47)

**Output:** $w_{\text{out}} = \frac{1}{T} \sum_{t=1}^{T} w_t$

---

# D   PROOF OF SECTION 4

Throughout this section and the next, we use the shorthand $V_\rho^{(k)}$ for $V_\rho(\theta^{(k)})$ and similarly, $Q_{s,a}^{(k)}$ for $Q_{s,a}(\theta^{(k)})$ and $A_{s,a}^{(k)}$ for $A_{s,a}(\theta^{(k)})$. We also use the shorthand $Q_s^{(k)}$ for the vector $\left[ Q_{s,a}^{(k)} \right]_{a \in \mathcal{A}} \in \mathbb{R}^{|\mathcal{A}|}$ and $A_s^{(k)}$ for the vector $\left[ A_{s,a}^{(k)} \right]_{a \in \mathcal{A}} \in \mathbb{R}^{|\mathcal{A}|}$.

We also introduce a weighted KL divergence given by

$$D_k^* \stackrel{\text{def}}{=} \mathbb{E}_{s \sim d^*} \left[ D(\pi_s^*, \pi_s^{(k)}) \right].$$

If we choose the uniform initial policy, i.e., $\pi_{s,a}^{(0)} = 1/|\mathcal{A}|$ for all $(s, a) \in \mathcal{S} \times \mathcal{A}$ (or $\theta^{(0)} = 0$), then

$$D_0^* \leq \log |\mathcal{A}|, \tag{48}$$

for all $\rho \in \Delta(\mathcal{S})$ and for any $\pi^* \in \Delta(\mathcal{A})^{\mathcal{S}}$.

We first provide the one step analysis of the Q-NPG update, which will be helpful for proving Theorem 1 and 3 and the sublinear convergence result of Theorem 2 further in Appendix D.3.

## D.1   THE ONE STEP Q-NPG LEMMA

The following one step analysis of Q-NPG is based on the mirror descent approach of Xiao (2022).

**Lemma 8** (One step Q-NPG lemma). *Fix a state distribution $\rho$; an initial state-action distribution $\nu$; an arbitrary comparator policy $\pi^*$. Let $w_\star^{(k)} \in \operatorname{argmin}_w L_Q(w, \theta^{(k)}, \tilde{d}^{(k)})$ denote the exact minimizer. Consider the $w^{(k)}$ and $\pi^{(k)}$ given in (18) and (16) respectively. We have that*

$$\vartheta_\rho (1 - \gamma) \left( V_\rho^{(k+1)} - V_\rho^{(k)} \right) + (1 - \gamma) \left( V_\rho^{(k)} - V_\rho(\pi^*) \right)$$

$$+ \vartheta_\rho \Bigg( \underbrace{\sum_{s \in \mathcal{S}} \sum_{a \in \mathcal{A}} d_s^{(k+1)} \pi_{s,a}^{(k+1)} \phi_{s,a}^\top \left( w^{(k)} - w_\star^{(k)} \right)}_{\textcircled{1}} + \underbrace{\sum_{s \in \mathcal{S}} \sum_{a \in \mathcal{A}} d_s^{(k+1)} \pi_{s,a}^{(k+1)} \left( \phi_{s,a}^\top w_\star^{(k)} - Q_{s,a}^{(k)} \right)}_{\textcircled{2}}$$

$$+ \underbrace{\sum_{s \in \mathcal{S}} \sum_{a \in \mathcal{A}} d_s^{(k+1)} \pi_{s,a}^{(k)} \phi_{s,a}^\top \left( w_\star^{(k)} - w^{(k)} \right)}_{\textcircled{3}} + \underbrace{\sum_{s \in \mathcal{S}} \sum_{a \in \mathcal{A}} d_s^{(k+1)} \pi_{s,a}^{(k)} \left( Q_{s,a}^{(k)} - \phi_{s,a}^\top w_\star^{(k)} \right)}_{\textcircled{4}} \Bigg)$$

$$+ \underbrace{\sum_{(s,a) \in \mathcal{S} \times \mathcal{A}} d_s^* \pi_{s,a}^{(k)} \phi_{s,a}^\top \left( w^{(k)} - w_\star^{(k)} \right)}_{\textcircled{a}} + \underbrace{\sum_{(s,a) \in \mathcal{S} \times \mathcal{A}} d_s^* \pi_{s,a}^{(k)} \left( \phi_{s,a}^\top w_\star^{(k)} - Q_{s,a}^{(k)} \right)}_{\textcircled{b}}$$

$$+ \underbrace{\sum_{(s,a) \in \mathcal{S} \times \mathcal{A}} d_s^* \pi_{s,a}^* \phi_{s,a}^\top \left( w_\star^{(k)} - w^{(k)} \right)}_{\textcircled{c}} + \underbrace{\sum_{(s,a) \in \mathcal{S} \times \mathcal{A}} d_s^* \pi_{s,a}^* \left( Q_{s,a}^{(k)} - \phi_{s,a}^\top w_\star^{(k)} \right)}_{\textcircled{d}}$$

$$\leq \frac{1}{\eta_k} D_k^* - \frac{1}{\eta_k} D_{k+1}^*. \tag{49}$$

*Proof.* In the context of the PMD method (16), we apply the three-point descent lemma (Lemma 11) with $\mathcal{C} = \Delta(\mathcal{A})$, $f$ is the linear function $\eta_k \left\langle \Phi_s w^{(k)}, \cdot \right\rangle$ and $h : \Delta(\mathcal{A}) \to \mathbb{R}$ is the negative entropy with $h(p) = \sum_{a \in \mathcal{A}} p_a \log p_a$. Thus, $h$ is of Legendre type with rint dom $h \cap \mathcal{C} = $ rint $\Delta(\mathcal{A}) \neq \emptyset$ and $D_h(\cdot, \cdot)$ is the KL divergence $D(\cdot, \cdot)$. From Lemma 11, we obtain that for any $p \in \Delta(\mathcal{A})$, we have

$$\eta_k \left\langle \Phi_s w^{(k)}, \pi_s^{(k+1)} \right\rangle + D(\pi_s^{(k+1)}, \pi_s^{(k)}) \leq \eta_k \left\langle \Phi_s w^{(k)}, p \right\rangle + D(p, \pi_s^{(k)}) - D(p, \pi_s^{(k+1)}).$$

Rearranging terms and dividing both sides by $\eta_k$, we get

$$\left\langle \Phi_s w^{(k)}, \pi_s^{(k+1)} - p \right\rangle + \frac{1}{\eta_k} D(\pi_s^{(k+1)}, \pi_s^{(k)}) \leq \frac{1}{\eta_k} D(p, \pi_s^{(k)}) - \frac{1}{\eta_k} D(p, \pi_s^{(k+1)}). \quad (50)$$

Letting $p = \pi_s^{(k)}$ yields

$$\left\langle \Phi_s w^{(k)}, \pi_s^{(k+1)} - \pi_s^{(k)} \right\rangle \leq -\frac{1}{\eta_k} D(\pi_s^{(k+1)}, \pi_s^{(k)}) - \frac{1}{\eta_k} D(\pi_s^{(k)}, \pi_s^{(k+1)}) \leq 0. \quad (51)$$

Letting $p = \pi_s^*$ and subtract and add $\pi_s^{(k)}$ within the inner product term in (50) yields

$$\left\langle \Phi_s w^{(k)}, \pi_s^{(k+1)} - \pi_s^{(k)} \right\rangle + \left\langle \Phi_s w^{(k)}, \pi_s^{(k)} - \pi_s^* \right\rangle \leq \frac{1}{\eta_k} D(\pi_s^*, \pi_s^{(k)}) - \frac{1}{\eta_k} D(\pi_s^*, \pi_s^{(k+1)}).$$

Note that we dropped the nonnegative term $\frac{1}{\eta_k} D(\pi_s^{(k+1)}, \pi_s^{(k)})$ on the left hand side to the inequality. Taking expectation with respect to the distribution $d^*$, we have

$$\mathbb{E}_{s \sim d^*} \left[ \left\langle \Phi_s w^{(k)}, \pi_s^{(k+1)} - \pi_s^{(k)} \right\rangle \right] + \mathbb{E}_{s \sim d^*} \left[ \left\langle \Phi_s w^{(k)}, \pi_s^{(k)} - \pi_s^* \right\rangle \right] \leq \frac{1}{\eta_k} D_k^* - \frac{1}{\eta_k} D_{k+1}^*. \quad (52)$$

For the first expectation in (52), we have

$$\mathbb{E}_{s \sim d^*} \left[ \left\langle \Phi_s w^{(k)}, \pi_s^{(k+1)} - \pi_s^{(k)} \right\rangle \right]$$

$$= \sum_{s \in \mathcal{S}} d_s^* \left\langle \Phi_s w^{(k)}, \pi_s^{(k+1)} - \pi_s^{(k)} \right\rangle$$

$$= \sum_{s \in \mathcal{S}} \frac{d_s^*}{d_s^{(k+1)}} d_s^{(k+1)} \left\langle \Phi_s w^{(k)}, \pi_s^{(k+1)} - \pi_s^{(k)} \right\rangle$$

$$\geq \vartheta_{k+1} \sum_{s \in \mathcal{S}} d_s^{(k+1)} \left\langle \Phi_s w^{(k)}, \pi_s^{(k+1)} - \pi_s^{(k)} \right\rangle$$

$$\geq \vartheta_\rho \sum_{s \in \mathcal{S}} d_s^{(k+1)} \left\langle \Phi_s w^{(k)}, \pi_s^{(k+1)} - \pi_s^{(k)} \right\rangle$$

$$= \vartheta_\rho \sum_{s \in \mathcal{S}} d_s^{(k+1)} \left\langle Q_s^{(k)}, \pi_s^{(k+1)} - \pi_s^{(k)} \right\rangle + \vartheta_\rho \sum_{s \in \mathcal{S}} d_s^{(k+1)} \left\langle \Phi_s w^{(k)} - Q_s^{(k)}, \pi_s^{(k+1)} - \pi_s^{(k)} \right\rangle$$

$$= \vartheta_\rho (1 - \gamma) \left( V_\rho^{(k+1)} - V_\rho^{(k)} \right) + \vartheta_\rho \sum_{s \in \mathcal{S}} d_s^{(k+1)} \left\langle \Phi_s w^{(k)} - Q_s^{(k)}, \pi_s^{(k+1)} - \pi_s^{(k)} \right\rangle, \quad (53)$$

where the last equality is due to the performance difference lemma (42) in Lemma 3 and the two inequalities above are obtained by the negative sign of $\left\langle \Phi_s w^{(k)}, \pi_s^{(k+1)} - \pi_s^{(k)} \right\rangle$ shown in (51) and by using the following inequality

$$\frac{d_s^*}{d_s^{(k+1)}} \overset{(20)}{\leq} \vartheta_{k+1} \overset{(20)}{\leq} \vartheta_\rho.$$

The second term of (53) can be decomposed into four terms. That is,

$$
\sum_{s \in \mathcal{S}} d_s^{(k+1)} \left\langle \Phi_s w^{(k)} - Q_s^{(k)}, \pi_s^{(k+1)} - \pi_s^{(k)} \right\rangle
$$

$$
= \sum_{s \in \mathcal{S}} \sum_{a \in \mathcal{A}} d_s^{(k+1)} \pi_{s,a}^{(k+1)} \left( \phi_{s,a}^\top w^{(k)} - Q_{s,a}^{(k)} \right) + \sum_{s \in \mathcal{S}} \sum_{a \in \mathcal{A}} d_s^{(k+1)} \pi_{s,a}^{(k)} \left( Q_{s,a}^{(k)} - \phi_{s,a}^\top w^{(k)} \right)
$$

$$
= \sum_{s \in \mathcal{S}} \sum_{a \in \mathcal{A}} d_s^{(k+1)} \pi_{s,a}^{(k+1)} \phi_{s,a}^\top \left( w^{(k)} - w_\star^{(k)} \right) + \sum_{s \in \mathcal{S}} \sum_{a \in \mathcal{A}} d_s^{(k+1)} \pi_{s,a}^{(k+1)} \left( \phi_{s,a}^\top w_\star^{(k)} - Q_{s,a}^{(k)} \right)
$$

$$
+ \sum_{s \in \mathcal{S}} \sum_{a \in \mathcal{A}} d_s^{(k+1)} \pi_{s,a}^{(k)} \phi_{s,a}^\top \left( w_\star^{(k)} - w^{(k)} \right) + \sum_{s \in \mathcal{S}} \sum_{a \in \mathcal{A}} d_s^{(k+1)} \pi_{s,a}^{(k)} \left( Q_{s,a}^{(k)} - \phi_{s,a}^\top w_\star^{(k)} \right)
$$

$$
= ① + ② + ③ + ④, \tag{54}
$$

where ①, ②, ③ and ④ are defined in (49).

For the second expectation in (52), by applying again the performance difference lemma (42), we have

$$
\mathbb{E}_{s \sim d^*} \left[ \left\langle \Phi_s w^{(k)}, \pi_s^{(k)} - \pi_s^* \right\rangle \right]
$$

$$
= \quad \mathbb{E}_{s \sim d^*} \left[ \left\langle Q_s^{(k)}, \pi_s^{(k)} - \pi_s^* \right\rangle \right] + \mathbb{E}_{s \sim d^*} \left[ \left\langle \Phi_s w^{(k)} - Q_s^{(k)}, \pi_s^{(k)} - \pi_s^* \right\rangle \right]
$$

$$
\stackrel{(42)}{=} \quad (1 - \gamma) \left( V_\rho^{(k)} - V_\rho(\pi^*) \right) + \mathbb{E}_{s \sim d^*} \left[ \left\langle \Phi_s w^{(k)} - Q_s^{(k)}, \pi_s^{(k)} - \pi_s^* \right\rangle \right]. \tag{55}
$$

Similarly, we decompose the second term of (55) into four terms. That is,

$$
\mathbb{E}_{s \sim d^*} \left[ \left\langle \Phi_s w^{(k)} - Q_s^{(k)}, \pi_s^{(k)} - \pi_s^* \right\rangle \right]
$$

$$
= \sum_{s \in \mathcal{S}} \sum_{a \in \mathcal{A}} d_s^* \pi_{s,a}^{(k)} \left( \phi_{s,a}^\top w^{(k)} - Q_{s,a}^{(k)} \right) + \sum_{s \in \mathcal{S}} \sum_{a \in \mathcal{A}} d_s^* \pi_{s,a}^* \left( Q_{s,a}^{(k)} - \phi_{s,a}^\top w^{(k)} \right)
$$

$$
= \sum_{(s,a) \in \mathcal{S} \times \mathcal{A}} d_s^* \pi_{s,a}^{(k)} \phi_{s,a}^\top \left( w^{(k)} - w_\star^{(k)} \right) + \sum_{(s,a) \in \mathcal{S} \times \mathcal{A}} d_s^* \pi_{s,a}^{(k)} \left( \phi_{s,a}^\top w_\star^{(k)} - Q_{s,a}^{(k)} \right)
$$

$$
+ \sum_{(s,a) \in \mathcal{S} \times \mathcal{A}} d_s^* \pi_{s,a}^* \phi_{s,a}^\top \left( w_\star^{(k)} - w^{(k)} \right) + \sum_{(s,a) \in \mathcal{S} \times \mathcal{A}} d_s^* \pi_{s,a}^* \left( Q_{s,a}^{(k)} - \phi_{s,a}^\top w_\star^{(k)} \right)
$$

$$
= ⓐ + ⓑ + ⓒ + ⓓ, \tag{56}
$$

where ⓐ, ⓑ, ⓒ and ⓓ are defined in (49).

Plugging (53) with the decomposition (54) and (55) with the decomposition (56) into (52) concludes the proof. □

Consequently, the convergence analysis of Q-NPG (Theorem 1, 3 and Theorem 2 further in Appendix D.3) will be obtained by upper bounding the absolute values of ①, ②, ③, ④, ⓐ, ⓑ, ⓒ, ⓓ in (49) with different set of assumptions (assumptions in Theorem 1 or assumptions in Theorem 3) and with different step size scheme (geometrically increasing step size for Theorem 1 and 3 or constant step size for Theorem 2).

## D.2 PROOF OF THEOREM 1

Here we provide the general Q-NPG convergence analysis with any initial policy, which is in contrast to Theorem 1 with the uniform initial policy.

**Theorem 5.** *Fix a state distribution $\rho$, an state-action distribution $\nu$ and a comparator policy $\pi^*$. We consider the Q-NPG method (18) with the step sizes satisfying $\eta_0 \geq \frac{1-\gamma}{\gamma} D_0^*$ and $\eta_{k+1} \geq \frac{1}{\gamma} \eta_k$. Suppose that Assumptions 1, 2, 3 and 4 all hold. Then we have for all $k \geq 0$,*

$$
\mathbb{E} \left[ V_\rho(\pi^{(k)}) \right] - V_\rho(\pi^*) \leq \left( 1 - \frac{1}{\vartheta_\rho} \right)^k \frac{2}{1 - \gamma} + \frac{2\sqrt{|\mathcal{A}|} \left( \vartheta_\rho \sqrt{C_\rho} + 1 \right)}{1 - \gamma} \left( \sqrt{\frac{\kappa_\nu}{1 - \gamma} \epsilon_{\text{stat}}} + \sqrt{\epsilon_{\text{bias}}} \right).
$$

**Remark.** A deviation from the setting of Theorem 1 is that here we require the step sizes satisfying $\eta_0 \geq \frac{1-\gamma}{\gamma} D_0^*$ instead of $\eta_0 \geq \frac{1-\gamma}{\gamma} \log |\mathcal{A}|$. This is because when using uniform initial policy in Theorem 1, from (48), we know that $\eta_0 \geq \frac{1-\gamma}{\gamma} \log |\mathcal{A}|$ is a sufficient condition for $\eta_0 \geq \frac{1-\gamma}{\gamma} D_0^*$. Consequently, to prove Theorem 1, it suffices to prove the more general Theorem 5.

*Proof.* From (49) in Lemma 8, we will upper bound $|①|$ and $|③|$ by the statistical error assumption (19) and upper bound $|②|$ and $|④|$ by using the transfer error assumption (22).

Indeed, to upper bound $|①|$, by Cauchy-Schwartz's inequality, we have

$$
\begin{aligned}
|①| &\leq \sum_{s \in \mathcal{S}} \sum_{a \in \mathcal{A}} d_s^{(k+1)} \pi_{s,a}^{(k+1)} \left| \phi_{s,a}^\top \left( w^{(k)} - w_\star^{(k)} \right) \right| \\
&\leq \sqrt{ \sum_{(s,a) \in \mathcal{S} \times \mathcal{A}} \frac{\left( d_s^{(k+1)} \right)^2 \left( \pi_{s,a}^{(k+1)} \right)^2}{d_s^* \cdot \mathrm{Unif}_{\mathcal{A}}(a)} \cdot \sum_{(s,a) \in \mathcal{S} \times \mathcal{A}} d_s^* \cdot \mathrm{Unif}_{\mathcal{A}}(a) \left( \phi_{s,a}^\top \left( w^{(k)} - w_\star^{(k)} \right) \right)^2 } \\
&\overset{(23)}{=} \sqrt{ \sum_{(s,a) \in \mathcal{S} \times \mathcal{A}} \frac{\left( d_s^{(k+1)} \right)^2 \left( \pi_{s,a}^{(k+1)} \right)^2}{d_s^* \cdot \mathrm{Unif}_{\mathcal{A}}(a)} \left\| w^{(k)} - w_\star^{(k)} \right\|_{\Sigma_{\tilde{d}^*}}^2 } \\
&\leq \sqrt{ \mathbb{E}_{s \sim d^*} \left[ \left( \frac{d_s^{(k+1)}}{d_s^*} \right)^2 \right] |\mathcal{A}| \left\| w^{(k)} - w_\star^{(k)} \right\|_{\Sigma_{\tilde{d}^*}}^2 } \\
&\overset{(25)}{\leq} \sqrt{ C_\rho |\mathcal{A}| \left\| w^{(k)} - w_\star^{(k)} \right\|_{\Sigma_{\tilde{d}^*}}^2 },
\end{aligned}
\tag{57}
$$

where the second inequality is obtained by Cauchy-Schwartz's inequality, and the third inequality is obtained by the following inequality

$$
\sum_{a \in \mathcal{A}} \left( \pi_{s,a}^{(k+1)} \right)^2 \leq \sum_{a \in \mathcal{A}} \pi_{s,a}^{(k+1)} = 1.
\tag{58}
$$

Then, by using Assumption 3 with the definition of $\kappa_\nu$, (57) is upper bounded by

$$
\begin{aligned}
|①| &\overset{(24)}{\leq} \sqrt{ C_\rho |\mathcal{A}| \kappa_\nu \left\| w^{(k)} - w_\star^{(k)} \right\|_{\Sigma_\nu}^2 } \\
&\overset{(5)}{\leq} \sqrt{ \frac{C_\rho |\mathcal{A}| \kappa_\nu}{1-\gamma} \left\| w^{(k)} - w_\star^{(k)} \right\|_{\Sigma_{\tilde{d}^{(k)}}}^2 },
\end{aligned}
\tag{59}
$$

where we use the shorthand

$$
\Sigma_{\tilde{d}^{(k)}} \overset{\mathrm{def}}{=} \mathbb{E}_{(s,a) \sim \tilde{d}^{(k)}} \left[ \phi_{s,a} \phi_{s,a}^\top \right].
\tag{60}
$$

Besides, by the first-order optimality conditions for the optima $w_\star^{(k)} \in \mathrm{argmin}_w L_Q(w, \theta^{(k)}, \tilde{d}^{(k)})$, we have

$$
(w - w_\star^{(k)})^\top \nabla_w L_Q(w_\star^{(k)}, \theta^{(k)}, \tilde{d}^{(k)}) \geq 0, \qquad \text{for all } w \in \mathbb{R}^m.
\tag{61}
$$

Therefore, for all $w \in \mathbb{R}^m$,

$$
\begin{aligned}
&L_Q(w, \theta^{(k)}, \tilde{d}^{(k)}) - L_Q(w_\star^{(k)}, \theta^{(k)}, \tilde{d}^{(k)}) \\
&= \mathbb{E}_{(s,a) \sim \tilde{d}^{(k)}} \left[ \left( \phi_{s,a}^\top w - \phi_{s,a}^\top w_\star^{(k)} + \phi_{s,a}^\top w_\star^{(k)} - Q_{s,a}^{(k)} \right)^2 \right] - L_Q(w_\star^{(k)}, \theta^{(k)}, \tilde{d}^{(k)}) \\
&= \mathbb{E}_{(s,a) \sim \tilde{d}^{(k)}} \left[ (\phi_{s,a}^\top w - \phi_{s,a}^\top w_\star^{(k)})^2 \right] + 2(w - w_\star^{(k)})^\top \mathbb{E}_{(s,a) \sim \tilde{d}^{(k)}} \left[ (\phi_{s,a}^\top w_\star^{(k)} - Q_{s,a}^{(k)}) \phi_{s,a} \right] \\
&= \left\| w - w_\star^{(k)} \right\|_{\Sigma_{\tilde{d}^{(k)}}}^2 + (w - w_\star^{(k)})^\top \nabla_w L_Q(w_\star^{(k)}, \theta^{(k)}, \tilde{d}^{(k)}) \\
&\overset{(61)}{\geq} \left\| w - w_\star^{(k)} \right\|_{\Sigma_{\tilde{d}^{(k)}}}^2.
\end{aligned}
\tag{62}
$$

Define
$$\epsilon_{\text{stat}}^{(k)} \stackrel{\text{def}}{=} L_Q(w^{(k)}, \theta^{(k)}, \tilde{d}^{(k)}) - L_Q(w_\star^{(k)}, \theta^{(k)}, \tilde{d}^{(k)}).$$
Note that from (19), we have
$$\mathbb{E}\left[\epsilon_{\text{stat}}^{(k)}\right] \leq \epsilon_{\text{stat}}. \tag{63}$$
Plugging (62) into (59), we have
$$|\text{①}| \leq \sqrt{\frac{C_\rho |\mathcal{A}| \kappa_\nu}{1 - \gamma} \epsilon_{\text{stat}}^{(k)}}. \tag{64}$$
Similar to (57), we get the same upper bound for $|\text{③}|$ by just replacing $\pi_{s,a}^{(k+1)}$ into $\pi_{s,a}^{(k)}$. That is,
$$|\text{③}| \leq \sqrt{\frac{C_\rho |\mathcal{A}| \kappa_\nu}{1 - \gamma} \epsilon_{\text{stat}}^{(k)}}. \tag{65}$$

To upper bound $|\text{②}|$ and $|\text{④}|$, we introduce the following term
$$\epsilon_{\text{bias}}^{(k)} \stackrel{\text{def}}{=} L_Q(w_\star^{(k)}, \theta^{(k)}, \tilde{d}^*).$$
Note that from (22), we have
$$\mathbb{E}\left[\epsilon_{\text{bias}}^{(k)}\right] \leq \epsilon_{\text{bias}}. \tag{66}$$
By Cauchy-Schwartz's inequality, we have
$$
\begin{aligned}
|\text{②}| &\leq \sum_{s \in \mathcal{S}} \sum_{a \in \mathcal{A}} d_s^{(k+1)} \pi_{s,a}^{(k+1)} \left| \phi_{s,a}^\top w_\star^{(k)} - Q_{s,a}^{(k)} \right| \\
&\leq \sqrt{\sum_{(s,a) \in \mathcal{S} \times \mathcal{A}} \frac{\left(d_s^{(k+1)}\right)^2 \left(\pi_{s,a}^{(k+1)}\right)^2}{d_s^* \cdot \text{Unif}_{\mathcal{A}}(a)} \cdot \sum_{(s,a) \in \mathcal{S} \times \mathcal{A}} d_s^* \cdot \text{Unif}_{\mathcal{A}}(a) \left(\phi_{s,a}^\top w_\star^{(k)} - Q_{s,a}^{(k)}\right)^2} \\
&= \sqrt{\sum_{(s,a) \in \mathcal{S} \times \mathcal{A}} \frac{\left(d_s^{(k+1)}\right)^2 \left(\pi_{s,a}^{(k+1)}\right)^2}{d_s^* \cdot \text{Unif}_{\mathcal{A}}(a)} \cdot \epsilon_{\text{bias}}^{(k)}} \\
&\stackrel{(58)}{\leq} \sqrt{\mathbb{E}_{s \sim d^*}\left[\left(\frac{d_s^{(k+1)}}{d_s^*}\right)^2\right] |\mathcal{A}| \epsilon_{\text{bias}}^{(k)}} \stackrel{(25)}{\leq} \sqrt{C_\rho |\mathcal{A}| \epsilon_{\text{bias}}^{(k)}}.
\end{aligned} \tag{67}
$$
Similar to (67), we get the same upper bound for $|\text{④}|$ by just replacing $\pi_{s,a}^{(k+1)}$ into $\pi_{s,a}^{(k)}$. That is,
$$|\text{④}| \leq \sqrt{C_\rho |\mathcal{A}| \epsilon_{\text{bias}}^{(k)}}. \tag{68}$$

Next, we will upper bound the absolute values of ⓐ, ⓑ, ⓒ and ⓓ of (49) separately by using again the statistical error (19) and by using the transfer error assumption (22).

Indeed, to upper bound $|\text{ⓐ}|$, by Cauchy-Schwartz's inequality, we have
$$
\begin{aligned}
|\text{ⓐ}| &\leq \sum_{(s,a) \in \mathcal{S} \times \mathcal{A}} d_s^* \pi_{s,a}^{(k)} \left| \phi_{s,a}^\top \left(w^{(k)} - w_\star^{(k)}\right) \right| \\
&\leq \sqrt{\sum_{(s,a) \in \mathcal{S} \times \mathcal{A}} \frac{(d_s^*)^2 \left(\pi_{s,a}^{(k)}\right)^2}{d_s^* \cdot \text{Unif}_{\mathcal{A}}(a)} \sum_{(s,a) \in \mathcal{S} \times \mathcal{A}} d_s^* \cdot \text{Unif}_{\mathcal{A}}(a) \left(\phi_{s,a}^\top \left(w^{(k)} - w_\star^{(k)}\right)\right)^2} \\
&\stackrel{(23)}{=} \sqrt{\sum_{(s,a) \in \mathcal{S} \times \mathcal{A}} \frac{(d_s^*)^2 \left(\pi_{s,a}^{(k)}\right)^2}{d_s^* \cdot \text{Unif}_{\mathcal{A}}(a)} \left\|w^{(k)} - w_\star^{(k)}\right\|_{\Sigma_{\tilde{d}^*}}^2} \\
&\stackrel{(58)}{\leq} \sqrt{|\mathcal{A}| \left\|w^{(k)} - w_\star^{(k)}\right\|_{\Sigma_{\tilde{d}^*}}^2}.
\end{aligned}
$$

From the definition of $\kappa_\nu$, we further obtain

$$
\begin{aligned}
|\text{\textcircled{a}}| &\overset{(24)}{\leq} \sqrt{|\mathcal{A}|\kappa_\nu \left\| w^{(k)} - w_\star^{(k)} \right\|_{\Sigma_\nu}^2} \\
&\overset{(5)}{\leq} \sqrt{\frac{|\mathcal{A}|\kappa_\nu}{1-\gamma} \left\| w^{(k)} - w_\star^{(k)} \right\|_{\Sigma_{\tilde{d}^{(k)}}}^2} \\
&\overset{(62)}{\leq} \sqrt{\frac{|\mathcal{A}|\kappa_\nu}{1-\gamma} \epsilon_{\text{stat}}^{(k)}}.
\end{aligned}
\tag{69}
$$

Similar to (69), we get the same upper bound for $|\text{\textcircled{c}}|$ by just replacing $\pi_{s,a}^{(k)}$ into $\pi_{s,a}^*$. That is,

$$
|\text{\textcircled{c}}| \leq \sqrt{\frac{|\mathcal{A}|\kappa_\nu}{1-\gamma} \epsilon_{\text{stat}}^{(k)}}.
\tag{70}
$$

To upper bound $|\text{\textcircled{b}}|$, by Cauchy-Schwartz's inequality, we have

$$
\begin{aligned}
|\text{\textcircled{b}}| &\leq \sum_{(s,a)\in\mathcal{S}\times\mathcal{A}} d_s^* \pi_{s,a}^{(k)} \left| \left( \phi_{s,a}^\top w_\star^{(k)} - Q_{s,a}^{(k)} \right) \right| \\
&\leq \sqrt{ \sum_{(s,a)\in\mathcal{S}\times\mathcal{A}} \frac{(d_s^*)^2 \left(\pi_{s,a}^{(k)}\right)^2}{d_s^* \cdot \text{Unif}_\mathcal{A}(a)} \sum_{(s,a)\in\mathcal{S}\times\mathcal{A}} d_s^* \cdot \text{Unif}_\mathcal{A}(a) \left( \phi_{s,a}^\top w_\star^{(k)} - Q_{s,a}^{(k)} \right)^2 } \\
&= \sqrt{ \sum_{(s,a)\in\mathcal{S}\times\mathcal{A}} \frac{(d_s^*)^2 \left(\pi_{s,a}^{(k)}\right)^2}{d_s^* \cdot \text{Unif}_\mathcal{A}(a)} \epsilon_{\text{bias}}^{(k)} } \\
&\overset{(58)}{\leq} \sqrt{|\mathcal{A}|\epsilon_{\text{bias}}^{(k)}}.
\end{aligned}
\tag{71}
$$

Similar to (71), we get the same upper bound for $|\text{\textcircled{d}}|$ by just replacing $\pi_{s,a}^{(k)}$ into $\pi_{s,a}^*$. That is,

$$
|\text{\textcircled{d}}| \leq \sqrt{|\mathcal{A}|\epsilon_{\text{bias}}^{(k)}}.
\tag{72}
$$

Plugging all the upper bounds (64) of $|\text{\textcircled{1}}|$, (67) of $|\text{\textcircled{2}}|$, (65) of $|\text{\textcircled{3}}|$, (68) of $|\text{\textcircled{4}}|$, (69) of $|\text{\textcircled{a}}|$, (71) of $|\text{\textcircled{b}}|$, (70) of $|\text{\textcircled{c}}|$ and (72) of $|\text{\textcircled{d}}|$ into (49) yields

$$
\begin{aligned}
\vartheta_\rho \left( \delta_{k+1} - \delta_k \right) + \delta_k \leq{}& \frac{D_k^*}{(1-\gamma)\eta_k} - \frac{D_{k+1}^*}{(1-\gamma)\eta_k} \\
&+ \frac{2\sqrt{|\mathcal{A}|} \left( \vartheta_\rho \sqrt{C_\rho} + 1 \right)}{1-\gamma} \left( \sqrt{\frac{\kappa_\nu}{1-\gamma} \epsilon_{\text{stat}}^{(k)}} + \sqrt{\epsilon_{\text{bias}}^{(k)}} \right),
\end{aligned}
\tag{73}
$$

where $\delta_k \overset{\text{def}}{=} V_\rho^{(k)} - V_\rho(\pi^*)$. Dividing both sides by $\vartheta_\rho$ and rearranging terms, we get

$$
\begin{aligned}
\delta_{k+1} + \frac{D_{k+1}^*}{(1-\gamma)\eta_k \vartheta_\rho} \leq{}& \left( 1 - \frac{1}{\vartheta_\rho} \right) \left( \delta_k + \frac{D_k^*}{(1-\gamma)\eta_k(\vartheta_\rho - 1)} \right) \\
&+ \frac{2\sqrt{|\mathcal{A}|} \left( \sqrt{C_\rho} + \frac{1}{\vartheta_\rho} \right)}{1-\gamma} \left( \sqrt{\frac{\kappa_\nu}{1-\gamma} \epsilon_{\text{stat}}^{(k)}} + \sqrt{\epsilon_{\text{bias}}^{(k)}} \right).
\end{aligned}
$$

If the step sizes satisfy $\eta_{k+1}(\vartheta_\rho - 1) \geq \eta_k \vartheta_\rho$, which is implied by $\eta_{k+1} \geq \eta_k/\gamma$ and (20), then

$$\delta_{k+1} + \frac{D_{k+1}^*}{(1-\gamma)\eta_{k+1}(\vartheta_\rho - 1)}$$
$$\leq \left(1 - \frac{1}{\vartheta_\rho}\right)\left(\delta_k + \frac{D_k^*}{(1-\gamma)\eta_k(\vartheta_\rho - 1)}\right)$$
$$+ \frac{2\sqrt{|\mathcal{A}|}\left(\sqrt{C_\rho} + \frac{1}{\vartheta_\rho}\right)}{1-\gamma}\left(\sqrt{\frac{\kappa_\nu}{1-\gamma}}\epsilon_{\text{stat}}^{(k)} + \sqrt{\epsilon_{\text{bias}}^{(k)}}\right)$$
$$\leq \left(1 - \frac{1}{\vartheta_\rho}\right)^{k+1}\left(\delta_0 + \frac{D_0^*}{(1-\gamma)\eta_0(\vartheta_\rho - 1)}\right)$$
$$+ \sum_{t=0}^{k}\left(1 - \frac{1}{\vartheta_\rho}\right)^{k-t}\frac{2\sqrt{|\mathcal{A}|}\left(\sqrt{C_\rho} + \frac{1}{\vartheta_\rho}\right)}{1-\gamma}\left(\sqrt{\frac{\kappa_\nu}{1-\gamma}}\epsilon_{\text{stat}}^{(t)} + \sqrt{\epsilon_{\text{bias}}^{(t)}}\right).$$

Finally, by choosing $\eta_0 \geq \frac{1-\gamma}{\gamma}D_0^*$ and using the fact that

$$(1-\gamma)(\vartheta_\rho - 1) \overset{(20)}{\geq} (1-\gamma)\left(\frac{1}{1-\gamma} - 1\right) = \gamma,$$

we obtain

$$\delta_k \leq \delta_k + \frac{D_k^*}{(1-\gamma)\eta_k\vartheta_\rho}$$
$$\leq \left(1 - \frac{1}{\vartheta_\rho}\right)^k \frac{2}{1-\gamma}$$
$$+ \frac{2\sqrt{|\mathcal{A}|}\left(\sqrt{C_\rho} + \frac{1}{\vartheta_\rho}\right)}{1-\gamma}\sum_{t=0}^{k-1}\left(1 - \frac{1}{\vartheta_\rho}\right)^{k-1-t}\left(\sqrt{\frac{\kappa_\nu}{1-\gamma}}\epsilon_{\text{stat}}^{(t)} + \sqrt{\epsilon_{\text{bias}}^{(t)}}\right).$$

Taking the total expectation with respect to the randomness in the sequence of the iterates $w^{(0)}, \cdots, w^{(k-1)}$, we have

$$\mathbb{E}\left[V_\rho(\pi^{(k)})\right] - V_\rho(\pi^*)$$
$$\leq \left(1 - \frac{1}{\vartheta_\rho}\right)^k \frac{2}{1-\gamma}$$
$$+ \frac{2\sqrt{|\mathcal{A}|}\left(\sqrt{C_\rho} + \frac{1}{\vartheta_\rho}\right)}{1-\gamma}\sum_{t=0}^{k-1}\left(1 - \frac{1}{\vartheta_\rho}\right)^{k-1-t}\left(\mathbb{E}\left[\sqrt{\frac{\kappa_\nu}{1-\gamma}}\epsilon_{\text{stat}}^{(t)}\right] + \mathbb{E}\left[\sqrt{\epsilon_{\text{bias}}^{(t)}}\right]\right)$$
$$\leq \left(1 - \frac{1}{\vartheta_\rho}\right)^k \frac{2}{1-\gamma}$$
$$+ \frac{2\sqrt{|\mathcal{A}|}\left(\sqrt{C_\rho} + \frac{1}{\vartheta_\rho}\right)}{1-\gamma}\sum_{t=0}^{k-1}\left(1 - \frac{1}{\vartheta_\rho}\right)^{k-1-t}\left(\sqrt{\frac{\kappa_\nu}{1-\gamma}\mathbb{E}\left[\epsilon_{\text{stat}}^{(t)}\right]} + \sqrt{\mathbb{E}\left[\epsilon_{\text{bias}}^{(t)}\right]}\right)$$
$$\overset{(63)+(66)}{\leq} \left(1 - \frac{1}{\vartheta_\rho}\right)^k \frac{2}{1-\gamma}$$
$$+ \frac{2\sqrt{|\mathcal{A}|}\left(\sqrt{C_\rho} + \frac{1}{\vartheta_\rho}\right)}{1-\gamma}\sum_{t=0}^{k-1}\left(1 - \frac{1}{\vartheta_\rho}\right)^{k-1-t}\left(\sqrt{\frac{\kappa_\nu}{1-\gamma}}\epsilon_{\text{stat}} + \sqrt{\epsilon_{\text{bias}}}\right)$$
$$\leq \left(1 - \frac{1}{\vartheta_\rho}\right)^k \frac{2}{1-\gamma} + \frac{2\sqrt{|\mathcal{A}|}\left(\vartheta_\rho\sqrt{C_\rho} + 1\right)}{1-\gamma}\left(\sqrt{\frac{\kappa_\nu}{1-\gamma}}\epsilon_{\text{stat}} + \sqrt{\epsilon_{\text{bias}}}\right),$$

where the second inequality is obtained by Jensen's inequality. This concludes the proof. $\qquad \square$

### D.3 PROOF OF THEOREM 2

Similar to Theorem 5, to prove Theorem 2, it suffices to prove the following sublinear convergence of Q-NPG with any initial policy.

**Theorem 6.** *Fix a state distribution $\rho$, an state-action distribution $\nu$ and an optimal policy $\pi^*$. We consider the Q-NPG method (18) with any constant step size $\eta_k = \eta > 0$. Suppose that Assumptions 1, 2, 3 and 4 all hold. Then we have for all $k \geq 0$,*

$$\frac{1}{k}\sum_{t=0}^{k-1}\mathbb{E}\left[V_\rho(\pi^{(t)})\right] - V_\rho(\pi^*) \leq \frac{1}{(1-\gamma)k}\left(\frac{D_0^*}{\eta} + 2\vartheta_\rho\right)$$
$$+ \frac{2\sqrt{|\mathcal{A}|}\left(\vartheta_\rho\sqrt{C_\rho}+1\right)}{1-\gamma}\left(\sqrt{\frac{\kappa_\nu}{1-\gamma}\epsilon_{\text{stat}}} + \sqrt{\epsilon_{\text{bias}}}\right).$$

**Remark.** A deviation from the setting of Theorem 1 is that here we require $\pi^*$ to be an optimal policy [5]. Compared to Theorem 20 in Agarwal et al. (2021), our convergence rate is also sublinear, but with an improved convergence rate of $\mathcal{O}(1/k)$, as opposed to $\mathcal{O}(1/\sqrt{k})$. Moreover, they use a diminishing step size of order $\mathcal{O}(1/\sqrt{k})$ while our constant step size is unconstrained.

*Proof.* By (73) and using a constant step size $\eta$, we have

$$\vartheta_\rho\left(\delta_{k+1}-\delta_k\right)+\delta_k \leq \frac{D_k^*}{(1-\gamma)\eta} - \frac{D_{k+1}^*}{(1-\gamma)\eta} + \frac{2\sqrt{|\mathcal{A}|}\left(\vartheta_\rho\sqrt{C_\rho}+1\right)}{1-\gamma}\left(\sqrt{\frac{\kappa_\nu}{1-\gamma}\epsilon_{\text{stat}}^{(k)}} + \sqrt{\epsilon_{\text{bias}}^{(k)}}\right).$$

Taking the total expectation with respect to the randomness in the sequence of the iterates $w^{(0)}, \cdots, w^{(k-1)}$, summing up from 0 to $k-1$ and rearranging terms, we have

$$\vartheta_\rho\mathbb{E}\left[\delta_k\right] + \sum_{t=0}^{k-1}\mathbb{E}\left[\delta_t\right] \leq \frac{D_0^*}{(1-\gamma)\eta} + \vartheta_\rho\delta_0 + k\cdot\frac{2\sqrt{|\mathcal{A}|}\left(\vartheta_\rho\sqrt{C_\rho}+1\right)}{1-\gamma}\left(\sqrt{\frac{\kappa_\nu}{1-\gamma}\epsilon_{\text{stat}}} + \sqrt{\epsilon_{\text{bias}}}\right),$$

where we use the following inequalities

$$\mathbb{E}\left[\sqrt{\epsilon_{\text{stat}}^{(t)}}\right] \leq \sqrt{\mathbb{E}\left[\epsilon_{\text{stat}}^{(t)}\right]} \overset{(63)}{\leq} \sqrt{\epsilon_{\text{stat}}},$$

$$\mathbb{E}\left[\sqrt{\epsilon_{\text{bias}}^{(t)}}\right] \leq \sqrt{\mathbb{E}\left[\epsilon_{\text{bias}}^{(t)}\right]} \overset{(66)}{\leq} \sqrt{\epsilon_{\text{bias}}}.$$

Finally, dropping the positive term $\mathbb{E}\left[\delta_k\right]$ on the left hand side as $\pi^*$ is the optimal policy and dividing both side by $k$ yields

$$\frac{1}{k}\sum_{t=0}^{k-1}\mathbb{E}\left[V_\rho(\pi^{(t)})\right] - V_\rho(\pi^*) \leq \frac{D_0^*}{(1-\gamma)\eta k} + \frac{2\vartheta_\rho}{(1-\gamma)k}$$
$$+ \frac{2\sqrt{|\mathcal{A}|}\left(\vartheta_\rho\sqrt{C_\rho}+1\right)}{1-\gamma}\left(\sqrt{\frac{\kappa_\nu}{1-\gamma}\epsilon_{\text{stat}}} + \sqrt{\epsilon_{\text{bias}}}\right).$$

$\square$

### D.4 PROOF OF THEOREM 3

Similar to Theorem 5, to prove Theorem 3, it suffices to prove the following Q-NPG result with any initial policy.

**Theorem 7.** *Fix a state distribution $\rho$, an state-action distribution $\nu$ and a comparator policy $\pi^*$. We consider the Q-NPG method (18) with the step sizes satisfying $\eta_0 \geq \frac{1-\gamma}{\gamma}D_0^*$ and $\eta_{k+1} \geq \frac{1}{\gamma}\eta_k$. Suppose that Assumptions 1, 5 and 6 hold. Then we have for all $k \geq 0$,*

$$\mathbb{E}\left[V_\rho(\pi^{(k)})\right] - V_\rho(\pi^*) \leq \left(1-\frac{1}{\vartheta_\rho}\right)^k\frac{2}{1-\gamma} + \frac{2\sqrt{C_\nu}\left(\vartheta_\rho+1\right)}{1-\gamma}\left(\sqrt{\epsilon_{\text{stat}}} + \sqrt{\epsilon_{\text{approx}}}\right).$$

[5]In our analysis, we need to drop the positive term $\mathbb{E}[V_\rho(\theta^{(k)})] - V_\rho(\pi^*)$ to obtain a lower bound, thus require $\pi^*$ to be an optimal policy.

*Proof.* Similar to the proof of Theorem 1, by Lemma 8, we upper bound the absolute values of ①, ②, ③, ④, ⓐ, ⓑ, ⓒ, ⓓ introduced in (49), separately, with the set of assumptions in Theorem 3.

In comparison with the proof of Theorem 1, we will also upper bound $|①|$, $|③|$, $|ⓐ|$ and $|ⓒ|$ by the statistical error assumption (19) as in the proof of Theorem 1. However, we will upper bound $|②|$, $|④|$, $|ⓑ|$ and $|ⓓ|$ by using the approximation error assumption (27) instead of the transfer error assumption (22).

To upper bound $|①|$, by Cauchy-Schwartz's inequality, we get

$$
\begin{aligned}
|①| &\leq \sum_{s \in \mathcal{S}} \sum_{a \in \mathcal{A}} d_s^{(k+1)} \pi_{s,a}^{(k+1)} \left| \phi_{s,a}^\top \left( w^{(k)} - w_\star^{(k)} \right) \right| \\
&\leq \sqrt{ \sum_{(s,a) \in \mathcal{S} \times \mathcal{A}} \frac{\left( d_s^{(k+1)} \right)^2 \left( \pi_{s,a}^{(k+1)} \right)^2}{\tilde{d}_{s,a}^{(k)}} \cdot \sum_{(s,a) \in \mathcal{S} \times \mathcal{A}} \tilde{d}_{s,a}^{(k)} \left( \phi_{s,a}^\top \left( w^{(k)} - w_\star^{(k)} \right) \right)^2 } \\
&\overset{(60)}{=} \sqrt{ \mathbb{E}_{(s,a) \sim \tilde{d}^{(k)}} \left[ \left( \frac{d_s^{(k+1)} \pi_{s,a}^{(k+1)}}{\tilde{d}_{s,a}^{(k)}} \right)^2 \right] \left\| w^{(k)} - w_\star^{(k)} \right\|_{\Sigma_{\tilde{d}^{(k)}}}^2 } \\
&\overset{(28)}{\leq} \sqrt{ C_\nu \left\| w^{(k)} - w_\star^{(k)} \right\|_{\Sigma_{\tilde{d}^{(k)}}}^2 } \\
&\overset{(62)}{\leq} \sqrt{ C_\nu \epsilon_{\text{stat}}^{(k)} }.
\end{aligned}
$$

Similar to $|①|$, by using Assumption 6 and Cauchy-Schwartz's inequality, and by simply replacing $\pi^{(k+1)}$ into $\pi^{(k)}$ or $\pi^*$ and replacing $d^{(k+1)}$ into $d^*$, we obtain the same upper bound of $|③|$, $|ⓐ|$ and $|ⓒ|$, that is

$$
|③|, |ⓐ|, |ⓒ| \leq \sqrt{ C_\nu \epsilon_{\text{stat}}^{(k)} }.
$$

Next, we define

$$
\epsilon_{\text{approx}}^{(k)} \overset{\text{def}}{=} L_Q(w_\star^{(k)}, \theta^{(k)}, \tilde{d}^{(k)})
$$

By Assumption 5, we know that

$$
\mathbb{E} \left[ \epsilon_{\text{approx}}^{(k)} \right] \leq \epsilon_{\text{approx}}.
$$

To upper bound $|②|$, by Cauchy-Schwartz's inequality, we have

$$
\begin{aligned}
|②| &\leq \sum_{s \in \mathcal{S}} \sum_{a \in \mathcal{A}} d_s^{(k+1)} \pi_{s,a}^{(k+1)} \left| \phi_{s,a}^\top w_\star^{(k)} - Q_{s,a}^{(k)} \right| \\
&\leq \sqrt{ \sum_{(s,a) \in \mathcal{S} \times \mathcal{A}} \frac{\left( d_s^{(k+1)} \right)^2 \left( \pi_{s,a}^{(k+1)} \right)^2}{\tilde{d}_{s,a}^{(k)}} \cdot \sum_{(s,a) \in \mathcal{S} \times \mathcal{A}} \tilde{d}_{s,a}^{(k)} \left( \phi_{s,a}^\top w_\star^{(k)} - Q_{s,a}^{(k)} \right)^2 } \\
&= \sqrt{ \mathbb{E}_{(s,a) \sim \tilde{d}^{(k)}} \left[ \left( \frac{d_s^{(k+1)} \pi_{s,a}^{(k+1)}}{\tilde{d}_{s,a}^{(k)}} \right)^2 \right] \cdot \epsilon_{\text{approx}}^{(k)} } \\
&\overset{(28)}{\leq} \sqrt{ C_\nu \epsilon_{\text{approx}}^{(k)} }.
\end{aligned}
$$

Similar to $|②|$, by using Assumption 5 and Cauchy-Schwartz's inequality, and by simply replacing $\pi^{(k+1)}$ into $\pi^{(k)}$ or $\pi^*$ and replacing $d^{(k+1)}$ into $d^*$, we obtain the same upper bound for $|④|$, $|ⓑ|$ and $|ⓓ|$, that is

$$
|④|, |ⓑ|, |ⓓ| \leq \sqrt{ C_\nu \epsilon_{\text{approx}}^{(k)} }.
$$

Consequently, plugging all these upper bounds into (49) leads to the following recurrent inequality

$$\vartheta_\rho \left( \delta_{k+1} - \delta_k \right) + \delta_k \leq \frac{D_k^*}{(1-\gamma)\eta_k} - \frac{D_{k+1}^*}{(1-\gamma)\eta_k} + \frac{2\sqrt{C_\nu}\left(\vartheta_\rho+1\right)}{1-\gamma} \left( \sqrt{\epsilon_{\text{stat}}^{(k)}} + \sqrt{\epsilon_{\text{approx}}^{(k)}} \right).$$

By using the same increasing step size as in Theorem 5 and following the same arguments in the proof of Theorem 5 after (73), we obtain the final performance bound with the linear convergence rate

$$\mathbb{E}\left[ V_\rho(\pi^{(k)}) \right] - V_\rho(\pi^*) \leq \left( 1 - \frac{1}{\vartheta_\rho} \right)^k \frac{2}{1-\gamma} + \frac{2\sqrt{C_\nu}\left(\vartheta_\rho+1\right)}{1-\gamma} \left( \sqrt{\epsilon_{\text{stat}}} + \sqrt{\epsilon_{\text{approx}}} \right).$$

$\square$

## D.5 PROOF OF COROLLARY 1

In this section, we aim to prove Corollary 1 presented in Section 4.3.

**Corollary 2** (Corollary 1). *Consider the setting of Theorem 3. Suppose that the sample-based Q-NPG Algorithm 2 is run for $K$ iterations, with $T$ gradient steps of Q-NPG-SGD (Algorithm 6) per iteration. Furthermore, suppose that for all $(s,a) \in \mathcal{S} \times \mathcal{A}$, we have $\|\phi_{s,a}\| \leq B$ with $B > 0$, and we choose the step size $\alpha = \frac{1}{2B^2}$ and the initialization $w_0 = 0$ for Q-NPG-SGD. If for all $\theta \in \mathbb{R}^m$, the covariance matrix of the feature map followed by the initial state-action distribution $\nu$ satisfies*

$$\mathbb{E}_{(s,a)\sim\nu}\left[ \phi_{s,a}\phi_{s,a}^\top \right] \overset{(23)}{=} \Sigma_\nu \geq \mu \mathbf{I}_m,$$

*where $\mathbf{I}_m \in \mathbb{R}^{m\times m}$ is the identity matrix and $\mu > 0$, then*

$$\mathbb{E}[V_\rho(\pi^{(K)})] - V_\rho(\pi^*) \leq \left( 1 - \frac{1}{\vartheta_\rho} \right)^K \frac{2}{1-\gamma} + \frac{2(\vartheta_\rho+1)\sqrt{C_\nu \epsilon_{\text{approx}}}}{1-\gamma}$$
$$+ \frac{4\sqrt{C_\nu}(\vartheta_\rho+1)}{(1-\gamma)^3\sqrt{T}} \left( \frac{B^2}{\mu}(\sqrt{2m}+1) + (1-\gamma)\sqrt{2m} \right).$$

In order to better understand our proof, we first identify an issue appeared in the sample complexity analysis of Q-NPG in Agarwal et al. (2021, Corollay 26). Agarwal et al. (2021) adopts the optimization results of Shalev-Shwartz & Ben-David (2014, Theorem 14.8) where the stochastic gradient $\widehat{\nabla} L_Q(w, \theta, \tilde{d}^\theta)$ in (47) needs to be bounded [6]. However, although they consider a projection step for the iterate $w_t$ and assume that the feature map $\phi_{s,a}$ is bounded, $\widehat{\nabla} L_Q(w, \theta, \tilde{d}^\theta)$ is still not guaranteed to be bounded. Indeed, recall the stochastic gradient of the function $L_Q$ in (47)

$$\widehat{\nabla}_w L_Q(w, \theta, \tilde{d}^\theta) = 2\left( w^\top \phi_{s,a} - \widehat{Q}_{s,a}(\theta) \right) \phi_{s,a}.$$

They incorrectly use the argument that $w, \phi_{s,a}$ and $\widehat{Q}_{s,a}(\theta)$ are bounded to imply that $\left\| \widehat{\nabla}_w L_Q(w, \theta, \tilde{d}^\theta) \right\|$ is bounded. In fact, $\widehat{Q}_{s,a}(\theta)$ can be unbounded even though $\mathbb{E}\left[ \widehat{Q}_{s,a}(\theta) \right] = Q_{s,a}(\theta) \in \left[ 0, \frac{1}{1-\gamma} \right]$ is bounded. To see this, we can rewrite $\widehat{Q}_{s,a}(\theta)$ from (43) as

$$\widehat{Q}_{s,a}(\theta) = \sum_{t=0}^{H} c(s_t, a_t),$$

with $(s_0, a_0) = (s,a) \sim \tilde{d}^\theta$ and $H$ is the length of the sampled trajectory for estimating $Q_{s,a}(\theta)$ in Algorithm 3. From Algorithm 3 and from the proof of Lemma 4, we know that the probability of $H = k+1$ is that

$$\Pr(H = k+1) = (1-\gamma)\gamma^k.$$

So, with exponentially decreasing low probability, $H$ can be unbounded. Consequently, $|\widehat{Q}_{s,a}(\theta)|$ upper bounded by $H$ is not guaranteed to be bounded.

---

[6] We are aware that Agarwal et al. (2021, Corollary 6.10) also use Bach & Moulines (2013, Theorem 1) in an early version https://arxiv.org/pdf/1908.00261v2.pdf to obtain $\epsilon_{\text{stat}} = \mathcal{O}(1/T)$. With further regularity (31), Agarwal et al. (2021) mentioned that $\epsilon_{\text{stat}} = \mathcal{O}(1/T)$ can also be achieved through Hsu et al. (2012, Theorem 16).

**Proof sketch.** Instead, we adopt the optimization results of Bach & Moulines (2013, Theorem 1) (see also Theorem 12), which does not require the boundedness of the stochastic gradient. However, in our following proof, we can verify that $\mathbb{E}\left[\widehat{Q}_{s,a}(\theta)^2\right]$ is bounded even though $\widehat{Q}_{s,a}(\theta)$ is unbounded. As to verify the condition (vi) in Theorem 12 in our proof, i.e., the covariance of the stochastic gradient at the optimum is upper bounded by the covariance of the feature map up to a finite constant, we use a conditional expectation argument to separate the correlated random variables $(s, a) \sim \tilde{d}^\theta$ and $\widehat{Q}_{s,a}(\theta)$ appeared in the stochastic gradient.

**Non-singularity of the covariance matrix $\Sigma_\nu$.** As for the condition (31), it is shown in Cayci et al. (2021, Proposition 3) that with $\nu$ chosen as uniform distribution over $\mathcal{S} \times \mathcal{A}$ and $\phi_{s,a} \sim \mathcal{N}(0, \mathbf{I}_m)$ sampled as Gaussian random features, (31) is guaranteed with high probability. More generally, with $m \ll |\mathcal{S}||\mathcal{A}|$, it is easy to find $m$ linearly independent $\phi_{s,a}$ among all $|\mathcal{S}||\mathcal{A}|$ features such that the covariance matrix $\Sigma_\nu$ has full rank.

*Proof.* From Theorem 3, it remains to upper bound the statistical error $\sqrt{\epsilon_{\text{stat}}}$ produced from the Q-NPG-SGD procedure (Algorithm 6) for each iteration $k$. We suppress the superscript $(k)$. Let $w_{\text{out}}$ be the output of $T$ steps Q-NPG-SGD with the constant step size $\frac{1}{2B^2}$ and the initialization $w_0 = 0$, and let $w_\star \in \arg\min_w L_Q(w, \theta, \tilde{d}^\theta)$ be the exact minimizer. To upper bound $\epsilon_{\text{stat}}$ from (19), we aim to apply the standard analysis for the averaged SGD, i.e., Theorem 12. Now we verify all the assumptions in order for Q-NPG-SGD.

First, (i) is verified by considering the Euclidean space $\mathcal{H} = \mathbb{R}^m$.

The observations $\left(\phi_{s,a},\ \widehat{Q}_{s,a}(\theta)\phi_{s,a}\right) \in \mathbb{R}^m \times \mathbb{R}^m$ are independent and identically distributed, sampled from Algorithm 3. Thus, (ii) is verified with $x_n = \phi_{s,a} \in \mathbb{R}^m$ and $z_n = \widehat{Q}_{s,a}(\theta)\phi_{s,a} \in \mathbb{R}^m$.

As the feature map $\|\phi_{s,a}\| \leq B$, we have $\mathbb{E}\left[\|\phi_{s,a}\|^2\right]$ finite. From (31), we know that the covariance $\mathbb{E}\left[\phi_{s,a}\phi_{s,a}^\top\right]$ is invertible. To verify (iii), it remains to verify that $\mathbb{E}\left[\left\|\widehat{Q}_{s,a}(\theta)\phi_{s,a}\right\|^2\right]$ is finite. Indeed, by using $\|\phi_{s,a}\| \leq B$, we have

$$\mathbb{E}\left[\left\|\widehat{Q}_{s,a}(\theta)\phi_{s,a}\right\|^2\right] \leq B^2 \mathbb{E}\left[\widehat{Q}_{s,a}(\theta)^2\right].$$

Thus, it remains to show $\mathbb{E}\left[\left(\widehat{Q}_{s,a}(\theta)\right)^2\right]$ finite for (iii). From (43), we rewrite $\widehat{Q}_{s,a}(\theta)$ as

$$\widehat{Q}_{s,a}(\theta) = \sum_{t=0}^{H} c(s_t, a_t),$$

with $(s_0, a_0) = (s, a) \sim \tilde{d}^\theta$ and $H$ is the length of the trajectory for estimating $Q_{s,a}(\theta)$. Thus, (iii) is verified as the variance of $\widehat{Q}_{s,a}(\theta)$ is upper bounded by

$$\mathbb{E}\left[\left(\widehat{Q}_{s,a}(\theta)\right)^2\right] = \mathbb{E}_{(s,a)\sim\tilde{d}^\theta}\left[\sum_{k=0}^{\infty} \Pr(H=k)\mathbb{E}\left[\left(\sum_{t=0}^{k} c(s_t, a_t)\right)^2 \mid H=k, s_0=s, a_0=a\right]\right]$$

$$= \mathbb{E}_{(s,a)\sim\tilde{d}^\theta}\left[(1-\gamma)\sum_{k=0}^{\infty} \gamma^k \mathbb{E}\left[\left(\sum_{t=0}^{k} c(s_t, a_t)\right)^2 \mid H=k, s_0=s, a_0=a\right]\right]$$

$$\leq \mathbb{E}_{(s,a)\sim\tilde{d}^\theta}\left[(1-\gamma)\sum_{k=0}^{\infty} \gamma^k (k+1)^2\right] \leq \frac{2}{(1-\gamma)^2}, \tag{74}$$

where the first inequality is obtained as $|c(s_t, a_t)| \in [0, 1]$ for all $(s_t, a_t) \in \mathcal{S} \times \mathcal{A}$.

Next, we introduce the residual

$$\xi \stackrel{\text{def}}{=} \left(\widehat{Q}_{s,a}(\theta) - w_\star^\top \phi_{s,a}\right)\phi_{s,a} \stackrel{(47)}{=} \frac{1}{2}\widehat{\nabla}_w L_Q(w_\star, \theta, \tilde{d}^\theta). \tag{75}$$

From Lemma 7, we know that

$$\mathbb{E}\left[\widehat{\nabla}_w L_Q(w_\star, \theta, \tilde{d}^\theta)\right] = \nabla_w L_Q(w_\star, \theta, \tilde{d}^\theta).$$

So, we have that

$$\mathbb{E}\left[\xi\right] = \frac{1}{2}\nabla_w L_Q(w_\star, \theta, \tilde{d}^\theta) = 0,$$

where the last equality is obtained as $w_\star$ is the exact minimizer of the loss function $L_Q$. Thus, (iv) is verified with that $f$ is $\frac{1}{2}L_Q$, $\xi_n$ is $\xi$ and $\theta$ is $w$ in our context.

From Q-NPG-SGD update 47, we have (v) verified with step size $\alpha/2$ in our context.

Finally, for (vi), from the boundedness of the feature map $\|\phi_{s,a}\| \le B$, we take $R = B$ such that $\mathbb{E}\left[\|\phi_{s,a}\|^2 \phi_{s,a}\phi_{s,a}^\top\right] \le B^2 \mathbb{E}\left[\phi_{s,a}\phi_{s,a}^\top\right]$. It remains to find $\sigma > 0$ such that

$$\mathbb{E}\left[\xi\xi^\top\right] \le \sigma^2 \mathbb{E}\left[\phi_{s,a}\phi_{s,a}^\top\right].$$

We rewrite the covariance of $\xi$ as

$$\mathbb{E}\left[\xi\xi^\top\right] \overset{(75)}{=} \mathbb{E}\left[\left(\widehat{Q}_{s,a}(\theta) - w_\star^\top \phi_{s,a}\right)^2 \phi_{s,a}\phi_{s,a}^\top\right]$$

$$= \mathbb{E}_{(s,a)\sim\tilde{d}^\theta}\left[\left(\widehat{Q}_{s,a}(\theta) - w_\star^\top \phi_{s,a}\right)^2 \phi_{s,a}\phi_{s,a}^\top \mid s, a\right]$$

$$= \mathbb{E}_{(s,a)\sim\tilde{d}^\theta}\left[\mathbb{E}\left[\left(\widehat{Q}_{s,a}(\theta) - w_\star^\top \phi_{s,a}\right)^2 \mid s, a\right] \phi_{s,a}\phi_{s,a}^\top\right].$$

Thus, it suffices to find $\sigma > 0$ such that

$$\mathbb{E}\left[\left(\widehat{Q}_{s,a}(\theta) - w_\star^\top \phi_{s,a}\right)^2 \mid s, a\right] = \mathbb{E}\left[\left(\widehat{Q}_{s,a}(\theta)\right)^2 \mid s, a\right] - 2Q_{s,a}(\theta)w_\star^\top \phi_{s,a} + \left(w_\star^\top \phi_{s,a}\right)^2 \le \sigma^2 \tag{76}$$

for all $(s, a) \in \mathcal{S} \times \mathcal{A}$ to verify (vi). Besides, we know that

$$\mathbb{E}\left[\left(\widehat{Q}_{s,a}(\theta)\right)^2 \mid s, a\right] \overset{(74)}{\le} \frac{2}{(1-\gamma)^2}.$$

We also know that $|Q_{s,a}(\theta)| \le \frac{1}{1-\gamma}$ and $\|\phi_{s,a}\| \le B$. Now we need to bound $\|w_\star\|$. Again, since $w_\star$ is the exact minimizer, we have $\nabla_w L_Q(w_\star, \theta, \tilde{d}^\theta) = 0$. That is

$$\mathbb{E}_{(s,a)\sim\tilde{d}^\theta}\left[\left(w_\star^\top \phi_{s,a} - Q_{s,a}(\theta)\right)\phi_{s,a}\right] = 0,$$

which implies

$$w_\star = \left(\mathbb{E}_{(s,a)\sim\tilde{d}^\theta}\left[\phi_{s,a}\phi_{s,a}^\top\right]\right)^\dagger \mathbb{E}_{(s,a)\sim\tilde{d}^\theta}\left[Q_{s,a}(\theta)\phi_{s,a}\right]$$

$$\overset{(5)}{\le} \frac{1}{1-\gamma}\left(\mathbb{E}_{(s,a)\sim\nu}\left[\phi_{s,a}\phi_{s,a}^\top\right]\right)^\dagger \mathbb{E}_{(s,a)\sim\tilde{d}^\theta}\left[Q_{s,a}(\theta)\phi_{s,a}\right].$$

By the boundness of the feature map $\|\phi_{s,a}\| \le B$ and the Q-function $|Q_{s,a}(\theta)| \le \frac{1}{1-\gamma}$, and the condition (31), we have the minimizer $w_\star$ bounded by

$$\|w_\star\| \overset{(31)}{\le} \frac{B}{\mu(1-\gamma)^2}.$$

By using the upper bounds of $\mathbb{E}\left[\left(\widehat{Q}_{s,a}(\theta)\right)^2 \mid s, a\right]$, $|Q_{s,a}(\theta)|$, $\|w_\star\|$ and $\|\phi_{s,a}\|$, the left hand side of (76) can be upper bounded by

$$\mathbb{E}\left[\left(\widehat{Q}_{s,a}(\theta) - w_\star^\top \phi_{s,a}\right)^2 \mid s, a\right] \le \frac{2}{(1-\gamma)^2} + \frac{2B^2}{\mu(1-\gamma)^3} + \frac{B^4}{\mu^2(1-\gamma)^4}$$

$$= \frac{1}{(1-\gamma)^2}\left(\left(\frac{B^2}{\mu(1-\gamma)} + 1\right)^2 + 1\right)$$

$$\le \frac{2}{(1-\gamma)^2}\left(\frac{B^2}{\mu(1-\gamma)} + 1\right)^2.$$

Thus, in order to satisfy (76), we choose

$$\sigma = \frac{\sqrt{2}}{1-\gamma}\left(\frac{B^2}{\mu(1-\gamma)}+1\right).$$

Now all the conditions (i) - (vi) in Theorem 12 are verified. With step size $\alpha = \frac{1}{2B^2}$, the initialization $w_0 = 0$ and $T$ steps of Q-NPG-SGD updates (47), we have

$$\mathbb{E}\left[L_Q(w_{\text{out}}, \theta, \tilde{d}^\theta)\right] - L_Q(w_\star, \theta, \tilde{d}^\theta) \leq \frac{4}{T}\left(\sigma\sqrt{m} + B\|w_\star\|\right)^2$$

$$\leq \frac{4}{T}\left(\frac{\sqrt{2m}}{1-\gamma}\left(\frac{B^2}{\mu(1-\gamma)}+1\right) + \frac{B^2}{\mu(1-\gamma)^2}\right)^2.$$

Consequently, Assumption 1 is verified by

$$\sqrt{\epsilon_{\text{stat}}} \leq \frac{2}{(1-\gamma)\sqrt{T}}\left(\frac{B^2}{\mu(1-\gamma)}\left(\sqrt{2m}+1\right) + \sqrt{2m}\right).$$

The proof is completed by replacing the above upper bound of $\sqrt{\epsilon_{\text{stat}}}$ in the results of Theorem 3. $\square$

# E  PROOF OF SECTION 5

## E.1  THE ONE STEP NPG LEMMA

To prove Theorem 4 and the sublinear convergence result of Theorem 9 further in Appendix F, we start from providing the one step analysis of the NPG update.

**Lemma 9** (One step NPG lemma). *Fix a state distribution $\rho$; an initial state-action distribution $\nu$; an arbitrary comparator policy $\pi^*$. At the $k$-th iteration, let $w_\star^{(k)} \in \text{argmin}_w L_A(w, \theta^{(k)}, \tilde{d}^{(k)})$ denote the exact minimizer. Consider the $w^{(k)}$ and $\pi^{(k)}$ NPG iterates given in (32) and (17) respectively. Note*

$$\epsilon_{\text{stat}}^{(k)} \stackrel{def}{=} L_A(w^{(k)}, \theta^{(k)}, \tilde{d}^{(k)}) - L_A(w_\star^{(k)}, \theta^{(k)}, \tilde{d}^{(k)}), \tag{77}$$

$$\epsilon_{\text{approx}}^{(k)} \stackrel{def}{=} L_A(w_\star^{(k)}, \theta^{(k)}, \tilde{d}^{(k)}), \tag{78}$$

$$\delta_k \stackrel{def}{=} V_\rho^{(k)} - V_\rho(\pi^*).$$

*If Assumptions 7, 8 and 9 hold for all $k \geq 0$, then we have that*

$$\vartheta_\rho\left(\delta_{k+1} - \delta_k\right) + \delta_k \leq \frac{D_k^*}{(1-\gamma)\eta_k} - \frac{D_{k+1}^*}{(1-\gamma)\eta_k} + \frac{\sqrt{C_\nu}\left(\vartheta_\rho+1\right)}{1-\gamma}\left(\sqrt{\epsilon_{\text{stat}}^{(k)}} + \sqrt{\epsilon_{\text{approx}}^{(k)}}\right). \tag{79}$$

*Proof.* From the three-point descent lemma (Lemma 11) and (17), we obtain that for any $p \in \Delta(\mathcal{A})$, we have

$$\eta_k\left\langle\bar{\Phi}_s^{(k)}w^{(k)}, \pi_s^{(k+1)}\right\rangle + D(\pi_s^{(k+1)}, \pi_s^{(k)}) \leq \eta_k\left\langle\bar{\Phi}_s^{(k)}w^{(k)}, p\right\rangle + D(p, \pi_s^{(k)}) - D(p, \pi_s^{(k+1)}).$$

Rearranging terms and dividing both sides by $\eta_k$, we get

$$\left\langle\bar{\Phi}_s^{(k)}w^{(k)}, \pi_s^{(k+1)} - p\right\rangle + \frac{1}{\eta_k}D(\pi_s^{(k+1)}, \pi_s^{(k)}) \leq \frac{1}{\eta_k}D(p, \pi_s^{(k)}) - \frac{1}{\eta_k}D(p, \pi_s^{(k+1)}).$$

Letting $p = \pi_s^{(k)}$ and knowing that

$$\left\langle\bar{\Phi}_s^{(k)}w^{(k)}, \pi_s^{(k)}\right\rangle = 0 \qquad \text{for all } k \geq 0,$$

which is due to (12), we have

$$\left\langle\bar{\Phi}_s^{(k)}w^{(k)}, \pi_s^{(k+1)}\right\rangle \leq -\frac{1}{\eta_k}D(\pi_s^{(k+1)}, \pi_s^{(k)}) - \frac{1}{\eta_k}D(\pi_s^{(k)}, \pi_s^{(k+1)}) \leq 0. \tag{80}$$

Letting $p = \pi_s^*$ yields

$$\left\langle \bar{\Phi}_s^{(k)} w^{(k)}, \pi_s^{(k+1)} - \pi_s^* \right\rangle \leq \frac{1}{\eta_k} D(\pi_s^*, \pi_s^{(k)}) - \frac{1}{\eta_k} D(\pi_s^*, \pi_s^{(k+1)}).$$

Note that we dropped the nonnegative term $\frac{1}{\eta_k} D(\pi_s^{(k+1)}, \pi_s^{(k)})$ on the left hand side to the inequality.

Taking expectation with respect to the distribution $d^*$, we have

$$\mathbb{E}_{s \sim d^*} \left[ \left\langle \bar{\Phi}_s^{(k)} w^{(k)}, \pi_s^{(k+1)} \right\rangle \right] - \mathbb{E}_{s \sim d^*} \left[ \left\langle \bar{\Phi}_s^{(k)} w^{(k)}, \pi_s^* \right\rangle \right] \leq \frac{1}{\eta_k} D_k^* - \frac{1}{\eta_k} D_{k+1}^*. \qquad (81)$$

For the first expectation in (81), we have

$$\mathbb{E}_{s \sim d^*} \left[ \left\langle \bar{\Phi}_s^{(k)} w^{(k)}, \pi_s^{(k+1)} \right\rangle \right]$$

$$= \sum_{s \in \mathcal{S}} d_s^* \left\langle \bar{\Phi}_s^{(k)} w^{(k)}, \pi_s^{(k+1)} \right\rangle$$

$$= \sum_{s \in \mathcal{S}} \frac{d_s^*}{d_s^{(k+1)}} d_s^{(k+1)} \left\langle \bar{\Phi}_s^{(k)} w^{(k)}, \pi_s^{(k+1)} \right\rangle$$

$$\overset{(20)+(80)}{\geq} \vartheta_{k+1} \sum_{s \in \mathcal{S}} d_s^{(k+1)} \left\langle \bar{\Phi}_s^{(k)} w^{(k)}, \pi_s^{(k+1)} \right\rangle$$

$$\overset{(20)+(80)}{\geq} \vartheta_\rho \sum_{s \in \mathcal{S}} d_s^{(k+1)} \left\langle \bar{\Phi}_s^{(k)} w^{(k)}, \pi_s^{(k+1)} \right\rangle$$

$$= \vartheta_\rho \mathbb{E}_{(s,a) \sim \bar{d}^{(k+1)}} \left[ (\bar{\phi}_{s,a}^{(k)})^\top w^{(k)} \right]$$

$$= \vartheta_\rho \mathbb{E}_{(s,a) \sim \bar{d}^{(k+1)}} \left[ A_{s,a}^{(k)} \right] + \vartheta_\rho \mathbb{E}_{(s,a) \sim \bar{d}^{(k+1)}} \left[ (\bar{\phi}_{s,a}^{(k)})^\top w^{(k)} - A_{s,a}^{(k)} \right]$$

$$= \vartheta_\rho (1 - \gamma) \left( V_\rho^{(k+1)} - V_\rho^{(k)} \right) + \vartheta_\rho \mathbb{E}_{(s,a) \sim \bar{d}^{(k+1)}} \left[ (\bar{\phi}_{s,a}^{(k)})^\top w^{(k)} - A_{s,a}^{(k)} \right], \qquad (82)$$

where the last line is obtained by the performance difference lemma (41), and we use the shorthand $\bar{\phi}_{s,a}^{(k)}$ as $\bar{\phi}_{s,a}(\theta^{(k)})$.

The second term of (82) can be lower bounded. To do it, we first decompose it into two terms. That is,

$$\mathbb{E}_{(s,a) \sim \bar{d}^{(k+1)}} \left[ (\bar{\phi}_{s,a}^{(k)})^\top w^{(k)} - A_{s,a}^{(k)} \right] = \underbrace{\mathbb{E}_{(s,a) \sim \bar{d}^{(k+1)}} \left[ (\bar{\phi}_{s,a}^{(k)})^\top (w^{(k)} - w_\star^{(k)}) \right]}_{\textcircled{1}}$$

$$+ \underbrace{\mathbb{E}_{(s,a) \sim \bar{d}^{(k+1)}} \left[ (\bar{\phi}_{s,a}^{(k)})^\top w_\star^{(k)} - A_{s,a}^{(k)} \right]}_{\textcircled{2}}. \qquad (83)$$

We will upper bound the absolute values of the above two terms $|\textcircled{1}|$ and $|\textcircled{2}|$ separately. More precisely, similar to the proof of Theorem 3, we will upper bound the first term $|\textcircled{1}|$ by the statistical error assumption (33) and upper bound the second term $|\textcircled{2}|$ by using the approximation error assumption (34).

To upper bound $\textcircled{1}$, we first define the following covariance matrix of the centered feature map

$$\Sigma_{\tilde{d}^{(k)}}^{(k)} \overset{\text{def}}{=} \mathbb{E}_{(s,a) \sim \tilde{d}^{(k)}} \left[ \bar{\phi}_{s,a}^{(k)} (\bar{\phi}_{s,a}^{(k)})^\top \right]. \qquad (84)$$

Here we use the superscript $(k)$ for $\Sigma_{\tilde{d}^{(k)}}^{(k)}$ to distinguish the covariance matrix of the feature map $\Sigma_{\tilde{d}^{(k)}}$ defined in (60) in the proof of Theorem 1, as the centered feature map $\bar{\phi}_{s,a}^{(k)}$ depends on the iterates $\theta^{(k)}$.

By Cauchy-Schwartz's inequality, we have

$$
\begin{aligned}
|\textcircled{1}| \quad &\leq \quad \sum_{(s,a)\in\mathcal{S}\times\mathcal{A}} \bar{d}_{s,a}^{(k+1)} \left|(\bar{\phi}_{s,a}^{(k)})^{\top}(w^{(k)} - w_{\star}^{(k)})\right| \\
&\leq \quad \sqrt{\sum_{(s,a)\in\mathcal{S}\times\mathcal{A}} \frac{\left(\bar{d}_{s,a}^{(k+1)}\right)^2}{\tilde{d}_{s,a}^{(k)}} \sum_{(s,a)\in\mathcal{S}\times\mathcal{A}} \tilde{d}_{s,a}^{(k)} \left((\bar{\phi}_{s,a}^{(k)})^{\top}(w^{(k)} - w_{\star}^{(k)})\right)^2} \\
&\overset{(84)}{=} \quad \sqrt{\mathbb{E}_{(s,a)\sim\tilde{d}^{(k)}}\left[\left(\frac{\bar{d}_{s,a}^{(k+1)}}{\tilde{d}_{s,a}^{(k)}}\right)^2\right] \left\|w^{(k)} - w_{\star}^{(k)}\right\|_{\Sigma_{\tilde{d}^{(k)}}^{(k)}}^2}.
\end{aligned}
$$

By further using the concentrability assumption 9, we have

$$
\begin{aligned}
|\textcircled{1}| \quad &\overset{(35)}{\leq} \quad \sqrt{C_{\nu}\left\|w^{(k)} - w_{\star}^{(k)}\right\|_{\Sigma_{\tilde{d}^{(k)}}^{(k)}}^2} \\
&\leq \quad \sqrt{C_{\nu}\left(L_A(w^{(k)}, \theta^{(k)}, \tilde{d}^{(k)}) - L_A(w_{\star}^{(k)}, \theta^{(k)}, \tilde{d}^{(k)})\right)} \quad &(85) \\
&\overset{(77)}{=} \quad \sqrt{C_{\nu}\epsilon_{\text{stat}}^{(k)}}, \quad &(86)
\end{aligned}
$$

where (85) uses that $w_{\star}^{(k)}$ is a minimizer of $L_A$ and $w_{\star}^{(k)}$ is feasible (see the same arguments of (62) in the proof of Theorem 1).

For the second term $|\textcircled{2}|$ in (83), by Cauchy-Schwartz's inequality, we have

$$
\begin{aligned}
|\textcircled{2}| \quad &\leq \quad \sum_{(s,a)\in\mathcal{S}\times\mathcal{A}} \bar{d}_{s,a}^{(k+1)} \left|(\bar{\phi}_{s,a}^{(k)})^{\top}w_{\star}^{(k)} - A_{s,a}^{(k)}\right| \\
&\leq \quad \sqrt{\sum_{(s,a)\in\mathcal{S}\times\mathcal{A}} \frac{\left(\bar{d}_{s,a}^{(k+1)}\right)^2}{\tilde{d}_{s,a}^{(k)}} \sum_{(s,a)\in\mathcal{S}\times\mathcal{A}} \tilde{d}_{s,a}^{(k)} \left((\bar{\phi}_{s,a}^{(k)})^{\top}w_{\star}^{(k)} - A_{s,a}^{(k)}\right)^2} \\
&= \quad \sqrt{\mathbb{E}_{(s,a)\sim\tilde{d}^{(k)}}\left[\left(\frac{\bar{d}_{s,a}^{(k+1)}}{\tilde{d}_{s,a}^{(k)}}\right)^2\right] L_A(w_{\star}^{(k)}, \theta^{(k)}, \tilde{d}^{(k)})} \\
&\overset{(35)+(78)}{\leq} \quad \sqrt{C_{\nu}\epsilon_{\text{approx}}^{(k)}}. \quad &(87)
\end{aligned}
$$

Plugging (86) and (87) into (82) yields

$$
\mathbb{E}_{s\sim d^*}\left[\left\langle\bar{\Phi}_s^{(k)}w^{(k)}, \pi_s^{(k+1)}\right\rangle\right] \geq \vartheta_{\rho}(1-\gamma)\left(V_{\rho}^{(k+1)} - V_{\rho}^{(k)}\right) - \vartheta_{\rho}\sqrt{C_{\nu}}\left(\sqrt{\epsilon_{\text{stat}}^{(k)}} + \sqrt{\epsilon_{\text{approx}}^{(k)}}\right). \tag{88}
$$

Now for the second expectation in (81), by using the performance difference lemma (41) in Lemma 3, we have

$$
\begin{aligned}
-\mathbb{E}_{s\sim d^*}\left[\left\langle\bar{\Phi}_s^{(k)}w^{(k)}, \pi_s^*\right\rangle\right] &= -\mathbb{E}_{(s,a)\sim\bar{d}^{\pi^*}}\left[A_{s,a}^{(k)}\right] + \mathbb{E}_{(s,a)\sim\bar{d}^{\pi^*}}\left[A_{s,a}^{(k)} - (\bar{\phi}_{s,a}^{(k)})^{\top}w^{(k)}\right] \\
&= (1-\gamma)\left(V_{\rho}^{(k)} - V_{\rho}(\pi^*)\right) + \mathbb{E}_{(s,a)\sim\bar{d}^{\pi^*}}\left[A_{s,a}^{(k)} - (\bar{\phi}_{s,a}^{(k)})^{\top}w^{(k)}\right]. \tag{89}
\end{aligned}
$$

The second term of (89) can be lower bounded. We first decompose it into two terms. That is,

$$\mathbb{E}_{(s,a)\sim\bar{d}^{\pi^*}}\left[A_{s,a}^{(k)} - (\bar{\phi}_{s,a}^{(k)})^\top w^{(k)}\right] = \underbrace{\mathbb{E}_{(s,a)\sim\bar{d}^{\pi^*}}\left[A_{s,a}^{(k)} - (\bar{\phi}_{s,a}^{(k)})^\top w_\star^{(k)}\right]}_{\text{ⓐ}}$$

$$+ \underbrace{\mathbb{E}_{(s,a)\sim\bar{d}^{\pi^*}}\left[(\bar{\phi}_{s,a}^{(k)})^\top (w_\star^{(k)} - w^{(k)})\right]}_{\text{ⓑ}}. \tag{90}$$

Now we will upper bound the absolute values of the above two terms $|\text{ⓐ}|$ and $|\text{ⓑ}|$ separately.
For the first one $|\text{ⓐ}|$, by Cauchy-Schwartz's inequality, we have

$$|\text{ⓐ}| \quad \leq \quad \sum_{(s,a)\in\mathcal{S}\times\mathcal{A}} \bar{d}_{s,a}^{\pi^*} \left|A_{s,a}^{(k)} - (\bar{\phi}_{s,a}^{(k)})^\top w_\star^{(k)}\right|$$

$$\leq \quad \sqrt{\sum_{(s,a)\in\mathcal{S}\times\mathcal{A}} \frac{\left(\bar{d}_{s,a}^{\pi^*}\right)^2}{\tilde{d}_{s,a}^{(k)}} \sum_{(s,a)\in\mathcal{S}\times\mathcal{A}} \tilde{d}_{s,a}^{(k)} \left((\bar{\phi}_{s,a}^{(k)})^\top w_\star^{(k)} - A_{s,a}^{(k)}\right)^2}$$

$$= \quad \sqrt{\mathbb{E}_{(s,a)\sim\tilde{d}^{(k)}}\left[\left(\frac{\bar{d}_{s,a}^{\pi^*}}{\tilde{d}_{s,a}^{(k)}}\right)^2\right] L_A(w_\star^{(k)}, \theta^{(k)}, \tilde{d}^{(k)})}$$

$$\overset{(35)+(78)}{\leq} \quad \sqrt{C_\nu \epsilon_{\text{approx}}^{(k)}}. \tag{91}$$

For the second term $|\text{ⓑ}|$ in (90), by Cauchy-Schwartz's inequality, we have

$$|\text{ⓑ}| \quad \leq \quad \sum_{(s,a)\in\mathcal{S}\times\mathcal{A}} \bar{d}_{s,a}^{\pi^*} \left|(\bar{\phi}_{s,a}^{(k)})^\top (w^{(k)} - w^{(k)})\right|$$

$$\leq \quad \sqrt{\sum_{(s,a)\in\mathcal{S}\times\mathcal{A}} \frac{\left(\bar{d}_{s,a}^{\pi^*}\right)^2}{\tilde{d}_{s,a}^{(k)}} \sum_{(s,a)\in\mathcal{S}\times\mathcal{A}} \tilde{d}_{s,a}^{(k)} \left((\bar{\phi}_{s,a}^{(k)})^\top (w^{(k)} - w_\star^{(k)})\right)^2}$$

$$\overset{(84)}{=} \quad \sqrt{\mathbb{E}_{(s,a)\sim\tilde{d}^{(k)}}\left[\left(\frac{\bar{d}_{s,a}^{\pi^*}}{\tilde{d}_{s,a}^{(k)}}\right)^2\right] \left\|w^{(k)} - w_\star^{(k)}\right\|_{\Sigma_{\tilde{d}^{(k)}}^{(k)}}^2}$$

$$\overset{(35)}{\leq} \quad \sqrt{C_\nu \left\|w^{(k)} - w_\star^{(k)}\right\|_{\Sigma_{\tilde{d}^{(k)}}^{(k)}}^2}$$

$$\overset{(85)}{\leq} \quad \sqrt{C_\nu \left(L_A(w^{(k)}, \theta^{(k)}, \tilde{d}^{(k)}) - L_A(w_\star^{(k)}, \theta^{(k)}, \tilde{d}^{(k)})\right)}$$

$$\overset{(77)}{=} \quad \sqrt{C_\nu \epsilon_{\text{stat}}^{(k)}}. \tag{92}$$

Thus, we lower bound (90) by

$$-\mathbb{E}_{s\sim d^*}\left[\left\langle \bar{\Phi}_s^{(k)} w^{(k)}, \pi_s^*\right\rangle\right] \overset{(91)+(92)}{\geq} (1-\gamma)\left(V_\rho^{(k)} - V_\rho(\pi^*)\right) - \sqrt{C_\nu}\left(\sqrt{\epsilon_{\text{stat}}^{(k)}} + \sqrt{\epsilon_{\text{approx}}^{(k)}}\right). \tag{93}$$

Substituting (88) and (93) into (81), dividing both side by $1 - \gamma$ and rearranging terms, we get

$$\vartheta_\rho\left(\delta_{k+1} - \delta_k\right) + \delta_k \leq \frac{D_k^*}{(1-\gamma)\eta_k} - \frac{D_{k+1}^*}{(1-\gamma)\eta_k} + \frac{\sqrt{C_\nu}\left(\vartheta_\rho + 1\right)}{1-\gamma}\left(\sqrt{\epsilon_{\text{stat}}^{(k)}} + \sqrt{\epsilon_{\text{approx}}^{(k)}}\right).$$

$$\square$$

### E.2 PROOF OF THEOREM 4

To prove Theorem 4, we prove the following general result with any initial policy by knowing that $D_0^* \leq \log |\mathcal{A}|$ when using the uniform initial policy in (48).

**Theorem 8.** *Fix a state distribution $\rho$, a state-action distribution $\nu$, and a comparator policy $\pi^*$. We consider the NPG method (32) with the step sizes satisfying $\eta_0 \geq \frac{1-\gamma}{\gamma} D_0^*$ and $\eta_{k+1} \geq \frac{1}{\gamma} \eta_k$. Suppose that Assumptions 7, 8 and 9 hold. Then we have for all $k \geq 0$,*

$$\mathbb{E}\left[V_\rho(\pi^{(k)})\right] - V_\rho(\pi^*) \leq \left(1 - \frac{1}{\vartheta_\rho}\right)^k \frac{2}{1-\gamma} + \frac{\sqrt{C_\nu} \, (\vartheta_\rho + 1)}{1-\gamma} \left(\sqrt{\epsilon_{\text{stat}}} + \sqrt{\epsilon_{\text{approx}}}\right).$$

*Proof.* From (79) in Lemma 9, by using the same increasing step size as in Theorem 5, i.e. $\eta_0 \geq \frac{1-\gamma}{\gamma} D_0^*$ and $\eta_{k+1} \geq \eta_k/\gamma$, and following the same arguments in the proof of Theorem 5 after (73), we obtain the final performance bound with the linear convergence rate

$$\mathbb{E}\left[V_\rho(\pi^{(k)})\right] - V_\rho(\pi^*) \leq \left(1 - \frac{1}{\vartheta_\rho}\right)^k \frac{2}{1-\gamma} + \frac{\sqrt{C_\nu} \, (\vartheta_\rho + 1)}{1-\gamma} \left(\sqrt{\epsilon_{\text{stat}}} + \sqrt{\epsilon_{\text{approx}}}\right).$$

$\square$

## F    SUBLINEAR CONVERGENCE OF NPG WITH CONSTANT STEP SIZE

In this section, we provide the sublinear convergence of NPG with arbitrary constant step size.

**Theorem 9.** *Fix a state distribution $\rho$, an state-action distribution $\nu$ and an optimal policy $\pi^*$. We consider the NPG method (32) with any constant step size $\eta_k = \eta > 0$. Suppose that Assumptions 7, 8 and 9 hold. Then we have for all $k \geq 0$,*

$$\frac{1}{k}\sum_{t=0}^{k-1} \mathbb{E}\left[V_\rho(\pi^{(t)})\right] - V_\rho(\pi^*) \leq \frac{1}{(1-\gamma)k}\left(\frac{D_0^*}{\eta} + 2\vartheta_\rho\right) + \frac{\sqrt{C_\nu} \, (\vartheta_\rho + 1)}{1-\gamma}\left(\sqrt{\epsilon_{\text{stat}}} + \sqrt{\epsilon_{\text{approx}}}\right).$$

Compared to Theorem 2, again here we require $\pi^*$ to be an optimal policy for the same reason as indicated in Footnote 5. Furthermore our sublinear convergence guarantees for both Q-NPG and NPG are the same. Compared to Theorem 29 in Agarwal et al. (2021), the main differences are also similar to those for Q-NPG as summarized right after Theorem 2: our convergence rate improves from $\mathcal{O}(1/\sqrt{k})$ to $\mathcal{O}(1/k)$; they use a diminishing step size of order $\mathcal{O}(1/\sqrt{k})$ while we can take any constant step size we want.

Despite the difference of using $\tilde{d}^{(k)}$ instead of $\bar{d}^{(k)}$ for the compatible function approximation $L_A(w^{(k)}, \theta^{(k)}, \tilde{d}^{(k)})$, notice that same sublinear convergence rate $\mathcal{O}(1/k)$ is established by Liu et al. (2020) for NPG with constant step size, while their step size is bounded by the inverse of a smoothness constant and they further require that the feature map is bounded and the Fisher information matrix (9) is strictly lower bounded for all parameters $\theta \in \mathbb{R}^m$ (see this condition later in (94)). With such additional conditions, we are able to provide a $\tilde{\mathcal{O}}(\frac{1}{(1-\gamma)^5 \epsilon^2})$ sample complexity result of NPG in Appendix G.

*Proof.* From (79) in Lemma 9 with the constant step size, we have

$$\vartheta_\rho \left(\delta_{k+1} - \delta_k\right) + \delta_k \leq \frac{D_k^*}{(1-\gamma)\eta} - \frac{D_{k+1}^*}{(1-\gamma)\eta} + \frac{\sqrt{C_\nu} \, (\vartheta_\rho + 1)}{1-\gamma}\left(\sqrt{\epsilon_{\text{stat}}^{(k)}} + \sqrt{\epsilon_{\text{approx}}^{(k)}}\right).$$

Taking the total expectation with respect to the randomness in the sequence of the iterates $w^{(0)}, \cdots, w^{(k-1)}$ yields

$$\vartheta_\rho \left(\mathbb{E}\left[\delta_{k+1}\right] - \mathbb{E}\left[\delta_k\right]\right) + \mathbb{E}\left[\delta_k\right] \quad \leq \quad \frac{\mathbb{E}\left[D_k^*\right]}{(1-\gamma)\eta} - \frac{\mathbb{E}\left[D_{k+1}^*\right]}{(1-\gamma)\eta}$$
$$+ \frac{\sqrt{C_\nu}\left(\vartheta_\rho + 1\right)}{1-\gamma}\left(\mathbb{E}\left[\sqrt{\epsilon_{\mathrm{stat}}^{(k)}}\right] + \mathbb{E}\left[\sqrt{\epsilon_{\mathrm{approx}}^{(k)}}\right]\right)$$
$$\leq \quad \frac{\mathbb{E}\left[D_k^*\right]}{(1-\gamma)\eta} - \frac{\mathbb{E}\left[D_{k+1}^*\right]}{(1-\gamma)\eta}$$
$$+ \frac{\sqrt{C_\nu}\left(\vartheta_\rho + 1\right)}{1-\gamma}\left(\sqrt{\mathbb{E}\left[\epsilon_{\mathrm{stat}}^{(k)}\right]} + \sqrt{\mathbb{E}\left[\epsilon_{\mathrm{approx}}^{(k)}\right]}\right)$$
$$\overset{(33)+(34)}{\leq} \quad \frac{\mathbb{E}\left[D_k^*\right]}{(1-\gamma)\eta} - \frac{\mathbb{E}\left[D_{k+1}^*\right]}{(1-\gamma)\eta}$$
$$+ \frac{\sqrt{C_\nu}\left(\vartheta_\rho + 1\right)}{1-\gamma}\left(\sqrt{\epsilon_{\mathrm{stat}}} + \sqrt{\epsilon_{\mathrm{approx}}}\right).$$

By summing up from $0$ to $k-1$, we get

$$\vartheta_\rho \mathbb{E}\left[\delta_k\right] + \sum_{t=0}^{k-1} \mathbb{E}\left[\delta_t\right] \leq \frac{D_0^*}{(1-\gamma)\eta} + \vartheta_\rho \delta_0 + k \cdot \frac{\sqrt{C_\nu}\left(\vartheta_\rho + 1\right)}{1-\gamma}\left(\sqrt{\epsilon_{\mathrm{stat}}} + \sqrt{\epsilon_{\mathrm{approx}}}\right).$$

Finally, dropping the positive term $\mathbb{E}\left[\delta_k\right]$ on the left hand side as $\pi^*$ is the optimal policy and dividing both side by $k$ yields

$$\frac{1}{k}\sum_{t=0}^{k-1} \mathbb{E}\left[V_\rho(\pi^{(t)})\right] - V_\rho(\pi^*) \leq \frac{D_0^*}{(1-\gamma)\eta k} + \frac{2\vartheta_\rho}{(1-\gamma)k} + \frac{\sqrt{C_\nu}\left(\vartheta_\rho + 1\right)}{1-\gamma}\left(\sqrt{\epsilon_{\mathrm{stat}}} + \sqrt{\epsilon_{\mathrm{approx}}}\right).$$

$\square$

## G   SAMPLE COMPLEXITY OF NPG

Combined with a regression solver, `NPG-SGD` in Algorithm 5, which uses a slight modification of `Q-NPG-SGD` for the unbiased gradient estimates of $L_A$, we consider a sampled-based NPG Algorithm 1 proposed in Appendix C and show its sample complexity result in the following corollary.

**Corollary 3.** *Consider the setting of Theorem 8. Suppose that the sample-based NPG Algorithm 1 is run for $K$ iterations, with $T$ gradient steps of `NPG-SGD` (Algorithm 5) per iteration. Furthermore, suppose that for all $(s,a) \in \mathcal{S} \times \mathcal{A}$, we have $\|\phi_{s,a}\| \leq B$ with $B > 0$, and we choose the step size $\alpha = \frac{1}{8B^2}$ and the initialization $w_0 = 0$ for `NPG-SGD`. If for all $\theta \in \mathbb{R}^m$, the covariance matrix of the centered feature map induced by the policy $\pi(\theta)$ and the initial state-action distribution $\nu$ satisfies*

$$\mathbb{E}_{(s,a)\sim \tilde{d}^\theta}\left[\bar{\phi}_{s,a}(\theta)(\bar{\phi}_{s,a}(\theta))^\top\right] \geq \mu \mathbf{I}_m, \tag{94}$$

*where $\mathbf{I}_m \in \mathbb{R}^{m\times m}$ is the identity matrix and $\mu > 0$, then*

$$\mathbb{E}\left[V_\rho(\pi^{(K)})\right] - V_\rho(\pi^*) \leq \left(1 - \frac{1}{\vartheta_\rho}\right)^K \frac{2}{1-\gamma} + \frac{(\vartheta_\rho + 1)\sqrt{C_\nu \epsilon_{\mathrm{approx}}}}{1-\gamma}$$
$$+ \frac{4\sqrt{C_\nu}\left(\vartheta_\rho + 1\right)}{(1-\gamma)^2\sqrt{T}}\left(\frac{2B^2}{\mu}\left(\sqrt{2m} + 1\right) + \sqrt{2m}\right).$$

Now we compare our Corollary 3 with Corollary 33 in Agarwal et al. (2021), which is their corresponding sample complexity results for NPG. The main differences between Corollary 3 and Corollary 33 in Agarwal et al. (2021) are similar to those for Q-NPG as summarized right after Corollary 1: Their sample complexity is $\mathcal{O}\big(\frac{1}{(1-\gamma)^{11}\epsilon^6}\big)$ while ours is $\tilde{\mathcal{O}}\big(\frac{1}{(1-\gamma)^5\epsilon^2}\big)$; they consider a projection

step for the iterates and incorrectly bound the stochastic gradient due to a similar error indicated in Footnote 3 (and see Appendix G.1 for more details), while we assume Fisher-non-degeneracy (94).

Compared to Corollary 1, the sample complexities for both Q-NPG and NPG are the same. The assumption (94) on the Fisher information matrix is much stronger than (31), as (31) is independent to the iterates. However, despite the difference of using $\nu$ instead of $\rho$, the Fisher-non-degeneracy (94) is commonly used in the optimization literature (Byrd et al., 2016; Gower et al., 2016; Wang et al., 2017) and in the RL literature (Liu et al., 2020; Ding et al., 2022; Yuan et al., 2022). It characterizes that the Fisher information matrix behaves well as a preconditioner in the NPG update (8). Indeed, (94) is directly assumed to be positive definite in the pioneering NPG work (Kakade, 2001) and in the follow-up works on natural actor-critic algorithms (Peters & Schaal, 2008; Bhatnagar et al., 2009). It is satisfied by a wide families of policies, including the Gaussian policy (Duan et al., 2016; Papini et al., 2018; Huang et al., 2020) and certain neural policy with log-linear policy as a special case. We refer to Liu et al. (2020, Section B.2) and Ding et al. (2022, Section 8) for more discussions on the Fisher-non-degenerate setting.

To prove Corollary 3, our approach is inspired from the proof of the sample complexity analysis of Liu et al. (2020, Theorem 4.9). That is, we require the Fisher-non-degeneracy (94) and apply Theorem 12 to the minimization of function $L_A(w, \theta, \tilde{d}^\theta)$ without relying on the boundedness of the stochastic gradient. A proof sketch is provided in Appendix G.1. Compared to their result, they obtain worse $\mathcal{O}\big(\frac{1}{(1-\gamma)^7 \epsilon^3}\big)$ sample complexity for NPG due to a slower $\mathcal{O}(1/k)$ convergence rate.

### G.1 PROOF OF COROLLARY 3

There is a similar remark for the proof of Corollary 3 to the one right before the proof of Corollary 1 in Appendix D.5. We notice that there is the same error occurred for the proof of NPG sample complexity analysis in Agarwal et al. (2021). Recall the stochastic gradient of $L_A$ in (44)

$$\widehat{\nabla}_w L_A(w, \theta, \tilde{d}^\theta) = 2 \left( w^\top \bar{\phi}_{s,a}(\theta) - \widehat{A}_{s,a}(\theta) \right) \bar{\phi}_{s,a}(\theta).$$

It turns out that $\widehat{\nabla}_w L_A(w, \theta, \tilde{d}^\theta)$ is unbounded, since the estimate $\widehat{A}_{s,a}(\theta)$ of $A_{s,a}(\theta)$ can be unbounded due to the unbounded length of the trajectory sampled in the sampling procedure, Algorithm 4. Thus, Agarwal et al. (2021) incorrectly verify $\widehat{\nabla} L_A(w, \theta, \tilde{d}^\theta)$ bounded by claiming that $\widehat{A}_{s,a}(\theta)$ is bounded by $\frac{2}{1-\gamma}$.

**Proof sketch.** Despite the difference of using either $\tilde{d}^\theta$ or $\bar{d}^\theta$ in the loss function $L_A$, we use the same assumptions of Liu et al. (2020), i.e., the Fisher-non-degeneracy (94) and the boundedness of the feature map, and verify all the conditions of Theorem 12 *without* relying on the boundedness of the stochastic gradient. In particular, similar to the proof of Corollary 1, we verify that $\mathbb{E}\big[\widehat{A}_{s,a}(\theta)^2\big]$ is bounded even though $\widehat{A}_{s,a}(\theta)$ is unbounded. To verify the condition (vi) in Theorem 12 in our proof, we use the same conditional expectation argument as in the proof of Corollary 1 to separate the correlated random variables $\widehat{A}_{s,a}(\theta)$ and $\bar{\phi}_{s,a}(\theta)$ with $(s,a) \sim \tilde{d}^\theta$ appeared in the stochastic gradient. Thanks to this argument, we fix a flaw in the previous proof of Liu et al. (2020, Proposition G.1) [7].

*Proof.* Similar to the proof of Corollary 1, we suppress the subscript $k$. First, the centered feature map is bounded by $\big\|\bar{\phi}_{s,a}(\theta)\big\| \leq 2B$. In order to apply Theorem 12, it remains to upper bound

---

[7] In a previous version of the proof in Section G, Liu et al. (2020, Proposition G.1) use the inequality

$$\mathbb{E}[(\widehat{A}_{s,a}(\theta) - w_\star^\top \bar{\phi}_{s,a}(\theta))^2 \bar{\phi}_{s,a}(\theta)(\bar{\phi}_{s,a}(\theta))^\top] \leq \mathbb{E}[(\widehat{A}_{s,a}(\theta) - w_\star^\top \bar{\phi}_{s,a}(\theta))^2]\mathbb{E}[\bar{\phi}_{s,a}(\theta)(\bar{\phi}_{s,a}(\theta))^\top],$$

which is incorrect since $\widehat{A}_{s,a}(\theta)$ and $\bar{\phi}_{s,a}(\theta)$ are correlated random variables. To fix it, we use the following conditional expectation computation argument

$$\mathbb{E}[(\widehat{A}_{s,a}(\theta) - w_\star^\top \bar{\phi}_{s,a}(\theta))^2 \bar{\phi}_{s,a}(\theta) \left(\bar{\phi}_{s,a}(\theta)\right)^\top] = \mathbb{E}[\mathbb{E}[(\widehat{A}_{s,a}(\theta) - w_\star^\top \bar{\phi}_{s,a}(\theta))^2 \mid s, a] \bar{\phi}_{s,a}(\theta)(\bar{\phi}_{s,a}(\theta))^\top],$$

and bound the term $\mathbb{E}\left[\left(\widehat{A}_{s,a}(\theta) - w_\star^\top \bar{\phi}_{s,a}(\theta)\right)^2 \mid s, a\right]$ in (95).

$\mathbb{E}\big[\|\widehat{A}_{s,a}(\theta)\bar{\phi}_{s,a}(\theta)\|^2\big]$ and $\|w_\star\|$ with $w_\star \in \operatorname{argmin}_w L_A(w, \theta, \tilde{d}^\theta)$, and find $\sigma > 0$ such that

$$\mathbb{E}\left[\left(\widehat{A}_{s,a}(\theta) - w_\star^\top \bar{\phi}_{s,a}(\theta)\right)^2 \mid s, a\right] = \mathbb{E}\left[\left(\widehat{A}_{s,a}(\theta)\right)^2 \mid s, a\right] \tag{95}$$

$$- 2A_{s,a}(\theta)w_\star^\top \bar{\phi}_{s,a}(\theta) + \left(w_\star^\top \bar{\phi}_{s,a}(\theta)\right)^2 \leq \sigma^2 \tag{96}$$

holds for all $(s,a) \in \mathcal{S} \times \mathcal{A}$ and $\theta \in \mathbb{R}^m$.

Similar to the proof of Corollary 1, the closed form solution of $w_\star$ can be written as

$$w_\star = \left(\mathbb{E}_{(s,a)\sim\tilde{d}^\theta}\left[\bar{\phi}_{s,a}(\theta)\bar{\phi}_{s,a}(\theta)^\top\right]\right)^\dagger \mathbb{E}_{(s,a)\sim\tilde{d}^\theta}\left[Q_{s,a}(\theta)\bar{\phi}_{s,a}(\theta)\right].$$

From (94), we have

$$\|w_\star\| \leq \frac{2B}{\mu(1-\gamma)}.$$

Now we need to upper bound $\mathbb{E}\left[\left(\widehat{A}_{s,a}(\theta)\right)^2 \mid s, a\right]$ from (95). Indeed, by using $\widehat{A}_{s,a}(\theta) = \widehat{Q}_{s,a}(\theta) - \widehat{V}_s(\theta)$, we have

$$\mathbb{E}\left[\left(\widehat{A}_{s,a}(\theta)\right)^2 \mid s, a\right] \leq 2\mathbb{E}\left[\left(\widehat{Q}_{s,a}(\theta)\right)^2 \mid s, a\right] + 2\mathbb{E}\left[\left(\widehat{V}_{s,a}(\theta)\right)^2 \mid s, a\right]$$

$$\overset{(74)}{\leq} \frac{8}{(1-\gamma)^2}, \tag{97}$$

where the last line is obtained, as $\mathbb{E}\left[\left(\widehat{V}_{s,a}(\theta)\right)^2 \mid s, a\right]$ shares the same upper bound (74) of $\mathbb{E}\left[\left(\widehat{Q}_{s,a}(\theta)\right)^2 \mid s, a\right]$ by using the similar argument.

From (97) and $\bar{\phi}_{s,a}(\theta) \leq 2B$, we verify $\mathbb{E}\left[\left\|\widehat{A}_{s,a}(\theta)\bar{\phi}_{s,a}(\theta)\right\|^2\right]$ bounded as well.

By using the upper bounds of $\mathbb{E}\left[\left(\widehat{A}_{s,a}(\theta)\right)^2 \mid s, a\right]$, $\|w_\star\|$, $|A_{s,a}(\theta)| \leq \frac{2}{1-\gamma}$ and $\left\|\bar{\phi}_{s,a}(\theta)\right\| \leq 2B$, the left hand side of (95) is upper bounded by

$$\mathbb{E}\left[\left(\widehat{A}_{s,a}(\theta) - w_\star^\top \bar{\phi}_{s,a}(\theta)\right)^2 \mid s, a\right] \leq \frac{8}{(1-\gamma)^2} + \frac{16B^2}{\mu(1-\gamma)^2} + \frac{16B^4}{\mu^2(1-\gamma)^2}$$

$$= \frac{4}{(1-\gamma)^2}\left(\left(\frac{2B^2}{\mu}+1\right)^2 + 1\right)$$

$$\leq \frac{8}{(1-\gamma)^2}\left(\frac{2B^2}{\mu}+1\right)^2.$$

Thus, we choose

$$\sigma = \frac{2\sqrt{2}}{1-\gamma}\left(\frac{2B^2}{\mu}+1\right).$$

Now all the conditions (i) - (vi) in Theorem 12 are verified. The reminder of the proof follows that of Corollary 1. $\qquad\square$

## H  DISCUSSION ON THE DISTRIBUTION MISMATCH COEFFICIENTS, THE CONCENTRABILITY COEFFICIENTS AND THE INCREASING STEP SIZE

We have already mentioned in the comparison with Agarwal et al. (2021) right after Theorem 1 that, although we have linear convergence rates, the magnitude of our error floor is worse (larger) by a factor of $\vartheta_\rho\sqrt{C_\rho}$ ($\vartheta_\rho\sqrt{C_\nu}$ for Theorem 3 and 4), due to the concentrability $C_\rho$ and the distribution

mismatch coefficients $\vartheta_\rho$ used in our proof. Such difference comes from different nature of the proof techniques. Here the distribution mismatch coefficients $\vartheta_\rho$ and the concentrability coefficients $C_\rho$ and $C_\nu$ are potentially large in our convergence theories. We give extensive discussions on them, respectively. We also provide further noteworthy observations about the connection between the use of the increasing step size and the policy iteration.

**Distribution mismatch coefficients $\vartheta_\rho$.** Our distribution mismatch coefficient $\vartheta_\rho$ in (20) is the same as the one in Xiao (2022). It contains both an upper bound and a lower bound. The linear convergence rate in our theories is $1 - \frac{1}{\vartheta_\rho} > 0$. Thus, the smaller $\vartheta_\rho$ is, the faster the resulting linear convergence rate. The best linear convergence rate is achieved when $\vartheta_\rho$ achieves its lower bound. Here our analysis is general that it includes all the distribution mismatch coefficient $\vartheta_\rho$ induced by any target state distribution $\rho$. Our results generalizes and sometimes also improves with respect to prior results.

A very pessimistic and trivial upper bound on $\vartheta_\rho$ is

$$\vartheta_\rho \leq \frac{1}{(1-\gamma)\rho_{\min}}.$$

However, if the target state distribution $\rho \in \Delta(\mathcal{S})$ does not have full support, i.e., $\rho_s = 0$ for some $s \in \mathcal{S}$, then $\vartheta_\rho$ might be infinite from this upper bound. Xiao (2022) just assumes that $\vartheta_\rho$ is finite. We further propose a solution to this particular issue. Indeed, if $\rho$ does not have full support, consider $\pi^*$ as an optimal policy. We can always convert the convergence guarantees for some state distribution $\rho' \in \Delta(\mathcal{S})$ with full support, i.e., $\rho'_s > 0$ for all $s \in \mathcal{S}$ as follows:

$$
\begin{aligned}
V_\rho(\pi^{(k)}) - V_\rho(\pi^*) &= \sum_{s \in \mathcal{S}} \rho_s \left( V_s(\pi^{(k)}) - V_s(\pi^*) \right) = \sum_{s \in \mathcal{S}} \frac{\rho_s}{\rho'_s} \rho'_s \left( V_s(\pi^{(k)}) - V_s(\pi^*) \right) \\
&\leq \left\| \frac{\rho}{\rho'} \right\|_\infty \sum_{s \in \mathcal{S}} \rho'_s \left( V_s(\pi^{(k)}) - V_s(\pi^*) \right) = \left\| \frac{\rho}{\rho'} \right\|_\infty \left( V_{\rho'}(\pi^{(k)}) - V_{\rho'}(\pi^*) \right).
\end{aligned}
$$

Then we only need convergence guarantees of $V_{\rho'}(\pi^{(k)}) - V_{\rho'}(\pi^*)$ for arbitrary $\rho'$ obtained from all our convergence analysis above. In this case, the linear convergence rate depends on

$$\vartheta_{\rho'} \overset{\text{def}}{=} \frac{1}{1-\gamma} \left\| \frac{d^{\pi^*}(\rho')}{\rho'} \right\|_\infty < \infty.$$

Equation (20) provides the lower bound $\frac{1}{1-\gamma}$ for $\vartheta_\rho$. Such lower bound can be achieved when the target state distribution $\rho$ satisfies that $\rho = d^{\pi^*}(\rho)$ where $\pi^*$ is an optimal policy. The advantage of this case is that, not only it implies the best linear convergence rate, more importantly, the fast linear convergence rate is known to be $\gamma$. So we know the convergence rate explicitly without any estimation, even though the optimal policy or the policy iterates are unknown before training. Hence, we know when to stop running the algorithm. Lan (2022) only considers the case when $\rho = d^{\pi^*}(\rho)$ and we are able to recover the same linear convergence rate $\gamma$ in their result.

Furthermore, the convergence performance $V_\rho(\pi^{(k)}) - V_\rho(\pi^*)$ depends on the target state distribution $\rho$. If the optimal policy $\pi^*$ is independent to the target state distribution $\rho$ which is usually the case in RL problems, then we are always allowed to fix $\rho = d^{\pi^*}(\rho)$ for the analysis without knowing $\rho$ and $\pi^*$ and derive this best linear convergence performance with rate $\gamma$, because we use the initial state-action distribution $\nu$ in training which is independent to $\rho$.

Finally, from (20), if $d^{(k)}$ converges to $d^*$, then $\vartheta_k$ converges to 1. This might imply superlinear convergence results as Section 4.3 in Xiao (2022). In this case, the notion of the distribution mismatch coefficients $\vartheta_\rho$ no longer exists for the superlinear convergence analysis. In other words, it is no longer concerned.

**Concentrability coefficients $C_\nu$.** The issue of having (potentially large) concentrability coefficients is unavoidable in all the fast linear convergence analysis of the inexact NPG that we are aware of, including even the tabular setting (e.g., Lan (2022) and Xiao (2022)) and the log-linear policy setting (Cayci et al. (2021), Chen & Theja Maguluri (2022) and ours).

First, in the fast linear convergence analysis of inexact NPG, the concentrability coefficients appear from the errors, including the statistical error and the approximation error. Thus, one way to avoid having the concentrability coefficients appear is to consider the exact NPG in the tabular setting (See Theorem 10 in Xiao (2022)). Because the tabular setting makes no approximation error and the exact NPG makes no statistical error. We consider the *inexact* NPG with the *log-linear* policy. Consequently, we have the concentrability coefficients multiplied by both the statistical error $\epsilon_{\text{stat}}$ and the approximation error ($\epsilon_{\text{bias}}$ in Assumption 2 or $\epsilon_{\text{approx}}$ in Assumption 5 and 8).

To remove the concentrability coefficients, one has to make strong assumptions on the errors with the $L_\infty$ supremum norm. In the tabular setting, Lan (2022) and Xiao (2022) assume that $\|\widehat{Q}(\pi) - Q(\pi)\|_\infty \leq \epsilon_{\text{stat}}$. The cons of such strong assumption requires high sample complexity and is already explained in Appendix A.1. In the log-linear policy setting, Chen & Theja Maguluri (2022) assume that $\|Q_s(\theta^{(k)}) - \Phi w_\star^{(k)}\|_\infty \leq \epsilon_{\text{bias}}$ for the approximation error, which is a very strong assumption in the function approximation regime. Due to the supremum norm, $\epsilon_{\text{bias}}$ is unlikely to be small, especially for large action spaces. Under this strong assumption, Lan (2022), Xiao (2022) and Chen & Theja Maguluri (2022) are able to eliminate the concentrability coefficients. To avoid assuming such strong assumptions, Cayci et al. (2021) and our paper consider the expected $L_2$ errors in the log-linear policy setting, which are much weaker assumptions, especially much more reasonable for the approximation error $\epsilon_{\text{bias}}$ compared to the one in Chen & Theja Maguluri (2022). The tradeoff is that, the concentrability coefficients can not be eliminated in this case both in Cayci et al. (2021) and our results.

Furthermore, as mentioned right after Theorem 4, under the expected error assumptions (Assumption 7 and 8), our concentrability coefficient $C_\nu$ is better presented than the one in Assumption 2 in Cayci et al. (2021) in the sense that it is independent to the policies throughout the iterations thanks to the use of $\tilde{d}^{(k)}$ instead of $\bar{d}^{(k)}$ (which is mentioned in Remark 1 as well) and is controllable to be finite by $\nu$, while the one in Cayci et al. (2021) depends on the iterates, thus is unknown and is not guaranteed to be finite.

Finally, like the distribution mismatch coefficient, the upper bound of $C_\nu$ in (30) is very pessimistic. By the definition of $C_\nu$ in (28), one can expect that $C_\nu$ is closed to 1, when $\pi^{(k)}$ and $\pi^{(k+1)}$ converge to $\pi^*$ with $\pi^*$ the optimal policy.

So our concentrability coefficient $C_\nu$ is the "best" one among all concentrability coefficients in the sense that, it takes the weakest assumptions on errors compared to Lan (2022), Xiao (2022) and Chen & Theja Maguluri (2022), it does not impose any restrictions on the MDP dynamics compared to Cayci et al. (2021) and it can be controlled to be finite by $\nu$ when other concentrability coefficients are infinite (Scherrer, 2014).

It is still an open question whether we can obtain fast linear convergence results of the inexact NPG in the log-linear policy setting, with small error floor and a much improved concentrability coefficient, e.g., as the same magnitude as the one in Agarwal et al. (2021).

**Increasing step size and the connection with policy iteration.** As for the use of the increasing step size, intuitively, the reason is that both NPG in (17) and Q-NPG in (16) behave more and more like policy iteration. For instance, when $\eta_k \to \infty$ and we replace the linear approximation $\Phi_s w^{(k)}$ by $Q_s(\theta^{(k)})$, (16) becomes

$$\pi_s^{(k+1)} = \arg\min_{p \in \Delta(\mathcal{A})} \left\{ \left\langle Q_s(\theta^{(k)}), p \right\rangle \right\}, \quad \forall s \in \mathcal{S},$$

which is exactly the classical Policy Iteration method (e.g., Puterman, 1994; Bertsekas, 2012). We refer to Xiao (2022, Section 4.4) for more discussion on the connection with policy iteration.

# I STANDARD OPTIMIZATION RESULTS

In this section, we present the standard optimization results from Beck (2017); Xiao (2022); Bach & Moulines (2013) used in our proofs.

First, we present the closed form update of mirror descent with KL divergence on the simplex. We provide its proof for the completeness.

**Lemma 10** (Mirror descent on the simplex, Example 9.10 in Beck (2017))**.** *Let $g \in \mathbb{R}^n$ which will often be a gradient and let $\eta > 0$. For $p, q$ in the unit $n$-simplex $\Delta^n$, the mirror descent step with respect to the KL divergence*

$$\min_{p \in \Delta^n} \ \eta \langle g, p \rangle + D(p, q) \tag{98}$$

*is given by*

$$p = \frac{q \odot e^{-\eta g}}{\sum_{i=1}^{n} q_i e^{-\eta g_i}}, \tag{99}$$

*where $\odot$ is the element-wise product between vectors.*

*Proof.* The Lagrangian of (98) is given by

$$L(p, \mu, \lambda) = \eta \langle g, p \rangle + D(p, q) + \mu(1 - \sum_{i=1}^{n} p_i) - \sum_{i=1}^{n} \lambda_i p_i,$$

where $\mu \in \mathbb{R}$ and $\lambda \in \mathbb{R}^n$ with non-negative coordinates are the Lagrangian multipliers. Thus the Karush–Kuhn–Tucker conditions are given by

$$\eta g + \log(p/q) + \mathbf{1}_n = \mu \mathbf{1}_n + \lambda,$$
$$\mathbf{1}_n^\top p = 1,$$
$$\lambda_i = 0 \text{ or } p_i = 0, \qquad \text{for all } i = 1, \cdots, n,$$

where the division $p/q$ is element-wise. Isolating $p$ in the top equation gives

$$p = q \odot e^{(\mu-1)\mathbf{1}_n + \lambda - \eta g} = e^{\mu-1} q \odot e^{\lambda - \eta g}.$$

Using the second constraint $\mathbf{1}_n^\top p = 1$ gives that

$$1 = e^{\mu-1} \sum_{i=1}^{n} q_i e^{\lambda_i - \eta g_i} \implies e^{\mu-1} = \frac{1}{\sum_{i=1}^{n} q_i e^{\lambda_i - \eta g_i}}.$$

Consequently, by plugging the above term into $p$, we have that

$$p = \frac{q \odot e^{\lambda - \eta g}}{\sum_{i=1}^{n} q_i e^{\lambda_i - \eta g_i}}.$$

It remains to determine $\lambda$. If $q_i = 0$ then $p_i = 0$ and thus $\lambda_i > 0$. Conversely, if $q_i > 0$ then $p_i > 0$ and thus $\lambda_i = 0$. In either of these cases, we have that the solution is given by (99). $\qquad\square$

Now we present the three-point descent lemma on proximal optimization with Bregman divergences, which is another key ingredient for our PMD analysis. Following Xiao (2022, Lemma 6), we adopt a slight variation of Lemma 3.2 in Chen & Teboulle (1993). First, we need some technical conditions.

**Definition 10** (Legendre function, Section 26 in Rockafellar (1970))**.** *We say a function $h$ is of* Legendre type *or a* Legendre function *if the following properties are satisfied:*

    *(i) $h$ is strictly convex in the relative interior of $\operatorname{dom} h$, denoted as $\operatorname{rint} \operatorname{dom} h$.*

    *(ii) $h$ is essentially smooth, i.e., $h$ is differentiable in $\operatorname{rint} \operatorname{dom} h$ and, for any boundary point $x_b$ of $\operatorname{rint} \operatorname{dom} h$, $\lim_{x \to x_b} \|\nabla h(x)\| \to \infty$ where $x \in \operatorname{rint} \operatorname{dom} h$.*

**Definition 11** (Bregman divergence (Bregman, 1967; Censor & Zenios, 1997))**.** *Let $h : \operatorname{dom} h \to \mathbb{R}$ be a Legendre function and assume that $\operatorname{rint} \operatorname{dom} h$ is nonempty. The* Bregman divergence $D_h(\cdot, \cdot) : \operatorname{dom} h \times \operatorname{rint} \operatorname{dom} h \to [0, \infty)$ *generated by $h$ is a distance-like function defined as*

$$D_h(p, p') \stackrel{def}{=} h(p) - h(p') - \langle \nabla h(p'), p - p' \rangle. \tag{100}$$

Under the above conditions, we have the following result. We also provide its proof for self-containment. (Xiao (2022) does not provide a formal proof.)

**Lemma 11** (Three-point decent lemma, Lemma 6 in Xiao (2022)). *Suppose that $\mathcal{C} \subset \mathbb{R}^m$ is a closed convex set, $f : \mathcal{C} \to \mathbb{R}$ is a proper, closed* [8] *convex function, $D_h(\cdot, \cdot)$ is the Bregman divergence generated by a function $h$ of Lengendre type and $\operatorname{rint} \operatorname{dom} h \cap \mathcal{C} \neq \emptyset$. For any $x \in \operatorname{rint} \operatorname{dom} h$, let*

$$x^+ \in \arg\min_{u \in \operatorname{dom} h \cap \mathcal{C}} \{f(u) + D_h(u, x)\}.$$

*Then $x^+ \in \operatorname{rint} \operatorname{dom} h \cap \mathcal{C}$ and for any $u \in \operatorname{dom} h \cap \mathcal{C}$,*

$$f(x^+) + D_h(x^+, x) \leq f(u) + D_h(u, x) - D_h(u, x^+).$$

*Proof.* First, we prove that for any $a, b \in \operatorname{rint} \operatorname{dom} h$ and $c \in \operatorname{dom} h$, the following identity holds:

$$D_h(c, a) + D_h(a, b) - D_h(c, b) = \langle \nabla h(b) - \nabla h(a), c - a \rangle. \tag{101}$$

Indeed, using the definition of $D_h$ in (100), we have

$$\langle \nabla h(a), c - a \rangle = h(c) - h(a) - D_h(c, a), \tag{102}$$
$$\langle \nabla h(b), a - b \rangle = h(a) - h(b) - D_h(a, b), \tag{103}$$
$$\langle \nabla h(b), c - b \rangle = h(c) - h(b) - D_h(c, b). \tag{104}$$

Subtracting (102) and (103) from (104) yields (101).

Next, since $h$ is of Legendre type, we have $x^+ \in \operatorname{rint} \operatorname{dom} h \cap \mathcal{C}$. Otherwise, $x^+$ is a boundary point of $\operatorname{dom} h$. From the definition of Legendre function, $\|\nabla h(x^+)\| = \infty$ which is not possible, as $x^+$ is also the minimum point of $f(u) + D_h(u, x)$. By the first-order optimality condition, we have

$$\left\langle u - x^+, g^+ + \nabla_y D_h(y, x)|_{y = x^+} \right\rangle \geq 0,$$

where $g^+ \in \partial f(x^+)$ is the subdifferential of $f$ at $x^+$. From the definition of $D_h$, the above inequality is equivalent to

$$\left\langle u - x^+, \nabla h(x^+) - \nabla h(x) \right\rangle \geq \left\langle x^+ - u, g^+ \right\rangle. \tag{105}$$

Besides, plugging $c = u, a = x^+$ and $b = x$ into (101), we obtain

$$\left\langle u - x^+, \nabla h(x^+) - \nabla h(x) \right\rangle = D_h(u, x) - D_h(u, x^+) - D_h(x^+, x) \overset{(105)}{\geq} \left\langle x^+ - u, g^+ \right\rangle.$$

Rearranging terms and adding $f(u)$ on both sides, we have

$$D_h(u, x) - D_h(u, x^+) + f(u) \geq f(u) + \left\langle x^+ - u, g^+ \right\rangle + D_h(x^+, x)$$
$$\geq f(x^+) + D_h(x^+, x),$$

which concludes the proof. The last inequality is obtained by the convexity of $f$ and $g^+ \in \partial f(x^+)$. $\square$

Finally, we use the following linear regression analysis for the proof of our sample complexity results, i.e., Corollary 1 and 3.

**Theorem 12** (Theorem 1 in Bach & Moulines (2013)). *Consider the following assumptions:*

*(i) $\mathcal{H}$ is a $m$-dimensional Euclidean space.*

*(ii) The observations $(x_n, z_n) \in \mathcal{H} \times \mathcal{H}$ are independent and identically distributed.*

*(iii) $\mathbb{E}\left[\|x_n\|^2\right]$ and $\mathbb{E}\left[\|z_n\|^2\right]$ are finite. The covariance $\mathbb{E}\left[x_n x_n^\top\right]$ is assumed invertible.*

*(iv) The global minimum of $f(\theta) = \frac{1}{2}\mathbb{E}\left[\langle \theta, x_n \rangle^2 - 2 \langle \theta, z_n \rangle\right]$ is attained at a certain $\theta_* \in \mathcal{H}$. Let $\xi_n = z_n - \langle \theta_*, x_n \rangle x_n$ denote the residual. We have $\mathbb{E}[\xi_n] = 0$.*

---

[8] A convex function $f$ is proper if $\operatorname{dom} f$ is nonempty and for all $x \in \operatorname{dom} f$, $f(x) > -\infty$. A convex function is closed, if it is lower semi-continuous.

(v) *Consider the stochastic gradient recursion defined as*

$$\theta_n = \theta_{n-1} - \eta(\langle \theta_{n-1}, x_n \rangle x_n - z_n),$$

*started from $\theta_0 \in \mathcal{H}$ and also consider the averaged iterates $\theta_{\text{out}} = \frac{1}{n+1} \sum_{k=0}^{n} \theta_k$.*

(vi) *There exists $R > 0$ and $\sigma > 0$ such that $\mathbb{E}\left[\xi_n \xi_n^\top\right] \leq \sigma^2 \mathbb{E}\left[x_n x_n^\top\right]$ and $\mathbb{E}\left[\|x_n\|^2 x_n x_n^\top\right] \leq R^2 \mathbb{E}\left[x_n x_n^\top\right]$.*

*When $\eta = \frac{1}{4R^2}$, we have*

$$\mathbb{E}\left[f(\theta_{\text{out}}) - f(\theta_*)\right] \leq \frac{2}{n} \left(\sigma \sqrt{m} + R \|\theta_0 - \theta_*\|\right)^2. \tag{106}$$

