# OpenReview forum: "Linear Convergence of Natural Policy Gradient Methods with Log-Linear Policies"
_ICLR.cc/2023/Conference — ICLR 2023 poster_

### Official Review · Reviewer_NuUP · 2022-10-24

**Confidence:** 4
**Correctness:** 4
**Technical Novelty And Significance:** 3
**Empirical Novelty And Significance:** Not applicable
**Recommendation:** 6

**Clarity, Quality, Novelty And Reproducibility:**

The paper has a clear presentation, with new results on the analysis of policy gradient methods in the linear function approximation setting.

**Strength And Weaknesses:**

Strength:
The paper is technically correct, and addresses the existing literature sufficiently. In addition to showing an improved convergence rate for the policy optimization, the paper also exploits a faster convergence result for the policy evaluation procedure, which seems the crucial factor in obtaining the final $O(1/\epsilon^2)$ complexity.

Weakness:
The only comment and suggestion that I have is the presentation of assumptions. There are some assumptions that seem less clearly motivated (e.g. Assumptions 3 and 4). Although some of these appear also in prior literature, it would be beneficial to provide some more context on why/where certain assumptions are needed.

Minor comment: The policy evaluation (e.g., Algorithm 6) seems to require the state-action pairs (Line 2, Algorithm 6) to be independent for different timesteps. Can the author comment on the feasibility of using Markovian sampling?

**Summary Of The Paper:**

This paper proposes linearly converging variants of NPG and Q-NPG methods, with log-linear policy class and compatible function approximation. The convergence results bare a similar spirit as the recent development of policy gradient methods in the tabular setting. In the stochastic setting, the authors establish an $O(1/\epsilon^2)$ sample complexity by showing an $O(1/T)$ convergence rate for the policy evaluation. Both of these results seem to improve upon previous results on policy gradient (NPG) method in the linear function approximation setting.

**Summary Of The Review:**

I appreciate the technical effort this paper puts into developing new understandings of how natural policy gradient converges with compatible function approximation, yielding a linearly converging policy optimization error and $O(1/\epsilon^2)$ sample complexity. The analysis bears certain novelty in terms of two perspectives. First, the analysis of policy optimization with compatible and linear function approximation is established, while accounting for both the error in policy evaluation and function approximation error. Second, the policy evaluation analysis exploits a sharper result than the one used in prior literature, which is needed for establishing the $O(1/\epsilon^2)$ sample complexity.

I believe the paper contains enough new results to the field of policy gradient methods.

---

> ### Author Response · Authors · 2022-11-18
> **Answer for Reviewer NuUP**
>
> Thank you for your positive and encouraging remarks. We hope the answers in **Differences between two analysis of Q-NPG** and below address your questions.
>
> **Feasibility of using Markovian sampling.** Thank you for your suggestion. Notice that our paper obtains $w^{(k)}$ by a regression solver. One can also use temporal difference (TD) learning (e.g., Chen et al. (2022)) with Markovian sampling to achieve similar $O(1/\epsilon^2)$ sample complexity result. Indeed, the performance analysis of TD learning will be expressed for $\epsilon_\mathrm{stat}$, which directly imply the total sample complexity results through our theorems. We have added this discussion in a new Appendix A.3 for the future work.
>
> Zaiwei Chen and Siva Theja Maguluri. Sample complexity of policy-based methods under off-policy sampling and linear function approximation, 2022.

---

> > ### Comment · Reviewer_NuUP · 2022-11-27
> > **Thank you for your response and clarifications**
> >
> > I would like to thank authors for providing responses to my previous questions on policy evaluation, together with detailed discussions on the difference of analysis for the two NPG variants in this paper. My questions have been sufficiently addressed, and I intend to keep my score toward accepting this paper.

---

### Official Review · Reviewer_sgtm · 2022-10-25

**Confidence:** 3
**Correctness:** 4
**Technical Novelty And Significance:** 2
**Empirical Novelty And Significance:** Not applicable
**Recommendation:** 5

**Clarity, Quality, Novelty And Reproducibility:**

The paper is well-written and easy to read. For the novelty, please refer to my previous comments.

**Strength And Weaknesses:**

Strength:
  The paper shows the both the natural policy gradient (NPG) and the Q-NPG methods have linear convergence rate and \tilt{O}(1/{\epsilon}^2) sample complexities, which is an improvement to the limited tabular setting shown in the previous works.

Weakness:
  Although the result is new and is an improvement to the existing literature, the technics it uses are very incremental to the existing tabular setting one. By using linear function approximation and reformulating the resulted update rules as approximate versions of the policy mirror descent (PMD) method, it's very incremental to demonstrate the linear convergence rate results in this paper.

**Summary Of The Paper:**

The paper considers infinite-horizon discounted Markov decision process with finite state and action space. By re-formulating the natural policy gradient (NPG) and the Q-NPG methods as approximate versions of the policy mirror descent (PMD) method, the paper shows that both methods enjoy linear convergence rates and \tilt{O}(1/{\epsilon}^2) sample complexities with the help of the compatible function approximation.

**Summary Of The Review:**

The paper has improvement in terms of demonstrating the linear convergence rate of the natural policy gradient (NPG) and the Q-NPG methods with the log-linear policy for the linear function approximation setup in the infinite-horizon discounted Markov decision processes. However, the efforts are incremental considering the previous works on the policy mirror descent result on tabular setting as well as the linear function approximation used in similar settings.

---

> ### Author Response · Authors · 2022-11-18
> **Answer for Reviewer sgtm**
>
> We respectfully disagree with the reviewer about the technical novelty. We hope the answers on the **Technical contribution/novelty compared to Xiao (2022)** above clarify the significance of our contribution.

---

### Official Review · Reviewer_Cu4x · 2022-10-29

**Confidence:** 4
**Correctness:** 4
**Technical Novelty And Significance:** 2
**Empirical Novelty And Significance:** Not applicable
**Recommendation:** 6

**Clarity, Quality, Novelty And Reproducibility:**

The paper is written clearly, and it is easy to follow. The results seem to be sound.

**Strength And Weaknesses:**

Strength:
- The authors consider a wide range of assumptions.
- For these settings the authors establish a geometric rate of convergence, which is superior to the prior work with function approximation
- The authors establish a tight 1/epsilon^2 sample complexity
Weakness:
- Some of the constants, such as C_v, can be arbitrary bad, and might be in the cardinality of state*action space. Appearance of this constant in the upper bound besides the 1/(1-\gamma), can result in a very poor bound.
- The contribution of the paper is limited compared to the prior work, specially Xiao (2022). It seems just a direct extension of Xiao (2022) theorem 10 to the linear function approximation, where we through all the errors due to function approximation to a big constant in the upper bound.
- The error terms appearing on the upper bound of Theorem 4.7 can be very loose. In particular, we have a very bad dependency in terms of 1/(1-\gamma). Can the authors comment on (possible) lower bounds of NPG or Q-NPG in terms of 1/(1-\gamma)?

**Summary Of The Paper:**

This paper studies the geometric convergence of NPG and Q-NPG parametrized by log linear function. The authors assume that approximation of Q-functions by linear function, and approximation of policy with log-linear function has some error bound, and they show that employing NPG or Q-NPG with a geometrically increasing step size results in exponential convergence to a ball around the global optimum. This ball is proportional to these function approximation errors. For the Q-NPG algorithm, the authors state their result under two different set of assumptions, and for NPG, the authors consider one set of assumption. Furthermore, by a standard sample based analysis of their result, the authors show 1/epsilon^2 sample complexity of their algorithm, which matches the best known result for Q-learning.

**Summary Of The Review:**

I believe the paper is a minor extension of the prior work. The different set of assumptions that the authors consider are not all that different.
----------------------------------------------------------------------------------------------------------------
After reviewing the rebuttal, I believe there are interesting points compared to the prior works. Specifically, the extension of this work compared to Xiao (2022) is not direct, specifically for the sample complexity analysis. I am increasing my score.

---

> ### Author Response · Authors · 2022-11-18
> **Answer for Reviewer Cu4x**
>
> Thank you for your review and remarks. We hope the answers in **Technical contribution/novelty compared to Xiao (2022)** and **Poor concentrability coefficients** address your questions.
>
> **Lower bounds of NPG and Q-NPG in terms of $1/(1-\gamma)$ in Theorem 4.7.** There is no Theorem 4.7 in the paper, we do not know which error terms the reviewer is referring.

---

### Official Review · Reviewer_yaHy · 2022-11-05

**Confidence:** 4
**Correctness:** 4
**Technical Novelty And Significance:** 3
**Empirical Novelty And Significance:** Not applicable
**Recommendation:** 8

**Clarity, Quality, Novelty And Reproducibility:**

The paper is well-written and all claimed results are supported by rigorous proofs. However, since many terminologies are borrowed from the literature, e.g., compatible function approximation, relative condition number, concentrability coefficient, and transfer error, it is helpful if the authors could add more explanations. Also, the comparison with existing results seems to be very technical. It is helpful if the authors could make a table to list all pros/cons for clarity. Although linear rate results are interesting, the improved sublinear rate is also important and I recommend stating it in the main context.

The paper builds on the policy mirror descent analysis of Xiao, 2022 while a careful introduction of problem-dependent parameters and function approximation errors could be an additional novelty. The improved linear rates are more expected, but they do advance our understanding of the natural policy gradient method in the function approximation setting. Overall, the technical novelty is average.

For some other questions, please find them here for consideration.

- Why call Eq. (18) 'approximated NPG' or 'approximated mirror descent'? It can be confused with approximation in the policy update, which is not your case.

- Is the analysis restricted to the log-linear policy? How about other normalization of linear function approximation instead of using softmax functions?

- There is a discrepancy between the transfer error type bound and the approximation error type bound. It is not very clear which one is better.

- How do you choose features that satisfies Eq. (32)?

-  Is there a way to increase stepsize when the discount factor is unknown?

**Strength And Weaknesses:**

**Strengths**

- The paper studies log-linear policy class for natural policy gradient methods. This linear function approximation setting is important since it is a basic case of widely-used policy gradient methods using general neural policy parametrization

- The authors employ the policy mirror descent analysis from Xiao, 2022 in the tabular case to analyze the log-linear version of natural policy gradient methods in the function approximation setting. This optimization perspective is meaningful to practical RL methods since it offers a more careful characterization of problem-dependent parameters and algorithm stepsizes.

- The established convergence results improve the results from Agarwarl et al. 2021 from sublinear convergence to linear convergence (both up to some errors), with new choices of algorithm stepsizes. Achieving fast convergence is beneficial to training RL algorithms with better efficiency.


**Weaknesses**

- The established convergence rates depend on the distribution mismatch coefficients and the concentrability coefficients. Since these coefficients either depend on the optimal policy or the policy iterates that are unknown before training, it is hard to estimate them. It is not very clear how useful these convergence rates are in practice. It is useful to discuss how bad the rate could be, e.g., all these coefficients are chosen in the worst way.

- The distribution mismatch coefficient can be very large in large MDPs even when the initial state distribution is uniform. Large distribution mismatch coefficients lead to poor convergence rates and dependence on function approximation errors, simultaneously. It is useful to discuss the reason for this limitation, and the possibility to address it.

- The established theory seems to be unique to the log-linear policy class. It is more useful if the authors could discuss its extension to the general policy class.

- The paper hasn't supplied empirical evidence to demonstrate the practical use of the suggested theory. Since function approximation is ubiquitously employed in RL, it is useful to provide experimental results as additional support.

**Summary Of The Paper:**

The paper studies the analysis improvement of existing policy gradient methods. The authors apply the recent new analysis of policy mirror descent to natural policy gradient methods ( and its Q-based version ) under log-linear policy class. The provided convergence results improve the state-of-the-art rates, using either adaptively increasing stepsize or constant stepsize. The authors also present a sample-based algorithm and sample complexity which improves previously-known results in the same setting.

**Summary Of The Review:**

The paper studies the natural policy gradient methods ( and its Q-based version ) under log-linear policy class by applying the recent new analysis of policy mirror descent. The established convergence rates improve the state-of-the-art rates using either adaptively increasing stepsize or constant stepsize. However, due to the problem/algorithm-dependence, it is not very clear how useful these convergence rates are in practice. The authors also present a sample-based algorithm and sample complexity which improves previously-known results in the same setting. The paper hasn't supplied empirical evidence to demonstrate the practical use of the provided theory.

============================

POST-REBUTTAL. Thank you for your response. Since my concerns are addressed in the new draft, I am inclined to recommend acceptance. The score has been updated.

---

> ### Author Response · Authors · 2022-11-18
> **Answer for Reviewer yaHy (1/3) : Poor distribution mismatch coefficients**
>
> Thank you for your constructive remarks. We hope the answers in **Technical contribution/novelty compared to Xiao (2022)**, **Poor concentrability coefficients**, **Differences between two analysis of Q-NPG** and below address your questions.
>
> ### Poor distribution mismatch coefficients
>
> Our distribution mismatch coefficient $\vartheta_\rho$ in equation (21) is the same as the one in Xiao (2022). It contains both an upper bound and a lower bound.
> The linear convergence rate in our theories is $1 - \frac{1}{\vartheta_\rho} > 0$. Thus, the smaller $\vartheta_\rho$ is, the faster the resulting linear convergence rate. The best linear convergence rate is achieved when $\vartheta_\rho$ achieves its lower bound.
> Here our analysis is general that it includes all the distribution mismatch coefficient $\vartheta_\rho$ induced by any target state distribution $\rho$. Our results generalizes and sometimes also improves with respect to prior results.
>
> **1. The worst case with the upper bound.** First, a very pessimistic and trivial upper bound on $\vartheta_\rho$ is
> $$\vartheta_\rho \leq \frac{1}{(1-\gamma)\rho_{\min}}.$$
> However, if the target state distribution $\rho \in \Delta(\mathcal{S})$ does not have full support, i.e., $\rho_s=0$ for some $s\in\mathcal{S}$, then $\vartheta_\rho$ might be infinite from this upper bound. Xiao (2022) just assumes that $\vartheta_\rho$ is finite. We further propose a solution to this particular issue in Appendix H. Indeed, we can always convert the convergence guarantees to some state distribution $\rho'$ with full support such that $\vartheta_{\rho'}$ is finite.
>
> **2. The best case with the lower bound.** Equation (21) provides the lower bound $\frac{1}{1-\gamma}$ for $\vartheta_\rho$.
> It is mentioned in Appendix H that such lower bound can be achieved when the target state distribution $\rho$ satisfies that $\rho = d^{\pi^*}(\rho)$ where $\pi^*$ is an optimal policy.
> The advantage of this case is that, not only it implies the best linear convergence rate, more importantly, the fast linear convergence rate is known to be $\gamma$. So we know the convergence rate explicitly without any estimation, even though the optimal policy or the policy iterates are unknown before training. Hence, we know when to stop running the algorithm.
> Lan (2022) only considers the case when $\rho = d^{\pi^*}(\rho)$ and we are able to recover the same linear convergence rate $\gamma$ in their result.
>
> **3. The best case is a common setting.** Furthermore, the convergence performance $V_\rho(\pi^{(k)}) - V_\rho(\pi^*)$ depends on the target state distribution $\rho$. If the optimal policy $\pi^*$ is independent to the target state distribution $\rho$ which is usually the case in RL problems, we are always allowed to fix $\rho = d^{\pi^*}(\rho)$ for the analysis without knowing $\rho$ and $\pi^*$ and derive this best linear convergence performance with rate $\gamma$, because we use the initial state-action distribution $\nu$ in training which is independent to $\rho$.
>
> **4. Superlinear convergence is even better.** Finally, from equation (21), if $d^{(k)}$ converges to $d^*$, then $\vartheta_k$ converges to 1. This might imply superlinear convergence results as Section 4.3 in Xiao (2022). In this case, the notion of the distribution mismatch coefficients $\vartheta_\rho$ no longer exists for the superlinear convergence analysis. In other words, it is no longer concerned.
>
> We have included the above discussion of the distribution mismatch coefficients to complete Appendix H.
>
> Lin Xiao. On the convergence rates of policy gradient methods, 2022.
>
> Guanghui Lan. Policy mirror descent for reinforcement learning: linear convergence, new sampling complexity, and generalized problem classes, 2022.

---

> > ### Author Response · Authors · 2022-11-18
> > **Answer for Reviewer yaHy (2/3): Extension to the general policy class.**
> >
> > Regarding _future work_, one can extend our analysis to general policy classes using a similar compatible function approximation framework. Concretely, consider the parameterized policy
> > $$\pi_{s,a}(\theta) = \frac{\exp(f_{s,a}(\theta))}{\sum_{a' \in \mathcal{A}} \exp(f_{s,a'}(\theta))},$$
> > where $f_{s,a}(\theta)$ is parameterized by $\theta \in \mathbb{R}^m$ and is differential. As Agarwal et al. (2021) mentioned, the gradient can be written as
> > $$\nabla_\theta \log \pi_{s,a}(\theta) = g_{s,a}(\theta), \quad \quad \mbox{where} \quad  g_{s,a}(\theta) = \nabla_\theta f_{s,a}(\theta) - \mathbb{E}\left[ \nabla_\theta f_{s,a'}(\theta) \right].$$
> > The expectation is taken on $a' \sim \pi_s(\theta)$. The NPG update is equivalent to the following compatible function approximation framework
> > $$\theta^{(k+1)} = \theta^{(k)} - \eta_k w_\star^{(k)}, \quad \quad w_\star^{(k)} \in \arg\min_w \mathbb{E}\left[\left(A_{s,a}(\theta^{(k)}) - w^\top g_{s,a}(\theta^{(k)})\right)^2\right].$$
> > Here the expectation is taken on $(s,a) \sim \bar{d}^{(k)}$. As Alfano and Rebeschini (2022, Remark 4.8) mentioned, if we assume that for all $s,a$, function $f(\theta)$ satisfies
> > $$f_{s,a}(\theta^{(k+1)}) = f_{s,a}(\theta^{(k)}) - \eta_k (w_\star^{(k)})^\top g_{s,a}(\theta^{(k)}),$$
> > then one can easily verify that the NPG update resulted in a new policy is also equivalent to the policy mirror descent update
> > $$\pi_s^{(k+1)} = \arg\min_{p \in \Delta(\mathcal{A})}  \eta_k \left<G_s^{(k)}w^{(k)}, p\right>  + D(p, \pi_s^{(k)}) , \quad  \forall s \in \mathcal{S},$$
> > where $G_s^{(k)} \in \mathbb{R}^{|\mathcal{A}| \times m}$ is a matrix with rows $(g_{s,a}(\theta^{(k)}))^\top\in\mathbb{R}^{1 \times m}$ for $a\in\mathcal{A}$. Consequently, one can extend our work naturally in this general setting to derive linear convergence analysis for NPG.
> >
> > We have added this discussion in a new Appendix A.3 for the future work.
> >
> >
> > Carlo Alfano and Patrick Rebeschini. Linear convergence for natural policy gradient with log-linear policy parametrization, 2022.

---

> > > ### Author Response · Authors · 2022-11-18
> > > **Answer for Reviewer yaHy (3/3): Other comments**
> > >
> > > **Experimental results.** The focus of this paper was the theoretical analysis of NPG method. We agree that providing experimental results is important. Thus, the results we have obtained open up several experimental questions related to parameter settings for NPG and Q-NPG. We leave such questions as an important future work to further support our theoretical findings.
> > >
> > > **Table of comparison, more explanations for terminologies.** We agree and will add a table of comparison in the appendix. We have also added more explanations for terminologies, e.g., the visitation probabilities, the compatible function approximation, the relative condition number, the concentrability coefficient, and the transfer error, to better present our work.
> > >
> > > **Sublinear rate stated in the main text.** We agree and have stated the sublinear convergence of Q-NPG in Section 4.1.
> > >
> > > **Confusion about "approximated NPG" and "approximated PMD".** Thanks to your valuable comment, we now modify them by "inexact NPG" and "inexact PMD". See also the answer in **Change in terminology**.
> > >
> > > **Other normalization of linear function approximation.** This will be an interesting question for the future work. Perhaps one can consider the _exponential tilting_, a generalization of Softmax to more general probability distributions. Another interesting venue of investigation is to consider the _generalized linear model_ instead of linear function approximation for the $Q$ function and the advantage function. We have added this discussion in a new Appendix A.3 for the future work.
> > >
> > > **Conditions for equation (32).** We have mentioned this in the paragraph _Non-singularity of the covariance matrix $\Sigma_\nu$_ in Appendix D.4. For instance, it is shown in Proposition 3 in Cayci et al. (2021) that with $\nu$ chosen as uniform distribution over $\mathcal{S} \times \mathcal{A}$ and $\phi(s,a) \sim \mathcal{N}(0, I_m)$ sampled as Gaussian random features, equation (32) is guaranteed with high probability.
> > > More generally, with $m \ll |\mathcal{S}||\mathcal{A}|$, it is easy to find $m$ linearly independent $\phi(s,a)$ among all $|\mathcal{S}||\mathcal{A}|$ features such that the covariance matrix $\Sigma_\nu$ has full rank.
> > >
> > > **Is there a way to increase stepsize when the discount factor is unknown?** This is an excellent question. So far the PMD proof techniques used in Lan (2022), Xiao (2022) and ours require that the discount factor is known. Perhaps the work of Li et al. (2022) can help to find a way to increase stepsize when the discount factor is unknown. Indeed, Li et al. (2022) consider the averaged MDP setting. So there is no discount factor. They achieve linear convergence for NPG by increasing the stepsize with some regularization parameters. It will be interesting to investigate if the way of increasing stepsize in Li et al. (2022) can be applied in our setting. We have added this discussion in a new Appendix A.3 for the future work.
> > >
> > > Lin Xiao. On the convergence rates of policy gradient methods, 2022.
> > >
> > > Guanghui Lan. Policy mirror descent for reinforcement learning: linear convergence, new sampling complexity, and generalized problem classes, 2022.
> > >
> > > Semih Cayci, Niao He, and R. Srikant. Linear convergence of entropy-regularized natural policy gradient with linear function approximation, 2021.
> > >
> > > Tianjiao Li, Feiyang Wu, and Guanghui Lan. Stochastic first-order methods for average-reward markov decision processes, 2022.

---

### Author Response · Authors · 2022-11-18
**Answer for Reviewer yaHy and NuUP: Differences between two analysis of Q-NPG**

The first Q-NPG analysis in Section 4.1 uses a transfer error type bound. The motivation here is to make the comparison to Agarwal et al. (2021) easier with the same assumptions.

Then we move to the second Q-NPG analysis in Section 4.2 with the approximation error type bound.
Using this type of analogous analysis to the NPG analysis in Section 5 allows us to directly compare Q-NPG to NPG, and show that in this setting the methods enjoy the same convergence result.

In terms of the assumptions, the transfer error (Assumption 2) is weaker than the approximation error (Assumption 5), as mentioned right after Assumption 5. While the approximation error type bound in Theorem 2 has a weaker assumption for the concentrability coefficient and does not require the relative condition number (Assumption 3).

In terms of the convergence bound, we agree that it is not clear which one is better. We have clarified this right after Theorem 3.

Alekh Agarwal, Sham M. Kakade, Jason D. Lee, and Gaurav Mahajan. On the theory of policy gradient methods: Optimality, approximation, and distribution shift, 2021.

---

### Author Response · Authors · 2022-11-18
**Answer for Reviewer yaHy and Cu4x: Poor concentrability coefficients**

The issue of having theses (potentially large) concentrability coefficients is unavoidable in all the fast linear convergence analysis of the inexact NPG that we are aware of, including even the tabular setting (e.g., Lan (2022) and Xiao (2022)) and the log-linear policy setting (Cayci et al. (2021), Chen et al. (2022) and ours).

*  **1. Removing concentrability coefficients using exact updates in the tabular setting.** First, in the fast linear convergence analysis of inexact NPG, the concentrability coefficients appear from the errors, including the statistical error and the approximation error.
Thus, one way to avoid having the concentrability coefficients appear is to consider the exact NPG in the tabular setting (See Theorem 10 in Xiao (2022)).
Because the tabular setting makes no approximation error and the exact NPG makes no statistical error.  We consider the *inexact* NPG with the *log-linear* policy. Consequently, we have the concentrability coefficients multiplied by both the statistical error $\epsilon_\mathrm{stat}$ and the approximation error ($\epsilon_\mathrm{bias}$ in Assumption 2 or $\epsilon_\mathrm{approx}$ in Assumption 5 and 8).

*  **2. Removing concentrability coefficients using stronger assumptions.** To remove the concentrability coefficients, one has to make strong assumptions on the errors with the $L_\infty$ supremum norm. In the tabular setting, Lan (2022) and Xiao (2022) assume that $\|\widehat{Q}(\pi) - Q(\pi)\|_\infty \leq \epsilon_\mathrm{stat}$. The cons of such strong assumption requires high sample complexity and is already explained in the answer about the **Technical contribution/novelty compared to Xiao (2022)**.
In the log-linear policy setting, Chen et al. (2022) assume that $\|Q_s(\theta^{(k)}) - \Phi w_\star^{(k)}\|_\infty \leq \epsilon_\mathrm{bias}$ for the approximation error, which is a very strong assumption in the function approximation regime. Due to the supremum norm, $\epsilon_\mathrm{bias}$ is unlikely to be small, especially for large action spaces.
Under this strong assumption, Lan (2022), Xiao (2022) and Chen et al. (2022) are able to eliminate the concentrability coefficients.
To avoid assuming such strong assumptions, Cayci et al. (2021) and our paper consider the expected $L_2$ errors in the log-linear policy setting, which are much weaker assumptions, especially much more reasonable for the approximation error $\epsilon_\mathrm{bias}$ compared to the one in Chen et al. (2022). The tradeoff is that, the concentrability coefficients can not be eliminated in this case both in Cayci et al. (2021) and our results.

*  **3. Best current concentrability coefficients using expected error assumptions.**
Furthermore, as mentioned right after Theorem 4, under the expected error assumptions (Assumption 7 and 8), our concentrability coefficient $C_\nu$ is better presented than the one in Assumption 2 in Cayci et al. (2021) in the sense that it is independent to the policies throughout the iterations thanks to the use of $\tilde{d}^{(k)}$ instead of $\bar{d}^{(k)}$ (which is mentioned in Remark 1 as well) and is controllable to be finite by $\nu$. While the one in Cayci et al. (2021) depends on the iterates, thus is unknown and is not guaranteed to be finite.

*  **4.** Finally, we also emphasize that the upper bound of $C_\nu$ in equation (31) is very pessimistic. By the definition of $C_\nu$ in equation (29), one can expect that $C_\nu$ is closed to 1, when $\pi^{(k)}$ and $\pi^{(k+1)}$ converge to $\pi^*$ with $\pi^*$ the optimal policy.

So our concentrability coefficient $C_\nu$ is the “best” one among all concentrability coefficients in the sense that, it takes the weakest assumptions on errors compared to Lan (2022), Xiao (2022) and Chen et al. (2022), it does not impose any restrictions on the MDP dynamics compared to Cayci et al. (2021) and it can be controlled to be finite by $\nu$ when other concentrability coefficients are infinite.

It is still an open question whether we can obtain fast linear convergence results of the inexact NPG in the log-linear policy setting, with small error floor and a much improved concentrability coefficient, e.g., as the same magnitude as the one in Agarwal et al. (2021). We have included the above discussion of the concentrability coefficients in a new paragraph in Appendix H.

Lin Xiao. On the convergence rates of policy gradient methods, 2022.

Alekh Agarwal, et al. On the theory of policy gradient methods: Optimality, approximation, and distribution shift, 2021.

Guanghui Lan. Policy mirror descent for reinforcement learning: linear convergence, new sampling complexity, and generalized problem classes, 2022.

Semih Cayci, Niao He, and R. Srikant. Linear convergence of entropy-regularized natural policy gradient with linear function approximation, 2021.

Zaiwei Chen and Siva Theja Maguluri. Sample complexity of policy-based methods under off-policy sampling and linear function approximation, 2022.

---

### Author Response · Authors · 2022-11-18
**Answer for Reviewer yaHy, Cu4x and sgtm: Technical contribution/novelty compared to Xiao (2022) (1/2)**

**1.** Our linear convergence results (i.e., Theorem 1, 3 and 4) are not direct applications of Theorem 10 in Xiao (2022). Indeed, Xiao (2022) establishes the connection between NPG and a specific form of policy mirror descent (PMD) with the use of the weighted Bregman divergence for the tabular setting, while we show that this connection can also be established for the function approximation setting via the compatible function approximation framework. We also modify the PMD framework of Xiao (2022) with the linear approximation of the advantage function, inspired from the compatible function approximation framework. Thus, the approaches of deriving the PMD form update are different. Without this work of using the compatible function approximation framework to bridge NPG and PMD, it was not clear at all that the analysis of Xiao (2022) could be extended in the log-linear policy setting. This interpretation is acknowledged by Reviewer yaHy: "the paper builds on the policy mirror descent (PMD) analysis of Xiao (2022), while a careful introduction of problem-dependent parameters and function approximation errors could be an additional novelty." So our work is the first step of showing that the proof techniques used in Xiao(2022) can be extended in function approximation regime. In fact, the extension is highly nontrivial and requires significant innovation (see details below). As for future work, one can extend our work to other function approximation setting through a similar compatible function approximation framework. Please also read our answers to Reviewer yaHy for the extension of our work.

**2.** Besides, our linear convergence results (i.e., Theorem 1, 3 and 4) only consider the \emph{inexact} NPG update. Compared to Theorem 14 in Xiao (2022), which is their corresponding result on the inexact PMD method, we improve their analysis by making much weaker assumptions on the accuracy of the estimation $Q(\pi)$. Xiao (2022) requires an $L_\infty$ supremum norm bound on the estimation error of $Q$, i.e., $\| \widehat{Q}(\pi) - Q(\pi) \|_\infty \leq \epsilon_\mathrm{stat}$, whereas our convergence guarantee depends on the expected error of the estimate, i.e., Assumption 1 and 7. For instance, Assumption 1 from equation (63) can be written as
$\mathbb{E}\left[(\phi(s,a)^\top w^{(k)} - \phi(s,a)^\top w_\star^{(k)})^2\right] \leq \epsilon_\mathrm{stat}$, which can be interpreted as $\mathbb{E}\left[(\widehat{Q}(\pi) - Q(\pi))^2\right] \leq \epsilon_\mathrm{stat}$ under the linear approximation setting. The techniques for handling $L_\infty$ and $L_2$ errors are very different. Not only our assumption is weaker, it also benefits from the sample complexity analysis that we explain next.

**3.** Consequently, when considering the sample complexity results we derived for sample-based (Q)-NPG in Corollary 1 and 2, the difference between our work and Theorem 16 in Xiao (2022), which corresponds to their sample complexity results, is even more significant.
Corollary 1 with Algorithm Q-NPG-SGD satisfies Assumption 1 with a number of samples that depends only on the feature dimension $m$ of $\phi$ and does not depend on the cardinality of state space $|\mathcal{S}|$ or action space $|\mathcal{A}|$. In contrast, the assumption $\|\widehat{Q}(\pi) - Q(\pi)\|_\infty \leq \epsilon_\mathrm{stat}$ with the $L_\infty$ norm in Xiao (2022) causes the sample complexity to depend on $|\mathcal{S}||\mathcal{A}|$ in Theorem 16.

Furthermore, Xiao (2022) uses a Monte-Carlo approach with multiple independent rollouts per iteration, while our sample-based (Q)-NPG uses one single rollout (Algorithm 3 and 4) combined with regression solvers; Xiao (2022) derives a high probability sample complexity result, while we derive the convergence of the optimality gap which can guarantee that the variance of $V_\rho(\pi^{(K)})$ converges to zero. Thus, our sample-based algorithms had not been considered in Xiao (2022) and our proofs of Corollary 1 and 2 require a different approach.

---

> ### Author Response · Authors · 2022-11-18
> **Answer for Reviewer yaHy, Cu4x and sgtm: Technical contribution/novelty compared to Xiao (2022) (2/2)**
>
> As Reviewer NuUP mentioned, our sample complexity analysis regarding the policy evaluation is novel. Although the sample-based algorithms have been considered previously in Agarwal et al. (2021) and Liu et al. (2020), none of their analysis on the sample complexity was correct. Thus, we argue that our contributions on the detailed sample complexity analysis are significant.
> Indeed, Agarwal et al. (2021) required the boundedness of the stochastic gradient estimator, which might not hold as we extensively discussed in Section D.5. We fixed this by showing that $\mathbb{E}\left[\widehat{Q}_{s,a}(\theta)^2\right]$ is bounded. Please refer to Section D.5 for all the subtleties, including a proof sketch of Corollary 1.
> Liu et al. (2020) also incorrectly used an inequality where the random variables are correlated. See the detailed explanation (Footnote 7) in Section G.1. We fixed this error with a careful conditional expectation argument.
> Please refer to Section G.1 for all the details, including a proof sketch of Corollary 2.
>  These dimensions are where an important part of the technical work was done.
>  Therefore, outside of the tabular setting, and considering NPG methods that make use of a regression solver, our complexity analysis is currently the only analysis that is entirely correct that we are aware of.
>
> **4.** Finally we not only extend the work of Xiao (2022) to NPG for log-linear policy, but also consider the Q-NPG method and establish its linear convergence analysis. This is a method that is unique to log-linear policy and again had not been considered in Xiao (2022).
>
> We have included the above discussion of differences between our paper and Xiao (2022) in a new Appendix A.1 in the appendix.
>
>  Lin Xiao. On the convergence rates of policy gradient methods, 2022.
>
>  Yanli Liu, Kaiqing Zhang, Tamer Basar, and Wotao Yin. An improved analysis of (variance-reduced) policy gradient and natural policy gradient methods, 2020.
>
>  Alekh Agarwal, Sham M. Kakade, Jason D. Lee, and Gaurav Mahajan. On the theory of policy gradient methods: Optimality, approximation, and distribution shift, 2021.

---

### Author Response · Authors · 2022-11-18
**Dear reviewers and Change in terminology**

**Dear reviewers,** thank you for taking the time to review our paper and your careful remarks.
To facilitate comparing our revised submission to the original, we have highlighted in red the major changes that we have committed to the manuscript.
We address general questions first
and reviewer-specific remarks below. We will integrate all reviewers' suggestions to improve the clarity of the paper.
Based on the comments of the reviewers, we have committed a change in terminology throughout the paper which we explain next. We also use this new terminology in our response.

**Change in terminology.** Thanks to the comments of Reviewer yaHy, to avoid confusion in terminology with the function approximation in the policy update, throughout the paper we consider the _inexact NPG_ and _inexact policy mirror descent_ (PMD) updates instead of "approximate NPG" and "approximate PMD". The NPG and PMD we considered in the paper are inexact, because $w^{(k)}$ is an _inexact_ evaluation of $w_\star^{(k)}$, obtained by a sample-based regression problem solver. Notice that the terminology "inexact PMD" is also used in Xiao (2022) in the same manner.

Lin Xiao. On the convergence rates of policy gradient methods, 2022.

---

### Decision · Program_Chairs · 2023-01-20

**Decision:**

Accept: poster

**Justification For Why Not Higher Score:**

NA

**Justification For Why Not Lower Score:**

NA

**Metareview: Summary, Strengths And Weaknesses:**

This work studies  natural policy gradient algorithms to solve Markov Decision Problems when the policy is a log-linear parameterization in terms of features. A specific inexact model that incorporates stochastic approximation error is studied in contrast to prior works. The key contribution is the establishment of linear convergence rate under a set of conditions that are strictly weaker than prior work.

Strengths are in contributing a novel analysis technique that exploits compatible function approximation, the extension to the log-linear policy parameterization, and the relaxation of the prior assumption on the infinity norm approximation error of the Q function estimator.

Weaknesses are in the fact that the problem class is extremely similar to prior work, and the the contribution of the work is clearly in the minutiae of the convergence analysis of natural policy gradient methods. In particular, policy mirror ascent is already well-studied, and the main contribution is the study of the log-linear class under compatible function approximation.

**Note From Pc:**

if the above contains the word "oral" or "spotlight" please see: "oral" presentation means -> notable-top-5% and "spotlight" means -> notable-top-25%. As stated in our emails, we are disassociating presentation type from AC recommendations

**Summary Of Ac-Reviewer Meeting:**

NA